



# River ice phenology and thickness from satellite altimetry. Potential for ice bridge road operation.

Elena    Zakharova[1,6],    Svetlana    Agafonova[2],    Claude    Duguay[3,4],    Natalia    Frolova[2],
Alexei Kouraev[5]

*1. IWP RAS, Moscow, Russia,*

*2. MSU, Moscow, Russia,*

*3. University of Waterloo, Waterloo, Canada*

*4. H2O Geomatics, Waterloo, Canada*

*5. LEGOS/OMP, Toulouse, France*

*6. EOLA, Toulouse, France*

**Abstract.**

River ice is an important component of land cryosphere. Satellite monitoring of river ice is
rapidly developing scientific area with an important outcome for many climate, environmental
and socio-economic applications. Radar altimetry, now widely used for monitoring of river water
regime, demonstrates a good potential for observation of river ice phenology and for an
estimation of river ice thickness. Jason-2 and -3 Ku-band backscatter measurements are sensitive
enough for detection of first appearance of the ice and of beginning of thermal ice degradation
on the Lower Ob River (Western Siberia). Uncertainties of the altimetric ice events timing are
less than 10 days for 88-90% of cases. River ice thickness retrieved from altimetric
measurements via empirical relations with in situ observations, has an accuracy (expressed as
RMSE) varying from 0.07 to 0.18 m. We demonstrated that using satellite altimetry the dates of
ice road opening at Salekhard city can be predicted quite accurately with 4 days delay.
Uncertainties for the prediction of dates of the ice road closure are of 3 days with the delay
varying from 4 days (for late melting start) to 22 days (for yearly melting start).

## 1   Introduction

River ice is a major component of the global cryosphere and hydrosphere, and its monitoring is
important for many environmental, climate  and societal applications. River ice plays a key role
in the functioning of aquatic and riparian ecosystems (Prowse, 2001),  and contributes to the
erosion of channels and banks, and the transport of sediments (Ettema, 2002; Beltaos et al.,
2018). Ice alters energy and water exchanges with the atmosphere (Kourzeneva, 2014) and
responds to regional climate variability, thus acting as a good indicator of hydro-climate changes
(Prowse et al., 2011a). River ice affects streamflow via withdrawal (immobilization) of part of
the water during freeze-up and via consequent release during break-up. Ice jams can cause
catastrophic flooding. Field measurements and satellite estimates of river discharge during the



ice/water transition (and vice versa) are not a trivial task (Morse and Hicks, 2005; Zakharova et
al., 2019). As a result, streamflow measurements during these periods are characterized by
higher uncertainties. River ice affects the operation of hydropower stations as well as
construction and navigation activities. In arctic regions, frozen rivers provide a unique
transportation infrastructure for the movement of merchandise and people via winter ice roads.
The presence of river ice cover also provides local population with access to fishing grounds and
in some cases (e.g. Central Yakutia, Russia) to fresh water.

However, operational monitoring of ice on northern rivers is difficult due to site accessibility.
Moreover, ice conditions can be unsafe for people who make *in situ* measurements, especially at
the beginning and end of ice seasons. Satellite remote sensing observations offer an excellent
alternative or complement to field measurements, allowing for the spatio-temporal
characterization of the river ice at frequencies suitable to address various climatic, scientific and
operational requirements.

Satellite-borne instruments provide observational capabilities of many river ice parameters.
Optical sensors such as the Moderate Resolution Imaging Spectroradiometer (MODIS) and the
Advanced Very High Resolution Radiometer (AVHRR) have been used to map river ice extent
and phenology - freeze-up and breakup dates (Pavelsky and Smith, 2004; Chaouch et al., 2014;
Chu and Lindenschmidt, 2016; Muhammad et al., 2016; Cooley and Pavelsky, 2016; Beaton et
al., 2019). However, the  presence of extensive cloud cover for many months of the year and low
solar illumination conditions, particularly during freeze-up period, are limiting factors for ice
monitoring on northern rivers. Active sensors operating in the microwave region are weather
independent and provide higher spatial resolution. Synthetic aperture radar (SAR) data have been
largely used for monitoring river ice phenology (Unterschultz et al., 2009; Mermoz et al., 2009),
deformation (Unterschultz et al., 2009), and the classification of ice types (Chu and
Lindenschmidt, 2016). Ice thickness is another parameter which is of particular interest for
operational purposes (public safety,  ice road service, jam forecast and mitigation). The
capability of passive microwave and thermal satellite instruments for the retrieval of ice
thickness has been demonstrated for large lakes (Kang et al., 2014; Duguay et al., 2015;
Kheyrollah Pour et al., 2017). The spatial dimension of rivers, notably the width of channels,
limits the application of these instruments due to the coarse spatial resolution they provide (km
to tens on km). Some studies have shown the potential of high-resolution active microwave SAR
instruments for retrieving river ice thickness through the establishment of statistical relations
between the backscatter coefficient ($\sigma^0$) and thickness (Unterschultz et al., 2009; Mermoz et al.,
2014). Radar altimeters are another class of active microwave sensors that are largely used for
observation of the water state and regime in oceans and inland waterbodies. The primary goal of
altimetric radars is water (ice) height measurements over the oceans. However, altimeters are
now widely used for monitoring of inland water starting with waterbodies of 100 m in width
(Michailovsky et al., 2012).

Radar signals incident upon the earth's surface are modified according to the physical properties
of materials. Similar to SAR systems, the signal recorded by radar altimeters can be interpreted
as a function of changes in material properties, and the backscatter coefficient (ratio between
power of reflected to received signal) can be used to characterize surface state within the radar





footprint. Radar backscatter over freshwater ice depends on radar configuration (viewing angle,
frequency band) and material properties such as snow/ice liquid water content, surface
roughness, dielectric contrast between snow/ice/underlying water layers, physical properties of
ice (thickness, layering, air bubble inclusions) and snow on ice (depth, density, grain size)
(Ulaby et al., 1986; Duguay et al., 2002; Leconte et al., 2009; Atwood et al., 2015; Gunn et al.,
2015a,b; Antonova et al., 2016; Gunn et al., 2018). Satellite-based SAR instruments used for
freshwater ice studies operate at X, C and L-band frequencies. Theoretical and experimental
studies using higher frequency Ku-band ground and airborne radars have been conducted during
last decade in the context of preparation of the European Space Agency's Earth Explorer
CoReH2O candidate satellite mission (not selected for launch in the end) (Rott et al., 2010; King
et al., 2013; King et al., 2015; Gunn et al., 2015a). Studies by Gunn et al. (2015a,b; 2018) have
showed good sensitivity of the Ku-band to changing freshwater ice properties. Many altimetric
instruments are dual-frequency (e.g. Envisat: Ku/S-band; Jason series and Sentinel-3: Ku/C-band
band) radars. Higher frequency Ku-band measurements are especially suitable for rivers due to
narrower ground radar footprint. Moreover, Ku-band penetration depth into dry freshwater ice is
in the order of 5 to 12 m depending on temperature and properties of the material (Legrésy and
Rémy, 1998; Gunn et al., 2015b; Beckers et al., 2017) and, therefore, sufficient for lake and river
ice applications.

Active (radar) and passive microwave (radiometric) measurements from altimeter missions have
already been used routinely for the determination of ice and open water during ice onset/break-
up periods on large Eurasian lakes (Kouraev et al., 2007, 2015). Compared to radiometric
measurements having footprint diameter of 10-20 km in Ku-band , over calm inland waters
altimetric signals (in the same band range) come from a narrower footprint of 1-3 km (Kouraev
et al., 2004; Jacob et al. 2010; Legrésy & Rémy, 1997). As a result, altimetric observations
acquired over small inland water bodies are less contaminated by the surrounding land. Knowing
that freezing and melting on land starts earlier than on rivers, the radar observations are less
biased by snow-on-land and are more appropriate for observation of river ice phenology than
radiometric measurements. In our previous studies dedicated to the altimetry-based water
discharge estimation of the Arctic rivers (Kouraev et al., 2005; Zakharova et al., 2019, 2020) we
noted that the returned altimetric signal (expressed as backscatter) has a specific seasonal
behavior. This behavior was found to be strongly related to the hydrological phase and it helped
us separate altimetric measurements for ice and ice-free conditions. This procedure made it
possible to improve the accuracy of discharge estimation during winter.

The altimetric radar return signal (waveform) is a combination of backscattering from surface
(surface echo) and subsurface layering (volume echo). The shape of the waveform has been
largely exploited for studies of the properties of ice sheets (Legrésy and Rémy, 1998; Lacroix et
al., 2007; Slater et al., 2019), sea ice (Ricker et al., 2014), snow on land (Papa et al., 2001) and,
more recently, lake ice (Beckers et al., 2017). Over terrestrial and ice surfaces the rising front of
the waveform (leading edge width) is related to local topography, surface roughness and
penetration depth (Legrésy and Rémy, 1997). The falling limb (trailing edge) is a result of the
same characteristics as well as of the extinction properties of the medium. Mercier et al. (2014)
and Beckers et al. (2017) have proposes to use the shape of the leading edge to estimate lake ice
thickness via retracking the heights corresponding to two different peaks on the leading edge.
They found an intermediate peak within the leading edge, which they interpreted as scattering



from the air/ice or air/snow interface (ice surface), while the main peak is considered to come
from the ice/water interface (ice bottom). This conclusion is based on studies dedicated to
investigation of the scattering properties of the freshwater lake ice (Atwood et al., 2015; Gunn et
al., 2015a,b). As shown later in this paper, on many waveforms from river ice, we also detect this
intermediate peak on the leading edge (see section 6.2 for details). However, considering that the
radar echo over rivers comes from very heterogeneous surfaces with variable proportion of land
and water, we avoid attributing this peak to any definitive reflecting boundary. Nonetheless, we
observe a distinct evolution in the main peak with the gradual decrease in its power during the
freeze-up period. In contrast to this peak, other parts of the waveforms do not vary significantly
with time, meaning that the changes in the value of the backscatter coefficient observed during
winter are mainly due to the changing magnitude of this peak. Considering that the change of the
main peak power is due to the radar signal absorption within the snow and the ice, we
hypothesize that a statistical relation can be established between the total value of backscatter
and river ice thickness.

This paper presents the development of algorithms for the retrieval of river ice phenology and
thickness based on altimeter measurements from the Jason-2 and -3 satellite radar altimeter
missions and describes the potential of such missions for climate-related and operational
purposes. The study region is first described (section 2), followed by the primary and secondary
data used in the study (section 3). The time of ice onset and break-up is an important factor
governing ice growth. Consequently, an algorithm of detection of the freezing and melting dates
is proposed in addition to the algorithm for ice thickness retrieval (section 4). The algorithms are
validated against *in situ* observations from four gauging stations (section 5). Using the suggested
algorithms, ice thickness was then retrieved for 48 virtual stations (satellite-river cross-overs)
located within a 400-km long lower reaches of the Ob River (Russia). A weekly product of ice
thickness allowing for extraction at any location of this reach was created through spatio-
temporal interpolation between the virtual stations (section 6.1). Finally, factors affecting radar
measurements over frozen rivers (section 6.2) and the capability of satellite altimeters for
monitoring of river ice parameters with societal benefits are discussed (section 6.3).

## 2  Regional setup

The Ob River is the third largest river of the Arctic Ocean watershed with an annual flow of 406
$km^3$. The river drains the Western Siberian Plain. The lower reach of the Ob River extends
approximately 800 km and begins its confluence with the Irtysh River at 61.08°N. This reach is
characterized by a particular wide floodplain (up to 50 km) with numerous branches. The
easternmost channel is the main, largest, branch called the Big Ob. The second largest branch
delineates the flood plain from the west (Figure 1). The Ob River watershed drains one of the
largest peat bog system in the world and many settlements, located on high terraces of the two
main branches, have limited inter-connection and access to supplies. The main branches are
navigable; however they are covered by ice for seven months of the year. In winter, when the
bogs freeze, the local communities intensify their socio-economic activities by constructing
winter roads and ice bridges over river crossings. River ice observations are sparse and are taken
at a few gauging stations dedicated to water level monitoring. For this study, we selected a
section of the lower reaches of the Ob River, which is located between two administrative
centers of the region (Salekhard and Khanty-Mansyisk). This river section is covered by a





sufficiently dense network of Jason-2 and 3 satellite tracks and represents a good case study area for the development and validation of the proposed ice thickness retrieval method.

In the lower reaches of the Ob River, ice formation begins on average 23-27 October. The earliest and latest records for the last 20 years are 1 October and 18 November, respectively. Ice cover forms quickly on this section of the river, typically within just 2-3 days. However, in 15 % of the cases, ice formation can last up to 10 days. Ice grows rapidly during the first month of the ice season and reaches 0.23-0.30 m in thickness by the end of November. This corresponds to the

time when ice has reached 30% of its maximum annual thickness. By the end of the ice growth period (March-April), ice thickness reaches 0.80-1.0 m on average (1.50 m maximum value). Snow depth on the ice surface varies from 0.09-0.13 m in November to 0.30-0.50 m in April.

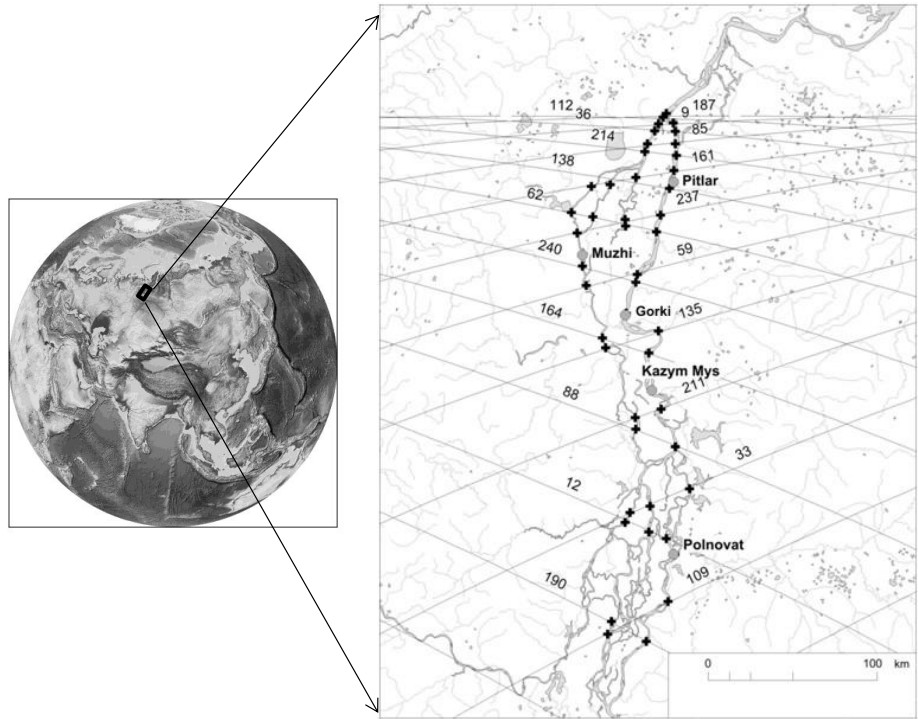

**Figure 1: The lower reaches of the Ob River and location of the virtual (crosses) and**
**gauging (circles) stations. The virtual stations correspond to satellite-river cross-overs. Jason-2 and 3 satellite tracks and corresponding track numbers are also shown. The global map is created using free The Matplotlib Basemap Toolkit. The main map is produced using public The World Bank data (https://datacatalog.worldbank.org/dataset/major-rivers-world).**


The temporal dynamics of ice growth on the large linear channel sections is similar along the studied reaches. However, in the south ice thickness is 0.07-0.20 m less than in the north of the region (Figure 2a). Climate change affecting river ice in the Canadian Arctic (Prowse et al., 2011b) and the European part of Russia (Agafonova and Vasilenko, 2020) has not yet resulted in





a significant change of the ice regime in the lower Ob River. The long-term trends in ice onset
and melt, as well as in maximum ice thickness observed on two gauging stations in the middle
and the north of our region of study, are not significant. Nevertheless, during the last ca. 10-15
years some trends are noticeable in later freezing, earlier melting and in the thinning of the ice
cover (Figure 2b).


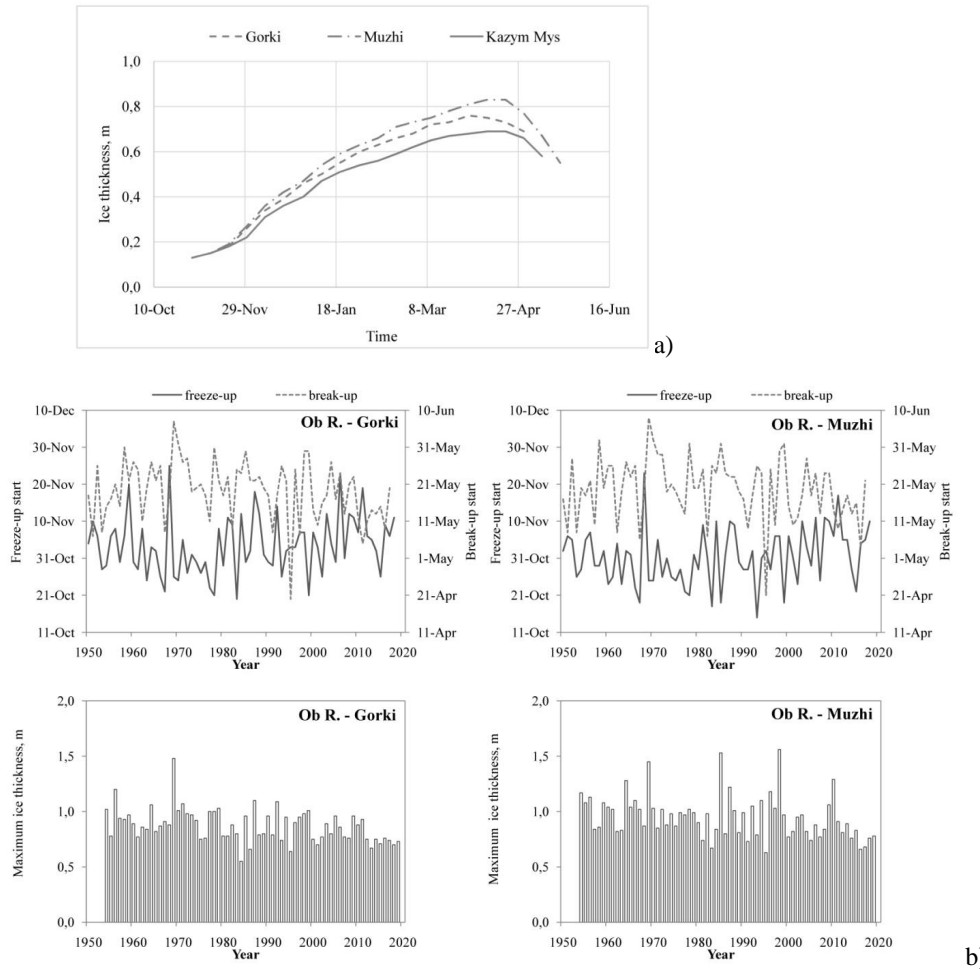

**Figure 2: Ice phenology and thickness climatology (1980-2017) at three stations (Gorki,
Muzhi and Kazym-Mys) along the studied river reaches: (a) temporal dynamic of ice setup
and break-up, and; b) maximum ice thickness.**

### 3  Data

### 3.1. Altimetry





The Jason-2 satellite is the third altimetric satellite of the Topex/Poseidon-Jason series. The
satellite operated during 2008-2016 and acquired data in a 10-day repeat orbit with an inclination
of 66.08°. The altimetric radar aboard Jason-2 provided measurements at Ku (13.6 GHz) and C
(5.3 GHz) bands. The theoretical footprint of the radar at Ku-band is 10-12 km in diameter over
the rough ocean surface. This diameter decreases over smooth surfaces such that the main return
signal can come from footprints of just a few kilometers in diameter (Legresy et al., 1998).

The satellite payload of Jason-2 also included a bi-frequency nadir-looking Advanced
Microwave Radiometer (AMR), providing measurements of brightness temperature at 18.7, 23.8,
34.0 GHz frequencies. Brightness temperature measurements acquired with other passive
microwave radiometers, such as AMSR-E, have been used successfully for the retrieval of ice
thickness on Great Slave Lake and Great Bear Lake, Canada (Kang et al., 2014). Since the Jason
AMR footprints are large, correspondingly to 42 km (18.7 GHz), 35 km (23.8 GHz) and 22 km
(34.0 GHz) in diameter (Kouraev et al., 2007), the radiometric measurements over rivers are
dominated by signals emitted mainly from land surfaces surrounding the river channels. In this
study, we used Jason-2 and 3 AMR measurements only as an additional check of the beginning
of freezing along the riverbanks and terraces, and for adjustment of the altimetric freezing
algorithm.

In 2016, the successor Jason-3 satellite mission was put into space with the same orbit as Jason-
2. For 20 cycles the two missions flew with an 80-second time lag ensuring continuity of
measurements. During this period, the difference (bias) between Jason-2 and Jason-3 for Ku-
band backscatter was within 1 dB. The difference between 34.0 GHz brightness temperature
measurements was within 3 K.

In this study, the satellite measurements were extracted from the geophysical research data
records (GDR) distributed by AVISO+ data portal (avisoftp.cnes.fr) with help of high-resolution
optical Landsat 8 images.

### 3.2. Optical imagery

Landsat-8 and Sentinel-2 georeferenced RGB colour composite images were downloaded from
the USGS data portal (https://earthexplorer.usgs.gov/). The images were used for 1) precise
selection of the Jason measurements over the river channels at corss-overs and 2) demonstration
of the spatial heterogeneity of the ice phenology between satellite-river cross-overs (virtual
stations or VS).

The ice season corresponds to the low-flow period, when the river width is minimal. Considering
this, for the first task, images acquired on 2 August 2013 (end of the flood recession) and 18
October 2013 (beginning of the winter low-flow) were used. This helped to minimize the impact
of land contamination when selecting the altimetric measurements.

### 3.3. In situ data

The Russian Hydrometeorological Service monitors ice at all gauging stations providing water
level measurements. In the lower Ob reach covered by Jason observations, there are five water
level gauging stations (Figure 1, Table 1). Four stations (Polnovat, Gorki, Kazym-Mys and





Pitlar) are located on the main branch of the Ob River and one station (Muzhi) provides
observations on the secondary channel called the Small Ob River.

**Table 1: Gauging stations located in the lower Ob reach.**

| River- station | Distance from mouth (km) | Beginning of observations | Observation gaps |
|---|---|---|---|
| Ob – Polnovat | 702 | 1970 | |
| Ob – Gorki | 487 | 1935 | |
| Small Ob – Muzhi | 463 | 1933 | |
| Ob – Kazym Mys | 551 | 1979 | 1988 –2003 |
| Ob – Pitlar | 386 | 1979 | 1990 – 2005 |

The standard protocol of river ice monitoring includes daily visual observations of ice
presence/absence and ice type; measurements of the ice thickness and on-ice snow depth (3-6
times per month). Ice thickness is measured by drilling one hole in the ice using ice augers.
Snow depth corresponds to the average value calculated from three snow depth measurements
located around the hole. As the dates of *in situ* measurements do not coincide with the Jason
measurements at virtual stations, ice thickness values were linearly interpolated between two
adjacent *in situ* observations for the dates of satellite overpasses.

## 4  Methods

### 4.1. Ice onset and break-up from altimetric measurements

The seasonal variability of the backscatter coefficient follows the seasonal evolution of the state
of the reflecting surface within radar altimeter footprint.  High backscatter values are observed
when the footprint contains a large fraction of surface water and the water is calm. Over large
flooded areas the water surface exhibits a certain roughness due to turbulent flow and wind,
while the presence of floating ice, frazil or slush increases its specularity; behaving similarly as
calm water.  Freezing in river channels starts from the banks (where the turbulence and flow are
small) by the formation of a fine skim ice with a smooth surface. This ice grows in area and
thickness, intercepts and accumulates floating frazil flocks and ice floes. During periods of snow
accumulation, shuga (new ice, composed of spongy, white lumps a few cm across resembling
slushy snowballs) forms and drifts along the river. This contributes to the growth of border ice,
reducing the open water area and leading to the formation of ice dams (bridging). The bridging
starts at tight bends or at narrow channel locations. The drift of frazil/shuga floes is of common
occurrence on many rivers in autumn. At this time of the year, the peak on backscatter time
series indicates the start of freezing (Figure 3). This peak is followed by a progressive winter
decrease, which forms a recession limb on backscatter time series.

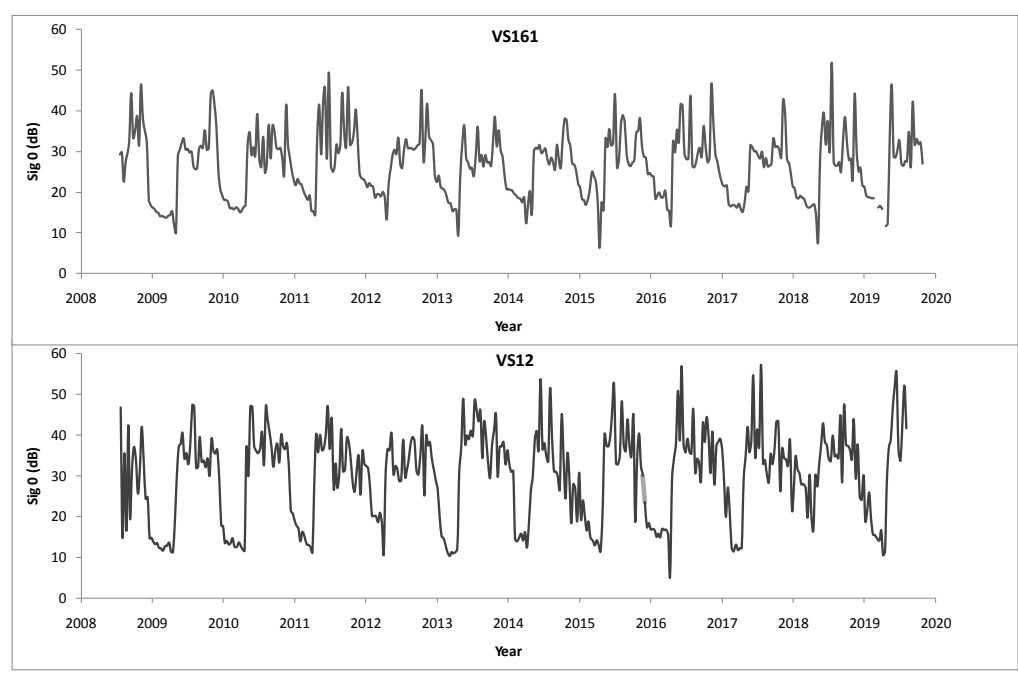

**Figure 3: Variability of backscatter at VS 161 and VS 12 (see Figure 1 for VS locations).**

The ice mainly grows as water freezes on the bottom of the ice cover (called congelation ice) and the latent heat of crystallization is conducted upwards through the ice and snow to the atmosphere. Growth can also occur on top of the ice cover when the snow load or hydrostatic pressure are high and water seeps through cracks wetting the snow. The wet snow refreezes forming porous white ice is called nalyed (Russian) or snow ice. As ice grows and volume scattering of the radar echo increases, the backscatter decreases. On the Ob River the ice gains about 30% of the thickness during the first freezing month. At many virtual stations the highest temporal changes of the backscatter (dSig0/dt) are observed exactly during that period. The situation is complicated if the open water (polynya) persists due to high local velocities or tributary inflow. As the real orbits of the Jason satellites oscillate within 400 m across the nominal mean orbit, the fraction of open water of polynya within the footprint varies, resulting in secondary peaks on the backscatter recession limb. Small winter peaks can also appear due to the strong redistribution of snow of the ice surface, snow wetting during the nalyed formation (in winter) or occasional snow melt during warm sunny days (in spring).

River ice break-up is influenced by both thermodynamic and hydrodynamic processes known as thermal and mechanical break-up, respectively. First, when air temperatures are still mostly negative, ice undergoes metamorphism under the influence of solar radiation. At that time a drop in backscatter in the order of 5-10 dB can be observed. This phenomenon has previously been observed during the ice period on Lake Baikal using SARAL/AltiKa altimeter data (Kouraev et al., 2015) where it was suggested that ice metamorphism (formation of dendroidal air channels



just below the ice surface and early stages of needle ice formation) are partly responsible for the decrease in backscatter. When air temperatures become positive, the snow on the ice surface
melts and the backscatter starts to increase. The melting progressively affects the ice and vast melt ponds can appear on the ice surface leading to an increase in backscatter.

The mechanical break-up starts when the water level rises. Water can flood the ice surface due to earlier flood on the tributaries or due to cracks through the weakened/fractured ice sheet. The first high (>25 dB) backscatter peak occurs at the beginning of the flood. The value of the peak
ranges from 25-50 dB, depending on the stage of breakup and river morphology (channel width, banks, oxbow lakes). The peak is high if the observation is acquired when the floating ice is still present within the radar footprint. However, on the Ob River the spring peak of backscatter rarely corresponds to  maximal  value of a given year.

As the water level rises, the backscatter decreases due to increasing waves on the surface water
that result in higher surface roughness. During the open water season in summer several peaks are frequently observable. Summer variability in backscatter depends on many factors including, but not limited to, virtual station location (banks, presence of islands, floodplain characteristics), part of water within footprint (intermittent summer rain floods inundation), and wind influence.

**Algorithm**

Considering the described behavior of the backscatter, we suggest that the last annual peak in the backscatter corresponds to the beginning of the river ice formation. In the case of a multi-peaky recession limb, this peak should be of order of spring and summer peaks. If the selection of peak is not straightforward (for example two high peaks within one month or prominence of peak is low), an additional criterion based on the brightness temperature difference (dTb) between 34.0
and 18.7 GHz frequencies is introduced. We select the first backscatter peak at time $t$ when in a window ($t$-1, $t$+2) days at least three dTb values are <2 K. In the GDR, the radiometric measurements are provided with the 1 Hz frequency and are interpolated at each 20 Hz radar measurement. This means that the Tb measurements integrate emissions about surface state from a larger surrounding area than altimetric radar backscatter measurements. Freezing on the
floodplain and banks can occur earlier or later, depending on eventual oxbow lake size/depth or from antecedent snow events. By applying the ($t$-1, $t$+2) window, we ensure that the freezing is progressing and the backscatter peak is not caused by a synoptic-scale cooling episode.

The beginning of the ice cover decay (thermal melting) marks the beginning of spring backscatter increase. The melt detection algorithm searches for the spring peak in the backscatter
time series. For the multi-peaky winter, the algorithm uses the dTB condition. In this case the algorithm searches for the peak, which is accompanied by a simultaneous increase in dTB in the order of values that are typical for an average summer dTB value for a given VS in a given year. In a few instances, the spring peak is absent or cannot be automatically detected because of a low prominence. In such case we use the date of maximal increase in backscatter (dSig0/d$t$) for a
period from January to mid-June.

A variety of combinations of different geomorphological (banks, floodplain, river width, islands), meteorological (synoptic cooling/warming episodes), and ice cover (polynya, nalyed, ridging) conditions can exist. Their complex impact on the backscatter variability during river ice freezing and melting make it difficult to address all variations in an automated manner. In

this context, we decided to compare the performance of the described automatic freeze/melt detection algorithm with its manual implementation (visual analysis of time series). Both results are compared to the ice flags (ice types or ice cover state) provided by the nearest gauging stations.

## 4.2. Altimetric ice thickness

Mercier et al. (2014) and Beckers et al. ( 2017) have previously exploited the radar return echo to estimate lake ice thickness. They found the intermediate peak on the leading edge of the radar echo, which is interpreted as the scattering from the air/ice or air/snow interface (ice surface), while the main peak is considered to come from the ice/water interface (ice bottom). This
conclusion is based on studies dedicated to investigations of the scattering properties of the fresh lake ice (Gunn et al., 2015b; Atwood et al., 2015). On many radar waveforms extracted from river ice we also detect this intermediate peak on the leading edge (Figure 4). As we noted above, considering that radar echoes over rivers come from very heterogeneous surfaces with variable proportions of land and water, we avoid to refer at this peak to any definitive reflecting
boundary. Instead, we propose to explore a statistical relation between the total value of backscatter provided by GDR product (guided over river ice by the main waveform peak) and the river ice thickness as they express very similar seasonal variability (Figure 5a).

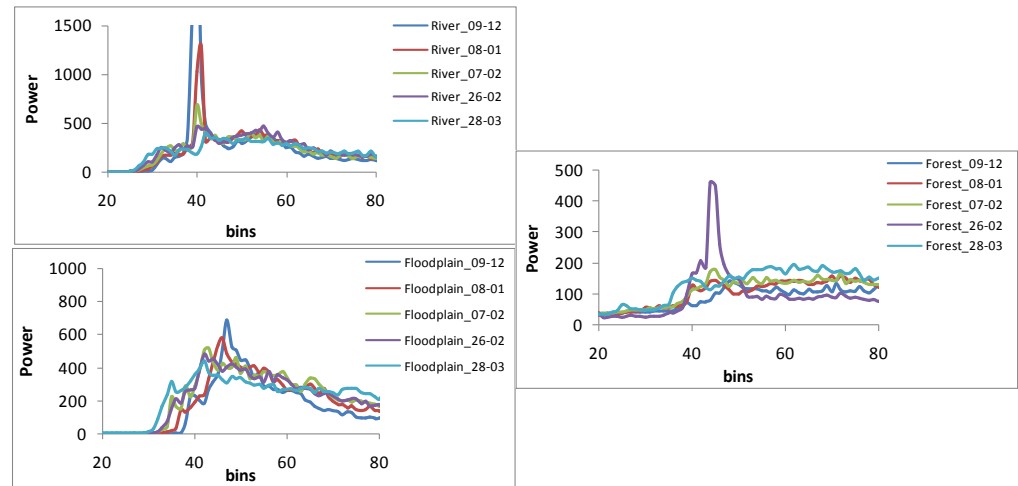


**Figure 4: Winter evolution of typical waveforms for track 88 (see Figure 1) over a river channel and other surrounding surface types. The coloured lines correspond to different dates of the winter 2013-2014.**

Year-to-year variations in backscatter at the beginning of the freeze-up period may be caused by different land/water/ice proportions within the radar altimeter footprint, wind conditions, floating ice concentration, etc. Assuming that the decrease in backscatter between two consecutive observations (dSig0/d$t$) is proportional to a gain in ice thickness, we use a relative backscatter decrease instead of the backscatter absolute values, thus, avoiding an impact of initial freezing


conditions. Starting from the first date of freezing, we estimate the backscatter cumulative
difference CumSum(dSig0/d$t$) and relate this parameter to the *in situ* ice thickness (Hice)
measured at nearest gauging station. As the satellite orbit oscillates within a 400-m band width,
the variable proportion of the land/water/ice within a footprint often produces the secondary
peaks on the backscatter winter curve. The application of a Loess filter to the CumSum(dSig0/d$t$)
parameter makes it possible to minimize this effect.

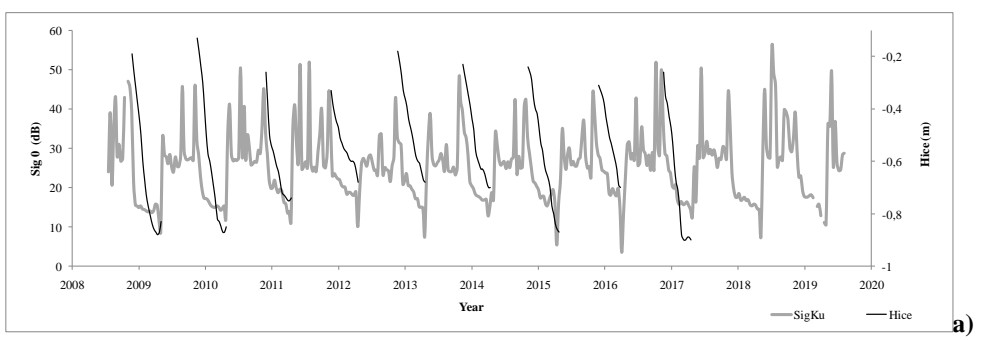

a)

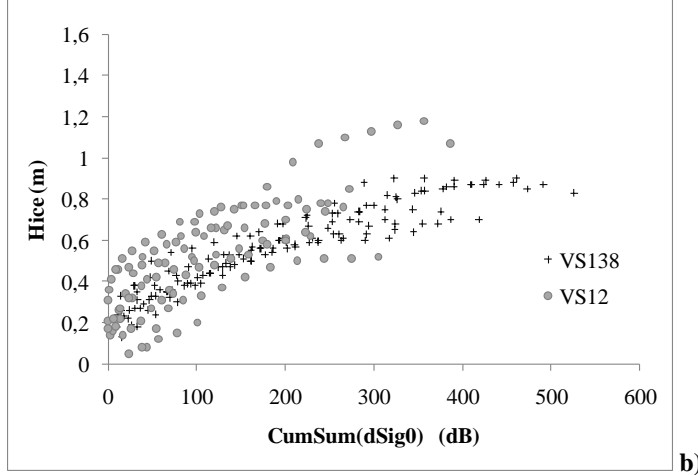

b)

**Figure 5: Seasonal variability of ice thickness at Pitlar gauging station and of the**
**backscatter coefficient at nearest 138 virtual stations (a) and relation between cumulative**
**dSig0/dt and ice thickness at two virtual stations(b).**

Along the 400 km-long Low Ob River reach covered by the 20 northernmost Jason satellite
tracks, 48 virtual stations were created for the main and secondary branches. Eight virtual
stations nearest to the location of the gauging stations were chosen as a training set for
calibration and validation of the ice thickness relations as well as for evaluation of the freeze-
up/break-up dates algorithm. The established relations were extrapolated to other "main set" of
virtual stations using two approaches: 1) by nearest distance to one of the eight VS from training
list; and 2) by best correlation between backscatter time series (see Section 6.1 and Figure 6) .





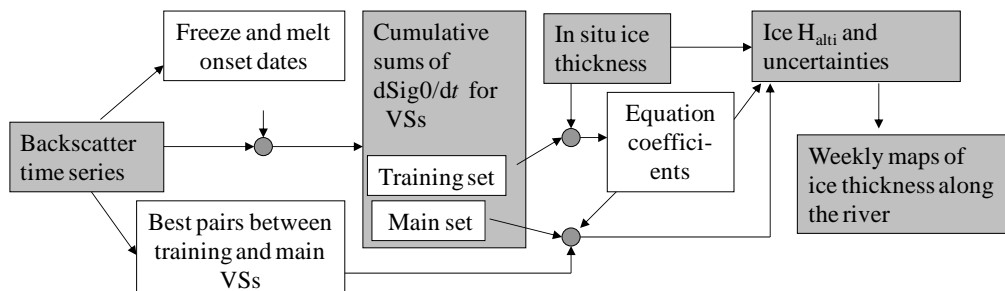

**Figure 6: Processing scheme of ice thickness retrievals from altimetric radar measurements.**

## 5 Results

Altimetry-derived dates of ice onset and break-up were verified against data from virtual stations located close to gauges. These dates were then used as ice start/end dates for the estimation of river ice thickness with specific equation coefficients. Following validation, the algorithm was applied to other stations of the studied reach.

### 5.1. Ice phenology

Freeze-up on rivers starts with the formation of frazil ice in turbulent fast flowing water followed by frazil consolidation into pans and floes. Along the banks where the water velocity is low, border ice forms and grows progressively. Floating and border ice reduce surface turbulence and wind action effects resulting in a decrease in surface roughness. This moment is well detected by Jason radar altimeter. Taking into account the 10-day repeat overpass of the satellite observations and the distance between gauging and virtual stations, we consider a 10-day time-step bias as an acceptable accuracy for Jason altimeters. For 90% of the retrievals based on manual procedure, the difference against observations of the first ice events at the gauge stations is less than 10 days (Figure 7,a). In 56% of the cases this difference is close to zero. As the radar footprint over rivers is heterogeneous and is affected by signals from the frozen/unfrozen state of land/river/floodplain lakes, there are numerous variations in the behavior of backscatter at the beginning of the freeze-up period. At this time, the automated routine misses certain behavior types and detection is less accurate for the first ice appearance. Only 70% of the altimetric freeze-up dates fall within 10 days of *in situ* observations at gauges and only 40% have biases close to zero. This results in earlier detection of ice onset by 20 days using the automated algorithm

Break-up is a more complex process and consists of two phases: thermal degradation of the ice cover and its mechanical break up and downstream movement. Comparing the dates of altimetry-derived melt onset with the ice state flags provided by gauging stations, we conclude that our approach detects better the start of ice thermal degradation. In 88% of the cases, the difference of manually-retrieved dates against *in situ* observations of water appearance on the ice cover is ±10 days (Figure 7, b). The automated algorithm performance is least for the detection of the start of melt (only 54% of the cases). It is more efficient for the detection of the end of melt (Figure 7 b,c), e.g. first date of open water appearance (67% of cases).





Manual retrieval of dates associated with freeze-up/break-up allows for a better control on the complex behavior in backscatter and, consequently, handling of the outliers. The automatic algorithm, naturally, passes through these cases and detects the outliers. For 48 virtual stations on both the main and secondary channels, for the full 12-year period of study, the automated algorithm fails (e.g. detects melting/freezing dates before 10 April and after 10 June) in less than 10% of the cases. Cases when the algorithm fails due to the long gaps between altimetric

measurements are also included.

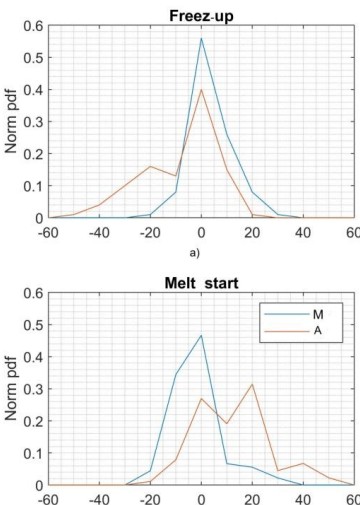

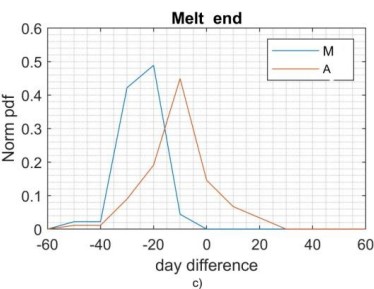

**Figure 7: Normalized distribution of bias between altimetric and *in situ* observed dates of freeze-up (a), melt start (b) and melt end (c) for 2008-2019 for 11 virtual stations from training set of VSs. M - manual algorithm, A - automated algorithm.**

The proposed algorithm shows a good sensitivity for monitoring interannual variability of ice events on the Ob River (Figure 8). The altimeter detects well earlier freeze-up in 2013-2015 as observed at gauging stations. Results from the automated algorithm are more noisy. Nevertheless, a clear coherence exists between the corresponding time series (Figure 8b). The earlier melt start and end in 2011 and 2016, and later melt start in 2015 are noticeable in both *in situ* and altimetric data (Figure 9a). Significant variability (order of 20 days) in melt dates is

observed in 2014 between gauging and virtual stations.

The average freezing dates calculated from *in situ* observations display an important spatial gradient, especially when adding the Salekhard gauging station located 65 km northward of the study reach (Figure 10a). The average calculated from altimetry does not show this gradient. Nevertheless, the time lag in freeze-up dates between the southern and northern reaches in the

order of 10 days can be observed in the half of the years (not shown), while in other years local site-specific factors dominate over the main regional climate drivers, hiding this lag.

I*n situ* observations reveal a clear latitudinal gradient in melt start and end dates. A gradient in the order of 20 days is observed from altimetric data for melt start (Figure 10b). For the melt end



dates, a lower gradient in the order of 10 days is recorded from both *in situ* and satellite data
(Figure 10c).

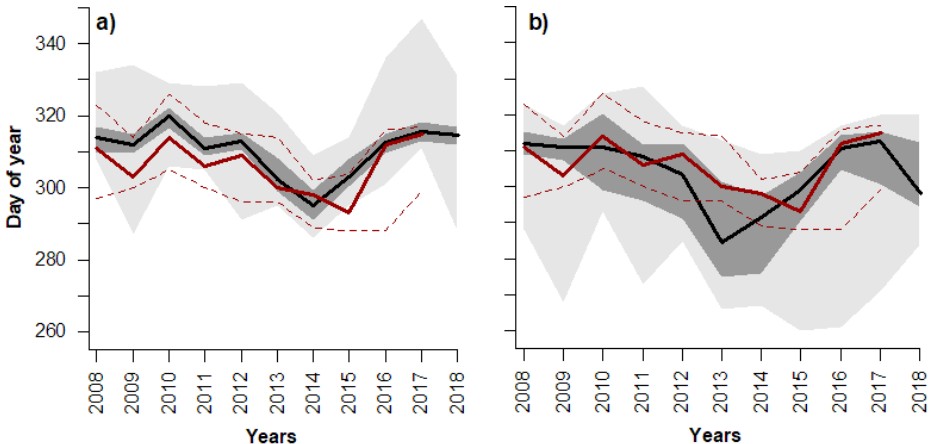

**Figure 8: Interannual variability of altimetry-derived dates of freeze-up start from manual
(a) and automated (b) approaches for the main Ob River channel. Red lines are the mean
(bold) and the min-max values (dashed) observed on the gauging stations along the Big Ob**
**River. The black line corresponds to the median value of dates observed at 20 virtual
stations. The dark grey zone is the spread between 3$^{rd}$ and 1$^{st}$ quartile, and light grey zone
is the spread between minimum and maximum values.**

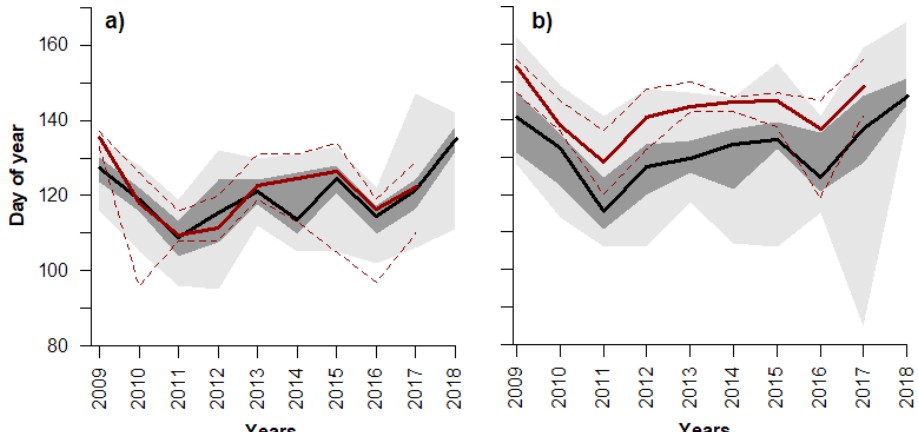

**Figure 9: Interannual variability of the altimetry-derived dates of melt start from manual**
**(a) and melt end from automated (b) approaches for the main Big Ob River channel. Red
lines are the mean (bold) and the min-max values (dashed) observed on the gauging
stations along the Big Ob River. The black line corresponds to the median value of dates
observed at 20 virtual stations. The dark grey zone is the spread between 3$^{rd}$ and 1$^{st}$
quartile, and light grey zone is the spread between minimum and maximum values.**


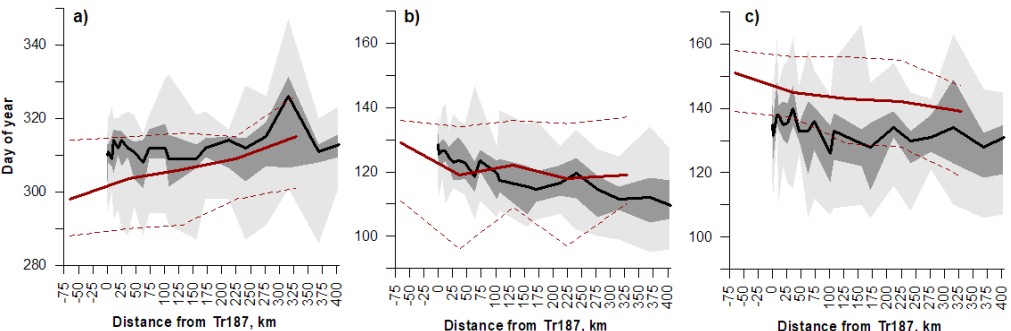

**Figure 10: Spatial dynamic of freeze-up (a), melt start (b) and melt end (c) dates from altimetric and *in situ* (bold line) observations (along the main Big Ob River channel from north to south) for different years from manual (a and b) and automated (c) approaches.**
**Red lines are the mean (bold) and the min-max values (dashed) observed on the gauging stations along the Big Ob River. The black line corresponds to the median value of dates observed at 20 virtual stations. The dark grey zone is the spread between 3$^{rd}$ and 1$^{st}$ quartile, and light grey zone is the spread between minimum and maximum values.**

The difference between freeze start from virtual stations located at a similar latitude on the main (Big Ob) and the secondary smaller channels can be significant and reach up to 30 days (Figure 11a). Variability in ice onset dates is higher for smaller channels and likely attributable to the effect of local hydraulics and geomorphology. For example, the systematic difference on the northernmost tail of the studied reach (0-50 km from VS187) reveals that freezing at these latitudes starts earlier on small branches. Narrower channels, multiple big islands and sandbars on secondary channels are particularly favourable for the earlier ice consolidation in numerous shallower parts and in head of meanders. This is shown clearly on a Sentinel-2 optical image acquired on November 4, 2016 during ice consolidation (Figure 12a). At another location (see Figure 11a at 310 km ), the altimeter observations show that the main branch freezes up to 30 days later than the smaller brunches. According to high resolution Landsat-8 image (not shown) an area of open water (polynya), formed due to the complex fluvial morphology (island, close bifurcation node, meander), can persist at this location until March.

The dates of altimetry-derived melt start are consistent between the branches (Figure 11b). Break-up on the Ob River begins from thermal ice degradation, which follows the propagation of warm air from the southwest of the West Siberian Plain. At the beginning of ice degradation local morphological controls only play a small role (Figure 11b). Their role amplifies during mechanical break-up, which is better captured by our automated algorithm (Figure 11c). While some uncertainty can be attributed to the algorithm, we suggest that the higher variability in melt dates derived from altimetry between adjacent virtual stations or between branches could be explained by local environmental conditions resulting in irregular spatial ice break-up or jamming (Figure 12b).

## 5.2. Ice Thickness





A power equation (1) produced the best fit between cumulative backscatter difference and *in situ* ice thickness measurements:

Hice_alti= a×CumSum(dSig0)$^{b}$                                                   (1)

Parameters a and b of the equations were estimated at the different gauging stations as the average from 9 leave-one-year-out runs. The average correlation coefficients and root mean square errors (RMSE) calculated by this method are presented in Table 2.

At the northern virtual stations, the relation between backscatter and ice thickness is stronger and
the errors in estimates of the altimeter-retrieved ice thickness are less than 0.12 m (Figure 13). For southernmost gauging station of Pitlar the errors increase up to 0.14-0.18 m. The uncertainty in ice thickness estimates is higher at the beginning and at the end of the ice period. Inaccurate detection of ice onset can affect the accuracy of thickness estimates, especially at the beginning of freeze-up. Another reason for the high uncertainty is the multi-peaky character of the
backscatter winter recession curve and the residual noise present in the backscatter time series after application of the smoothing procedure. This occurs, for example, at VS 12 where a polynya persisted until March in at least four years of the study period, producing noisy backscatter in wintertime series.

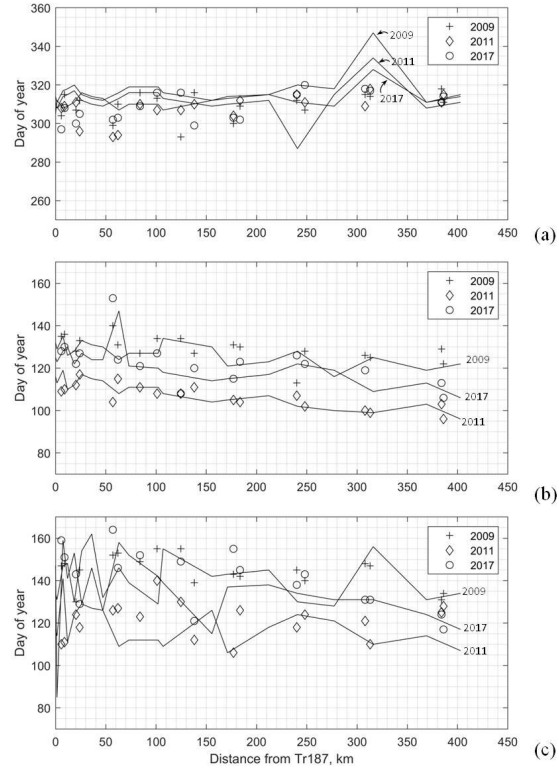

**Figure 11: Spatial variability of freeze-up (a), melt start (b) and melt end (c) dates from altimetric observations for main channel (lines) and for small channel (markers) for three years.**



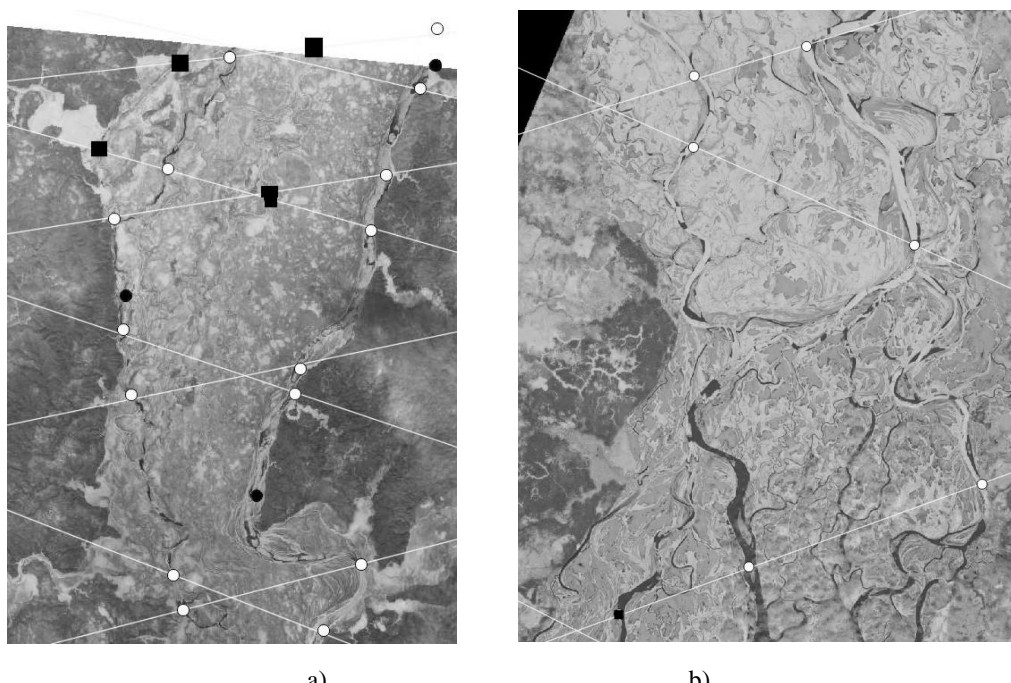

a)                                                    b)

**Figure 12: Optical satellite images illustrating the spatial variability of freeze-up (left) and melt (right) processes provided by USGS EarthExplorer portal (https://earthexplorer.usgs.go). a) Sentinel-2 image acquired on November 4, 2016 covering the 50-200 km reach and virtual (open circles and squares) and *in situ* (black circles) stations (VS on small secondary branches are represented by squares). Large and deep channels have large open water areas, while the numerous small channels are already completely frozen. b) Landsat-8 image acquired on May 15, 2017 covering the 200-300 km reach and virtual stations. Small and large secondary branches on the south of the image are already ice-free, while the main channel is still covered by the ice. The thermal melt, seen as open water along the banks, affects all branches on the north of the image.**




Except for VS 109, the variability in the values of parameters a and b in Equation (1) is low which indicates a good stability in the relations and their potential validity for other virtual stations located far from the gauged reaches. One of the way of verifying the sensitivity of the Hice_alti retrievals to fitting parameters consists in the application of the equation developed for an adjacent virtual station. The results obtained demonstrate a robust of fitting with high correlation with *in situ* Hice and low errors of Hice_alti for the northern virtual stations. The results are not as good for the southernmost virtual stations. However, the retrievals at the southern VS could be improved by selecting the equation parameters (a and b) from a virtual station located far away from the corresponding gauging station, but expressing better correlation between backscatter time series. For example, when applying the equation built for VS135-Gorki gauging station to VS109 and VS12 (backscatter on the VS135 demonstrated the highest






correlation with the backscatter on VS109 and VS12), the RMSE of retrieved Hice_alti for these virtual stations decreases from 0.23 to 0.18 m.

**Table 2. Scores for built relations between cumulative difference of backscatter and *in situ* ice thickness measurements for different gauging and virtual stations from training set.**

| Gauging stations | Virtual stations | a | b | R | RMSE, (m) | VS for cross-validation equation** | R cross-validation** | RMSE cross-validation** (m) |
|---|---|---|---|---|---|---|---|---|
| Pitlar | 161 | 8.69 | 0.39 | 0.94 | 0.07 | 138 | 0.94 | 0.09 |
| | 138 | 6.54 | 0.42 | 0.94 | 0.07 | 161 | 0.94 | 0.09 |
| Muzhi | 237 S_Ob | 7.64 | 0.42 | 0.90 | 0.10 | 240 S_Ob | 0.90 | 0.10 |
| Pitlar | 240 | 9.22 | 0.36 | 0.86 | 0.10 | 135 | 0.86 | 0.10 |
| Gorki | 240 | 7.70 | 0.39 | 0.81 | 0.12 | 135 | 0.81 | 0.13 |
| Muzhi | 240 S_Ob | 7.96 | 0.41 | 0.90 | 0.10 | 237 S_Ob | 0.89 | 0.11 |
| Gorki | 135 | 6.88 | 0.42 | 0.87 | 0.11 | 240 | 0.87 | 0.11 |
| Kazym Mys | 164 | 8.83 | 0.35 | 0.84 | 0.10 | 211 | 0.84 | 0.10 |
| | 211 | 10.7 | 0.31 | 0.76 | 0.12 | 164 | 0.76 | 0.13 |
| Polnovat | 12 | 8.23 | 0.41 | 0.77 | 0.18 | 109/135* | 0.76/0.76 | 0.23/0.18* |
| | 109 | 2.92 | 0.55 | 0.84 | 0.14 | 12/135* | 0.76/0.76 | 0.23/0.19* |

* two equations built for corresponding virtual stations are used for cross-validation.
** for cross-validation explication see Section 6.1.

**6 Discussion**

**6.1. Ice thickness for the entire studied river reach**

Using parameters of the equation developed for the VS from the training set, we estimated ice thickness for all 48 Jason virtual stations located on the main and secondary branches of the Ob River. These retrievals were used for the creation of weekly maps, which were generalized into a
2D spatio-temporal ice thickness product (Figure 14). For this the altimeter-derived ice thicknesses were interpolated and smoothed in the 2D spatio-temporal coordinates using a moving average filter with 40 km/30 days window size.





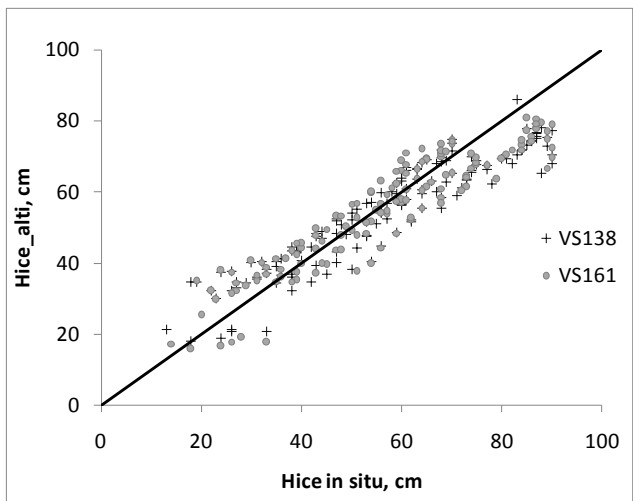

**Figure 13: Retrieved ice thickness from backscatter altimetric measurements at VS 161 and VS 138 against *in situ* measurements at Pitlar gauging station. The black solid line is the 1:1 relation line.**

Two approaches for selection of analog equation for each individual VS were tested. The first
approach consisted in the application of the equation developed for nearest virtual station referred to one of the four gauging stations (Table 1). In the second approach, we searched for the best correlation between backscatter at main and backscatter at training VSs considering potential time shift ± 1 satellite cycle (e.g. ≤10 days ). The performance of the both approaches was evaluated at 11 virtual stations nearest to the location of the gauging stations. The second
approach outperformed the other one, achieving better accuracies with RMSE values varying between 0.09-0.19 m (Table 2).

The along-river variability in ice thickness controlled by local morphological factors can be important. In the absence of validation data for the retrieved ice thickness for inter-station areas, we suggest to examine the interannual dynamics of two parameters derived from altimetric and
in situ observations: the maximum ice thickness and the ice thickness observed on 1 December. From a practical standpoint, knowledge of the maximum river ice thickness is relevant for climate monitoring, while the ice thickness determined on 1 December is crucial for local and regional socio-economic stakeholders as this is an average date for the opening of the ice bridge road to the north of the study area. The interannual variability in maximum ice thickness
retrieved from altimetric measurements at many virtual stations indicates a clear decrease from 2008 to 2012. This decrease corresponds well to the measurements made at all gauging stations. Since 2013, the maximum ice thickness has slowly increased; however altimetric and *in situ* observations both exhibit spatio-temporal variability (Figure 15a) that are not always in agreement. The disagreement may be related to the simplicity of the empirical approach of ice
thickness retrievals based on correlation or to the combination of environmental factors such as winter temperatures, snow amount, autumn ice drift and accumulation, ridging and ice flood (water-on-ice). For example, the ridging flags appear more frequently after 2012 in the records





of the northernmost gauging station Pitlar. The spatio-temporal smoothing of the altimetric retrievals used in the map production can also contribute to the disagreement in areas when the spatial variability prevails over temporal variability.

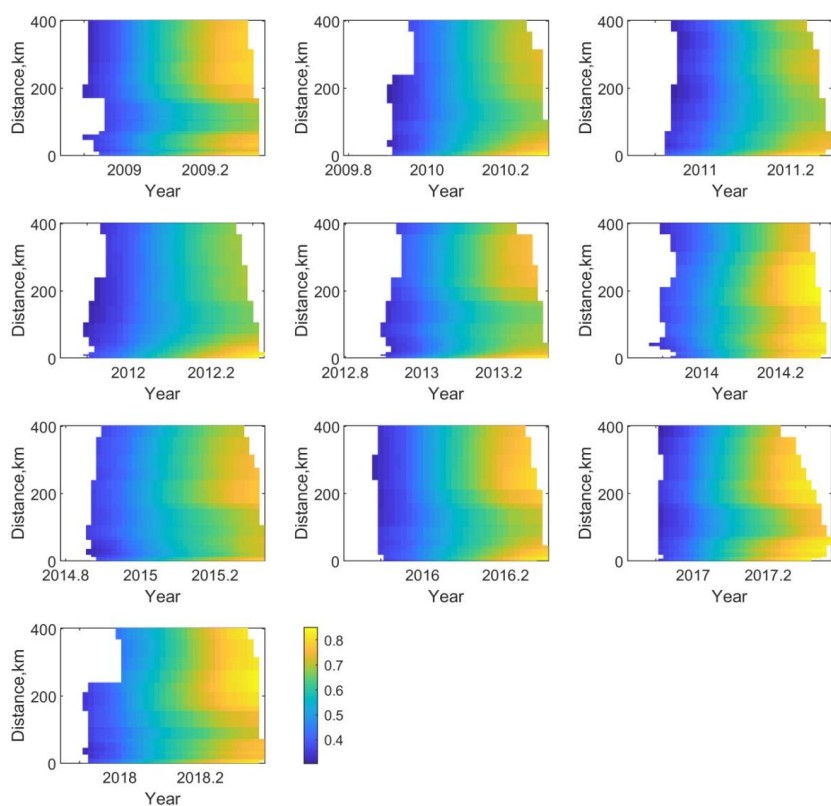

**Figure 14: Spatio-temporal ice thickness variability (m) for main branch of entire Low Ob River reach for the 2008-1017 period from generalized weekly altimetric product. Distance is indicated from the northernmost virtual station 187.**

The interannual variability of altimetric ice thickness on 1 December shows more disagreement with *in situ* observations. This does not strongly contradict expectations as for most virtual stations this disagreement lies within estimated RMSE values (0.07-0.18 m). Besides the reasons noted earlier, the degradation in quality of the *in situ* time series and the lower representativeness of the one-hole sampling protocol can be evoked. The cross-sectional and along-river low-scale heterogeneity of the ice thickness is highest at the beginning of the freeze-up period. This heterogeneity reduces as the ice grows. According to Fedorov et al. (2019), cross- and along-river variability of ice thickness on Siberian rivers can reach 10-20% (or 0.07-0.16 m for typical values on the Ob River) by the end of the ice growth period.

Results obtained demonstrate that the altimetric ice thickness retrievals are accurate enough for use in climate studies. However, for this first version of the product, we cannot recommend its





use for winter road operational purposes as it seems that for many locations we overestimate the ice thickness at the beginning of the freeze-up period which is critical for people's safety. Several improvements are suggested for future work. First, improvement in the accuracy of the detection of the ice onset algorithm. The detection of the date of the first consolidated ice instead

of first ice event (bank ice or frazil floes as in our case) could help to reduce the dispersion of points in the low range of the Hice-CumSum(dSig0) scatter plot. This would produce a better fit of the statistical relations. Another improvement consists in use of other parameters of the altimetric radar waveform instead of backscatter coefficient. The backscatter coefficient correlates well with the amplitude of the main peak. However, the contribution of the trailing

edge or the leading edge cannot be neglected, especially for lower primary peaks observed for thicker ice. Amongst the suggested parameters are the amplitude of the main peak or the area under this peak. Unfortunately, these parameters are not directly provided in the AVISO+ Jason GDR product, but they potentially could be estimated from the initial waveforms.

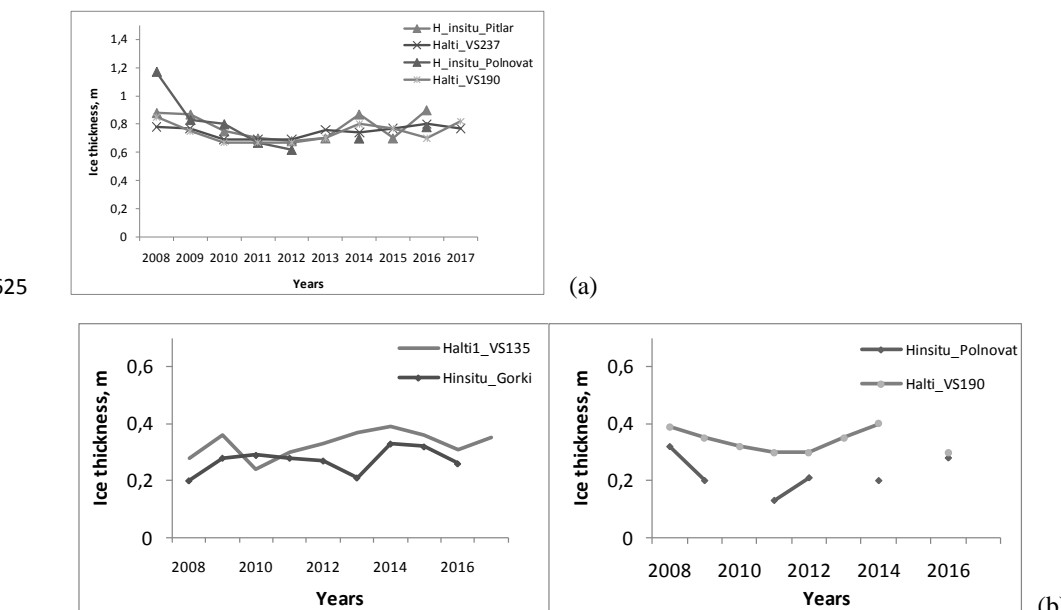


**Figure 15: Interannual variability of the maximum ice thickness (a) and ice thickness on 1 December (b) retrieved from altimetric measurements and observed at gauging stations.**

**6.2. Impact of different geophysical factors on radar return**

One source of uncertainty in the retrieval of ice thickness can originate from neglecting the role of snow in altimetric signal scattering. Willatt et al. (2011) demonstrated that the Ku-band electromagnetic wave scattering by snow at nadir is low and we neglected the presence of snow on ice. However, in winter the snow cover undergoes both thermal and mechanical

transformation: re-crystallization, wind compaction or redistribution, refreezing after melt/slushing by atmospheric or river water. These processes can change the snow wetness or



surface roughness and, thus, modify surface scattering (Rémy et al., 2006) and its contribution to dispersion of the returned signal.

Over ice sheets and over land, snow cover affects the altimetric radar return echo through its extinction depending on snow grain size, water content and depth (Legrésy et al., 1997; Papa et al., 2001; Slater et al., 2019; Lacroix et al., 2007). Over snow-covered or snow-free lake ice, the behavior of the altimetric signal has been studied by Kouraev et al (2015) and Beckers et al. (2017). However, the effect of snow on the backscattering processes of river ice has never been investigated. According to measurements at the gauging stations, snow depth on the ice cover of the Ob River rarely exceeds 0.40 m. At the beginning of freeze-up period, snow rapidly accumulates up to 0.20-0.25 m and then grows gradually from January until April. Over the surrounding forested land of our region, with typical snow depth values of 1-1.2 m, the impact of snow on the winter backscatter decrease is in the order of 10 dB. This effect is clearly visible on the trailing edge of the waveform (see Figure 4a), resulting in the echo shifting towards higher values with time. In radar waveforms over rivers, the increase in backscattered power of the trailing edge is observed only at the beginning of ice growth, when the snow depth/ice thickness ratio is highest (~40% of total snow+ice thickness). Starting from January (snow depth ~ 20-25% of total thickness), the variability of the trailing edge power from cycle to cycle is low, especially compared to the decrease in the main peak power. Based on this fact, we considered that for the establishment of the empirical relations proposed that the impact of snow on backscatter can be neglected. Using precipitation data from the nearest meteorological station, we noted that not all heavy snow accumulation episodes affect the backscatter over river ice. In several cases, snowfall resulted in a backscatter increase of 1.5 dB. The smoothing procedure applied to cumulative dSig0 series helped to eliminate this effect. Moreover, after adding *in situ* snow depth to the ice thickness the Hice-Sig0 correlations worsen.

While considering that the main peak return power comes from ice/water interface, one can suggest the impact of the ice bottom roughness on the radar echo scattering (Atwood et al., 2015). The roughness of the ice bottom is high at the beginning of the freeze-up period, especially in the upper reach of bridging areas, where floes of different size juxtapose and accumulate underside. Further congelation of inter-floes volume as well as ice growth lead to leveling of the ice lower boundary. This means that the effect of altimetric signal dissipation by rough ice bottom can be expected mostly at the beginning of freezing. We consider that this effect can be observed in the backscatter time series. During the first two cycles after freezing start the dSig0/dt is maximal. However, the further decrease in backscatter (due to the waveform peak power) cannot be explained by the ice bottom roughness, which reduces with time and, thus, has to increase the signal reflection for nadir-looking instruments. This support our assumption that the progressive decrease in peak power (and consequently in backscatter) of winter river waveforms reflects the signal diffusion of thickening ice.

The ice internal layering is important for backscattering of the radar signal (Legrésy et al., 1997; Nilsson et al., 2015). The layering can significantly affect the leading edge of the waveform resulting in biased retracking of the surface height or in the backscattering increase due to the reflection from internal layers. In spite of the high nose of the Jason waveforms over the rivers, we are likely seeing the cumulative effect of the layering as the gradual migration of the echo power upward in the first ten bins of the leading edge with time. However, this migration works in the opposite way for the observed dynamic of backscatter governed by the reduced power of



the main peak. Under the climatic conditions of northwestern Siberia, the ice layering characterized by dense reflective icy surfaces is rare as the air temperature of winter warming episodes never approaches the melting point. Daily positive temperatures lasting several hours can occur starting from the end of March in the southern portion of the study area and 1-2 weeks later to the north. During this time of the year, the ice is well developed and almost reaches its maximum thickness. The layering can also occur after river water floods the ice surface through cracks. According to *in situ* observations at gauging stations, this phenomenon was observed in the last several years at the end of the ice season in the southern portion of the study area. Both warming episodes and flooding events lead to a backscatter increase in the order of 1-5 dB and render altimetric ice retrievals difficult by the end of the ice season. The highest underestimation of 0.15-0.20 m is observed in such cases.

The ice internal structure can also affect the backscatter value. During ice formation jamming and ridging can occur on Arctic rivers. On the Ob River, in the area of gauging stations, ridging is rare. However, there is no information about the state of the ice at other ungauged reaches. We can only speculate that the ridging/hummocking could be one of the reasons explaining the high difference in the coefficients of equation (1) determined for virtual station 109. On Landsat-8 images acquired between 2019/04/25 and 2015/05/01 (not shown), the irregular spatial ice structure in the area of  VS 109 indirectly confirms our hypothesis. More studies involving the simultaneous analysis of SAR imagery and altimetric signals could help to clarify this issue.

### 6.3. Altimetric river ice product: importance and potential applications

### 6.3.1 Climate research and long-term regional development strategy

As we noted in the section 2.1, during the last 10-15 years clear tendencies are observed in later freeze-up, earlier melt and the thinning of the ice cover (Figure 2b). Knowledge as to whether the detected changes are robust or not is important for climate research and for long-term regional development planning strategies. The most pronounced changes in the snow and ice cover have been reported for the southern and mid-latitude regions of the Northern Hemisphere .Observations at the southern gauging stations of our study area are located just above 60°N latitude are not complete and show a significant number of outliers. They are not suitable for robust evaluation of changes. The decreasing number of *in situ* observations and degradation of the quality of the time series are a good argument for boosting the development of satellite methods for freshwater ice monitoring. The method proposed in this paper show a good sensitivity of altimetric instruments for river ice changing  and promising results were obtained. In a future investigation, following improvements, this method could be applied to earlier altimeter missions of the same series and time series of the satellite- derived ice parameters (ice onset, melting start, ice thickness could be studied back to 1992, when the first altimetric satellite mission of this series, TOPEX/Poseidon, was placed into orbit. A similar approach could be adapted for the recent Copernicus program altimetric missions, such as Sentinel-3A and 3B. The combination of several altimetric missions will permit a densification of virtual stations and an extension of ice monitoring toward the upper reaches of the Ob River, which are more vulnerable to climate change.





### 6.3.2 Winter ice roads operation forecast

In many regions with the seasonal ice, frozen rivers enhance the connection and supply of the numerous small and even big cities. Many remote villages linked in summer to supply centers only via expensive helicopter or boat transport, get an opportunity to directly access the main land transport arteries using frozen-ground and ice roads. An importance of the ice roads is highest for the Arctic regions, where construction of the bridges through the rivers is restraint by the presence of permafrost and its destabilization.

One of the good examples is the Salekhard city located on the north of the Ob River near the Polar Circle in the zone of discontinuous permafrost. The city has 50 000 habitant and is supplied primarily via the Northern Railway, which connects the small town Labytnangi on left bank of the Ob River with European part of the Russia and main supplying centers. Merchandises from Labytnangi are delivered to Salekhard by ferry. Every winter the ice road is constructed to ensure the transfer of goods and people. Due to security reasons the ferry ceases the operation after appearance of first ice. The ice road construction begins when the ice thickness attains an allowed value of 20-25 cm (Instructions..., 1969). The ice road operation usually starts 3-4 weeks after beginning of freezing. The average date of the ice road opening is 30 November. The road operation closes gradually starting from limitation of the lorry load in the middle of April until full halt in the beginning of May. The ferry connection restores about 3 weeks later. Between the ferry and the ice road operation the connection is ensured via hovercraft boat only for a limited number of passengers or in emergency cases.

The dates of the autumnal halt of ferry operation for 2010-2019 correspond very well to dates of the first ice occurrence on 4 northernmost tracks of the Jason satellite located in 65-75 km southward from the city. The observations on the VS 112 and 9 are especially good for the short-term forecast with 2-8 days delay (average 4 days). During 2010-2019 only one exception was observed in 2015, when the ice installed first in Salekhard river reach and then, in several days later in upper and lower adjacent reaches of the Ob River. The ice road opens when the ice thickness in natural conditions reaches 30 cm. On the road ice is thicker as it is grown artificially by pumping the water on the top of the ice for later congelation. As we noted in the section 6.1, for the dates of beginning of the winter circulation (December,1th) our retrievals have a tendency to overestimate the ice thickness. Considering that an average value of the dates when Hice_alti at four northernmost VSs reaches 30 cm may provide an estimate for the road opening, we predict the beginning of circulation 2-3 days earlier. Although the predicted average is quite good, in half of the years the predictions differ from observations for more than 5 days (with 11 days in maximum). We consider that at the moment the altimetric algorithm and the ice thickness product are not sufficiently mature for the forecast of the ice road opening. Nevertheless, their accuracy is sufficient for climatic perspectives as we capture quite well the interannual variation of dates of ice road opening (Figure 16a).

Earlier we demonstrated the wide heterogeneity of freezing conditions along the Ob River. Considering the 65 km distance between Jason virtual stations and Salekhard city, the forecast has to be adjusted using an additional predictor. Such a predictor could be derived from optical or SAR images acquired after the beginning of freeze-up and confirming that the freezing in both reaches progresses in a similar manner. We are also hopeful that with the use of the Copernicus satellite altimeters Sentinel-3A and 3B an improvement can be made in the retrieval of ice





thickness. These satellite missions carry more advanced altimetric SAR instruments which footprints consist of narrow band and return signals are less contaminated by land. Though the nominal repeat frequency of the Sentinel-3 satellites (22 days) is not suitable for operational applications, they provide five overpasses within a 25 km distance around the ice road and, thus,

the temporal resolution of observations may be significantly improved. The combination of data from the Jason and Sentinel-3 missions could also be fruitful.

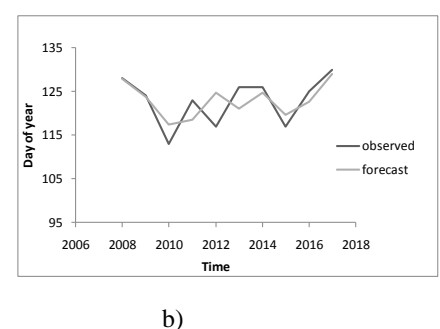

a)                                                          b)

**Figure 16: Observed and predicted dates of ice road opening (a) and closure (b).**

For the prediction of dates when the ice road ceases its operation, the use of the northernmost Jason virtual stations is not possible. Hauling on the ice road closes before the altimeter detects melt onset in this reach. However, information on melt onset at the virtual stations located in the lower reaches can be used. We found that a second record of melt onset (ensuring that the first is not an outlier) detected at virtual stations within the study area can serve as a predictor of dates

of ice road closure at Salekhard. The correlation between these dates is significant (p-value is 0.025) and equals to 0.70 (Figure 17a). The forecast delay is negatively related to melt onset dates (Figure 17b). After application of a correction on the delay, we obtain forecast dates with a RMSE of 3 days.

The Salekhard ice road is very well instrumented, monitored and maintained by local authorities,
thanks to the high demand for its use and high traffic flow. In other regions, ice roads connecting small cities and villages are less monitored and access to operational information is poor. Moreover, many intermittent river crossings are developed each year by local people. Often, the lack of information on the state of the ice results in accidents and requires intervention by the Emergency Service. The demonstrated capacity of the first version of the altimetric river ice

product to provide a tool during the operational period of the ice road on the north of the Ob River is quite promising. Further product improvements and will allow a development of predicting criteria that could be adapted to other reaches of the Ob River. The combination of different altimetric mission and different satellite instruments could be envisaged as one of the most perspective approach providing enhanced spatio-temporal resolution.






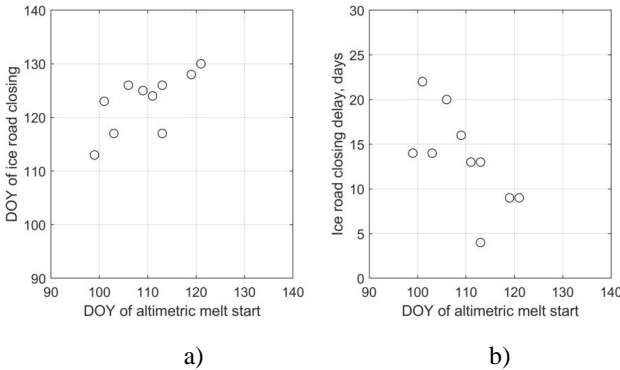

Figure 17:. **Relation between altimetric melt start in ROI and Salekhard ice road closure (a) and delay between these dates (b).**


## 7 Conclusion

Present paper investigates a potential use of satellite radar altimetry for monitoring river ice parameters such as freeze-up, break-up and ice thickness in the context of elaboration of a satellite product and its application for climate change monitoring and for operation of ice bridge roads in the Arctic.


1. An algorithm was developed based on the analysis of backscatter coefficients from the Jason-2 and 3 satellite altimeters and provides a good estimation of river ice onset with an accuracy of ±10 days (corresponding to the 10-day satellite overpass frequency) in 90% of the cases.

2. River ice melt consists of two phases: thermal degradation and mechanical break-up and movement. The algorithm detects well the beginning of thermal degradation with the same accepted accuracy of ±10 days for 88% of cases.


3. River ice thickness retrieved from the altimetric measurements via empirical relations with *in situ* observations is with an accuracy (expressed as RMSE) of 0.07 to 0.18 m. The lower accuracy is observed in the southern part of the study area due to the complex ice texture - often occurrence of ridging and hummocking in the area of the satellite overpasses.


4. The spatio-temporal smoothing of satellite-derived river ice thickness at 48 virtual stations along the 400 km reach of the lower Ob River allowed for the generation of the weekly maps generalized in the form of an annual spatio-temporal product. The ice thickness time series could be extracted for any location and used for climate and ice road operational purposes.


5. Using this first version of the product, we demonstrated that the dates of opening of the Salekhard ice road can be predicted with a 4-day delay. However, we consider that the current version of the product is not sufficiently mature for this forecast as it overestimates the ice thickness at the beginning of the ice season. Errors in the prediction of dates of the ice road closure are within 3 days and the delay varies between 4 to 22 days depending on earlier or later


melt onset in the upper reaches of the Ob River. This first attempt shows a promising potential for a river ice product targeted at applications in support of operational services. The algorithm





and product could be significantly improved through a multi-mission and multi-instrument approach, and predicting criteria could be extracted and adapted for other reaches and ice roads.

**Acknowledgements**. This research was made possible with support from RFBR project № 18-05-60021-Arctic and ESA (EO Science for Society Element) LIAM project (Contract No. 4000130930/20/I-DT).

**Author contribution**. All authors contributed to the data collection, algorithm development, analysis and presentation of results.

**Declaration of Interests**. The authors declare no competing interests.

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
