# Peer review of "River ice phenology and thickness from satellite altimetry: Potential for ice bridge road operation and climate studies"

_The Cryosphere, 2020_

## Referee Comment (RC1) · Anonymous Referee #1 · 4 Feb 2021

The paper titled "River ice phenology and thickness from satellite altimetry. Potential for ice bridge road operation" by lead author Elena Zakharova and coauthors explored using radar altimetry data to infer river ice phenology and ice thickness. By conducting the study over the lower Ob river in Russia, the authors reported accurate retrieval of river ice phenology and ice thickness by comparing ice phenology/thickness estimation from altimetry data at virtual stations to those obtained manually and those from the in situ gauge records. The authors have done an excellent job of describing the details and nuances of the ice processes, and how it can complicate the radar backscatter signals. The authors thoroughly described the uncertainties of the studies and provided valuable recommendations for future work and an assessment of the social impact of the conducted research.

[Figure]

My major concerns with the paper is the lack of clarity in the methods section. The authors reported many interesting results however, as I detailed below, not all of their methods were well described. Please see below for my comments. I would recommend the authors make the methods clearer and make the figures more informative and easier to read. Overall, I think the paper is well written and the implication and uncertainty of the study thoroughly discussed.

Major comments

Figure 4: are the dates in the format of dd-mm? I suggest to make the dates more explicit and move the surface types to a more prominent places (e.g. using a.b.c and refer to the surface type in the caption)

On line 373: the authors argue that decrease in backscatter is proportional to gain in ice thickness. If this is the argument, would it make sense to plot the changes in ice thickness (Hice) against cumsum(dsig0/dt)?

On line 375: the authors mention cumsum(dSig0/dt), which should be negative for the freeze-up period. However, in Figure 5b all the values are positive along the x-axis. Am I missing something here?

Lines 387–393: calibration and validation using the eight VSs were mentioned in this paragraph, however, no detailed methods were provided in text nor in figure 6. I would highly recommend providing how the calibration and validation were carried out.

Figure 5: labels for the subfigures should be placed at more prominent locations. The legends should be placed at a consistent location of the figures.

Figure 6: it is nice to have a flowchart to guide the readers through the processing steps. However, I found the one presented here hard to follow: data and procedure are better separated and represented using different boxes.

Line 400: shouldn't phenology estimations be compared to gauge records closest to the VSs?

Line 413: please clarify how "close to zero" was defined.

Figure 7: the author should discuss why for melt end, the results from the manual algorithm have a much bigger bias than that from the automated algorithm.

Figure 8, 9, and 10: the authors need to justify why for the gauge data the mean was used and for the VS data the median.

Line 444: "significant variability"—does this refer to the difference between the manually determined and the gauge mean, or does it refer to the variability amongst the gauge data. Need clarification.

Line 440–451: it is easy to attribute years lacking north-south detected difference to local effect. However, such explanation is not satisfying without any evidence backing up the claim, especially given that so many factors (e.g. uncertainties in the percentage of pixels of different surface features) can affect the detected dates.

Figure 10 and 11: the authors need to clarify or show the location of Tr187 in the x-axis label.

Figure 11: highly recommend the authors using color to represent data from different years or use a better way to differentiate the data.

Lines 494–500: the active melting period (melt end) is highly dynamic and presents a challenge, as the authors noted, to any automated algorithm. I think the patterns presented in Figure 11b–c and the interpretation given in the paper is very interesting. However, it will make a much stronger argument if similar patterns contrasting the similarity at the melt start and variability at the melt end can be found in the in situ gauge data.

Figure 12: it is rather hard to see the rivers when everything is frozen. I suggest the authors add some labels on the images for a few key locations discussed in the paper to orient the readers.

[Figure]

Lines 507–508: the authors need to clarify whether the correlation and RMSE were calculated based on the gauge that was left out of the parameter estimation step.

Lines 573–576: the authors mentioned both approaches of building relations were evaluated at 11 VSs nearest to gauges. However, only two sets of values were presented for VS 109 and 12. I think it will be really helpful for the authors to explain Table 2 in detail since it is pivotal to the understanding of the algorithm performance.

Line 592: please explain the term "ridging flags"

Figure 14: instead of using decimal year in x-axis labels, it will help readability by converting it to month.

Line 613: could optical remote sensing provide information on the ice onset on rivers?

The authors should add scale bar and north arrow for all the maps presented in the paper.

Figures in the paper are of various styles and should be made in a consistent style with consistent places for subfigure labels and the legend

Minor comments

Line 33: "erosion of channels and banks": need citation here.

Line 38: "catastrophic flooding": need citation here.

Line 75: "state and regime" is there a difference between the two?

The paper needs some language editing.

Line 126: proposes should be "proposed"

Line 198: delete "in" before later freezing, and "in the" thinning of the ice cover.

Many cases of unnecessary "an" and "the" (e.g. Line 187: "The Matplotlib Basemap Toolkit"; Line 573: delete the "the" before "both approaches"; Line 18: for "an" estimation (should be "the");

---

## Referee Comment (RC2) · Anonymous Referee #2 · 8 Feb 2021

General comment

This study analyses the potential of radar altimetry to detect the seasonal freeze and melt events as well as the ice thickness on the Ob River, Russia. The study employs the backscatter (sigma nought) signal from Jason-2 and Jason-3 missions, as well as in situ observations from several gauging stations.

While the idea behind the study is clear and definitely shows a good potential of the radar altimetry backscatter to monitor river ice, the presentation of the methods and results lacks structure and scientific rigour. The quality of figures and captions is in my opinion poor. I also recommend to concise the paper by focusing on fewer aspects and removing redundant parts. I also insist on a professional English proofreading to simplify and smooth the style and improve the clarity. Please check the terminology: reflect vs scatter, bias vs difference vs delay, algorithm vs method vs approach, etc. Please also check the comparatives: you very often use "higher", "more", "less", etc. without mentioning what do you compare it with.

Please find my comments below. There are both specific and more general ones, but I found it easier to comment the parts at their place.

Introduction:

In my opinion too long and needs concision. Think about subsections and narrow down your story from general introduction to your specific objectives. For example: The importance of river ice monitoring → remote sensing observations so far → altimetry principles and observations of fresh ice (1-phenology, 2-ice thickness) → Jason mission → knowledge gap → your objectives.

49-50: I think you show with your own study that the word "excellent" is a little bit of a stretch.

51: please specify: you mean here the temporal resolution and not the frequency of the EM wave, correct?

61: "Active sensors operating in the microwave region are weather independent and provide higher spatial resolution" – higher than what? What about high- and medium-resolution optical sensors? Please include a sentence or two for the consistency.

63: to my knowledge, "largely" is an exaggeration for such studies.

75: please explain what means "water state and regime"?

74-79: there are repetitions, please compact

81, 82: what are the "materials"?

83: please specify "reflected" or "scattered"?

84-101: I propose not to go deep into the SAR studies, as they are not very relevant to your study. Instead, formulate briefly the main principles of the altimetry, how the data are collected, what are the data obtained (backscatter, waveform, what else?)

You can also combine some of the information with the paragraph 54-79.

97: Higher than what?

98-99: is the penetration depth important here considering that the typical ice thickness < 2 m? Would C-band have insufficient penetration? I would focus on the properties of Jason mission later when you describe it for your study.

103: you haven't mentioned before, that the altimetry missions are accompanied by radiometers

103-112: is it only about resolution or the different nature of measurements plays a role too?

113-117: is it already a description of the method you are using?

118-132: this reads rather specific and too long. Please consider revising, shortening or moving to the Methods section

132-142: this part is a presentation of your own results and should not appear in the Introduction

143-156: this is a place for your objectives, please state them instead of listing the sections

151: please introduce and explain what is a virtual station before

2. Regional setup

There is not even one reference to all the information you provide for your study area. Please provide them.

Please work on the study area figure. Make it colorful. There is no way to understand where is the study area on the globe, especially in black and white version. Is it topography or is it an optical imagery? The scale of the globe is too coarse and of the zoom-in is too fine. The zoom-in area lacks geographical coordinates and the tracks are strangely projected. The circles (gauging stations) are invisible. Please think about the reader who has no idea about your research.

173: please show these cities on the map.

175-205: If this is part of your results, please move it to the corresponding section. You also introduce your in situ data only later.

195-199: please show and explain the trends, I do not see any.

Figure 2: please improve the quality of the figure. Is that mean on figure a)? why only three stations of five are used?

3.1 Altimetry

Please provide more information on the data selection: tracks, cycles, virtual stations, repeat overpass etc.

231-233: please provide more information on how did you extracted the data via portal (programs used, codes?) and how did you use Landsat images for the selection. Is it a manual / visual selection? What exactly are the track numbers in relation to cycle and pass numbers?

3.2 Optical imagery

235-243: I do not think you need to describe the optical imagery in a separate data section. The first purpose you already mentioned in the previous section. The second purpose is not represented later in the paper substantial enough. You can describe the images in place.

I think the radiometric measurements deserve to be described in a separate subsection.

Please explain what do you refer to as an altimetric measurement, i.e. you talk about a precise selection over the river channel but at the same time mention multiple times that the signal is contaminated by the surrounding land. I encountered later the 400 m band oscillation – please refer to it early enough.

3.3 In situ data

254-255: please clarify what is measured daily and what is measured 3-6 times per month.

256: do they drill a new hole every time for a new measurement?

How do you credit the data from gauging stations? Are they publicly available?

4. Methods

4.1 This does not look like methods to me. You describe the environmental processes and the response of altimetric backscatter on them. Create a separate (sub-)section and provide references to all the processes you describe. Exclude your own results and move them to the Results section.

I might disagree that the presence of ice increases the specularity. The ice-water interface should be rough, shouldn't it?

Mention the difference between the nadir-looking altimeter and side-looking SAR instruments (i.e. low and high backscatter from rough and smooth surfaces).

Figure 3: This figure would be more informative if you show a mean backscatter with the range of values across all the stations for one typical year. Then you would guide the reader through the seasonal evolution of backscatter and attribute the backscatter change to environmental processes. Adding the in situ dates of freeze up and melt begin/end onto figure would help. The plot in its current version does not add any value in my opinion.

304-305: I am not sure that wet snow leads to an increase of the backscatter. To my understanding, wet snow attenuates the signal stronger than the dry snow. Please provide a proof.

312-313: the sentence is not clear.

314-315: I do not agree that the waves are increasing with the water level. Should depend on the wind?

319: Provide more specific title for the section

Was this algorithm ever applied already or it is brand new?

320: referring to an example figure as described above would help to follow.

320: not sure about the word "annual" in this context

320: how do you define a peak?

321-322: "In the case of a multi-peaky recession limb, this peak should be of order of spring and summer peaks". Why? What is spring and summer peaks?

322: what is recession limb?

322: "If the selection of peak is not straight forward..." – how does the algorithm select a peak?

324: Explain why and how do use the dTb and add dTb on the figure described above.

326: why exactly t-1, t+2? Why 2K? Can you refer to previous studies or explain this choice?

Please provide a conceptual scheme for the automated algorithm.

351-360: if you do not use this approach in the end, I do not see a reason to include it in the paper. Or "guided over river ice by the main waveform peak" means that you use this information? Please clarify.

You never refer to the Fig. 5b. The information on a) and b) seems to me redundant. Is a) just one example? Why did you choose only two virtual stations on b)? Why not to include all gauging stations? Improve the quality of the figures, avoid work-in-progress axis titles and legend entries.

388: how did you create a virtual station? Did you use the same stations for the dates of ice freeze-up and ice break extraction?

390: what is "ice thickness relations"?

391: by "extrapolated' you mean "applied"?

391: what is "other "main set""? the rest 40 stations?

393: 2) not clear what do you correlate with what

395: the scheme needs to explain better what are you doing. What do white and grey color mean? What are the grey outlined circles?

Results:

Maybe it would make sense to make subsections and give them titles. It is not easy to understand when the topic is suddenly changing. For example, **algorithm evaluation, interannual variability, spatial variability**, etc…

400-403: this paragraph seems redundant to what you already described in the Methods.

5.1 Ice phenology

Please start with an accurate description of your results.

405-408: this seems to me a repetition of your methods section, at the same time rather a hypothesis of what you think is captured by the altimetric measurements.

407: would open river water always appear rougher than the young ice? Even if the flow is calm and there is no wind?

408: decrease compared to open water?

411: I think "bias" is not correct term for what you describe since bias has a direction. Difference would do better?

411: "accuracy for Jason" – not for the altimeter but for your algorithm, for the date extraction?

411: how many retrievals are there? 8 stations by 12 years? 48 stations?

413: what is close to zero and if it is not exact how did you end up with 56%?

417: less accurate than what?

 491: what results and what do you refer to here?

424: better than what? Do you mean the algorithm or manual approach?

426-428: what means "least" here and what do 54% and 67% refer to?

430: what do you mean by outliers? Please explain. Is it possible to adapt the algorithm? What about radiometric measurements? Do they not help? Can the manual peak extraction be affected if the person knows in advance the true date of freeze up and melt from the in situ measurements?

433: melting **and freezing** before 10 April and after 10 June? How did you define these dates?

Figure 7: please use the total number instead of norm pdf. Please mark the 90% cases on the graph. How do you explain

- Bimodality and bias towards earlier freeze up of the automatic algorithm for freeze up? Do I understand correctly that negative values correspond to the earlier date?
- Bias towards the later melt start for both automatic and manual approach?
- Poor results for the melt end for both approaches, especially for the manual approach?

436: as mentioned, this is generally not a bias.

437: why 11 stations? You mentioned 48 in the text. Why not to include all of them?

438: I thought that training set applies to ice thickness retrieval? Why do you need a training set for the ice dates?

439: how do you evaluate that? How do you define what is a good sensitivity? I think you need to find a rigid approach to evaluate your algorithm performance. Maybe you do not even need to go on with the automatic algorithm in describing multiyear variability and spatial variability? You can choose manual and show the results only for it as it is clearly better.

440: not the altimeter but your algorithm?

441: what do you mean by "noisy"? Noisier than what?

442: "Nevertheless, a clear coherence exists between the corresponding time series". Please clarify this sentence. What do you mean by coherence, what are the corresponding time series, and why is it clear. Not clear to me.

439-445. Please find more consistent and smooth way to describe your results, than by picking some years and saying that some events are noticeable.

Why on Fig. 9 you present the different events (melt start and melt end) AND different approaches (manual and automatic)? Please use the same approach for consistency.

446: why important?

447: Why Salekhard station is not included from the very beginning? "Adding" to where?

Describe this gradient, i.e. earlier freeze up at the northern stations? If you talk about trends, provide the parameters, i.e. days per degree (or km).

448: average what? Why "calculated"? Are you talking about the algorithm?

449-451: either show what you mean or remove. Very difficult to follow. What years, what half?

452: the same comment as before, please describe these gradients more detailed.

452-453: what do you mean by "gradient in the order of 20 days"? What is denominator of the gradient? Km?

454: the same here.

Figure 8: why do you use median for the altimetric data and mean for the gauging stations?

Please use legend to display the information in the caption, it is very difficult to follow.

458: does it mean 4 gauging stations average? Why 20 virtual stations? How did you decide which ones to include?

Figure 9: as already mentioned, make it consistent with the approaches. Comments for Fug. 8 applies here as well.

Figure 10: what is the x-axis? Please use more reader-friendly labels. Use figure titles in addition to a,b,c. Why do you show manual approach for a and b and the automatic one for c? Please be consistent. Please indicate stations on the red line. Use legend to explain the lines and shaded areas.

474: what do you mean by "different years"?

479-500: this whole section is a little hard to follow. How did you select the shown years? How certain you can be with the algorithm results? What about in situ data? I am not sure that the way of presenting those results as in Fig. 11 is optimal. Would it be possible to use mean values for the entire period of observations instead of only three years?

486-487: "This is shown clearly on a Sentinel-2 optical image..." I do not see anything clearly on the Sentinel image. Please use color version, and mark on the image what is what (land, ice, water, main channel, narrow channels, etc). The same comment applies to the Fig. 12b.

489: branches, not brunches

489-491: Please show the image. If there is a polynya until March, does it mean it is not freezing at all during some years?

492: "between the branches" – do you mean main and secondary branches?

494-496: "At the beginning of ice degradation local morphological controls only play a small role (Figure 11b). Their role amplifies during mechanical break-up, which is better captured by our automated algorithm (Figure 11c).". How do Figures 11b and 11c illustrate both statements?

497: uncertainty in what?

497: higher that what?

500: please explain Fig. 12b in more details and support the statement.

5.2 Ice thickness

Would it make sense to show the Fig. 5b here? To support the equation 1?

501: what means different gauging stations? All of them? Some of them?

501-503: how do you come up with 9 runs? Please explain the approach more detailed. How to read Table 2? How did you sort rows in the Table 2? Why there are two Pitlar, two Gorki, and two Muzhi entries? Why did you decide to work with an individual relationship for each station and not with one universal relationship? How would Fig. 5b look if you include all the gauging stations? What do

the correlation coefficients and RMSE describe? The relationship between backscatter and ice thickness? Then RMSE should be not in m but m per dB? Why correlation coefficient and not coefficient of determination, especially considering that the relationship is not linear? Or R and RMSE describe the relationship between in situ and altimetry ice thicknesses?

509-510: How do I see on the Fig. 13 that those stations are northern? I'm confused here, I thought that the stations 138 and 161 are used in the training set? Please include all the stations in the Fig. 13 and use gradient color scale for the timing of measurements. The same for the spatial distribution of the stations – maybe different point style?

Fig. 11. Please consider revising the Figure as already mentioned. It should be in color, bigger in size and better resolved, use different line colors and styles (if you keep individual years). In the left part of the graphs, especially on c) nothing is visible. Explain x-axis, what is Tr187? Give the direction of the north. An additional map with the location of the shown stations would be useful here.

Fig.12. Please see the comments before. What band do you use or is it a color composite? Also please include coordinates, scale. Please put the labels on the in situ and virtual stations. What is the difference between open circles and squares for the virtual stations?

540-542: where could I see it myself?  In the table 2? Then please organize the Table in a clearer way.

Table 2: please organize it in a more understandable way and provide more explanations on how to read it.

555-608: Please include these sections into the Results. This is clearly a continuation of your results, and not discussion.

557: please indicate which parameters did you use, there are many different ones.

562: please include a short explanation how did you choose the window size

Fig. 13: please see comments to this figure above.

569: please change the word "analog", it is unclear what you mean.

569-576: I think it would make sense to move this paragraph to the beginning of the section, and to describe the ice thickness product after that. Do I understand correctly that you use the second approach to create the ice thickness product?

571: why only four stations if you have five? Please mark on Fig. 1 the clusters of virtual stations attributed to the in situ stations based on their proximity.

571-573: please reformulate sentence, it is unclear. What is main VS? What is time shift?

573: "The performance of the both approaches was evaluated at 11 virtual stations nearest to the location of the gauging stations". Now I am again confused. These 11 are in the training set? But why?

574-575: how can I see the results of the first approach?

584: "The interannual variability in maximum ice thickness retrieved from altimetric measurements at many virtual stations indicates a clear decrease from 2008 to 2012." Is this something shown in Fig. 15? Then refer to the figure 15 right there. What means many? Why did you include only 2 of in

situ and 2 of virtual stations in the Figure? Could you show a plot similar to the Fig. 8-9,, which would include all of the stations?

592: please explain what is ridging flag

Figure 14: I suggest to combine the yearly plots into one multiyear plot, and add some vertical lines to mark the timing - to show the interannual variability. Please indicate the north-south direction, add unit on the color scale. Please describe results shown in Figure 14 in the text. Right now, you only mention that you created the product. What is the area we see in this product? What is the extent of it?

600: more disagreement than what? Is this statement shown in Fig. 15? Refer to the Figure.

601: "This does not strongly contradict expectations as for most virtual stations this disagreement lies within estimated RMSE values (0.07-0.18 m).". Please reformulate and explain what you mean. I cannot follow logics here.

602: "Besides the reasons noted earlier..." – where, what do you refer to?

603: why there is a degradation in the in situ time series? Lower representativeness – lower than what?

604-608: Please explain what are you trying to point out here, it is not clear.

609-610: how do you demonstrate that they are accurate enough?

611-612: do you present this result? Is it something that I can see in Fig. 15?

614: how could the date of first consolidated ice be detected and why then you did not use this approach?

614-616: please reformulate the sentence to support your suggestion

618-623: is this something recommended in the previous studies (then please include references) or this is your own hypothesis?

Figure 15: please see above the comments to this Figure. Why there are two plots for the b)?

The Discussion section has only one subsection which actually discusses the results in the context of the physics of the radar signal return and potential errors related to that. Another subsection considers a study case of an ice road and applies the developed methods to this study case, which again reads more like results. I was missing a discussion of your results in the context of the relevant studies (which are not only methodologically relevant).

I have an impression that the whole section 6.2 would benefit from shortening and compacting. Some of the discussed issues are mere speculations and raise more questions than give answers, and some other points are not really relevant to your study (e.g. layering as you pointed out in 682).

646: "…grows gradually from January until April." – is there no wind redistribution?

649: Please include the figure 4 here if needed

652-653: please explain better what you mean with the ratios and 40%, and 25%.

660: you mean the power relationship becomes weaker? What correlation do you mean?

665: "Further congelation of inter-floes volume as well as ice growth lead to leveling of the ice lower boundary.". Can you support this hypothesis? Do your observations of the decreasing backscatter (669-671) contradict this statement?

668: please explain what do you mean by the first two cycles

669: please explain what do you mean by the note "due to the waveform peak power"?

703: I do not think that we have seen any clear tendencies in the referred section.

705: can changes be robust?

706-707: please provide a reference for this statement

708-709: how do you define an outlier? These are the valid observations, right?

712-721: this part sounds like an outlook to me and should belong to the Conclusions section.

746: please expand on what you mean by a delay. Delay to what? Is it the best term in this case? Please check it further in the text as well.

746-748: this sentence is hard to follow. Exception from what? Why this is an exception? Where is the Salekhard river reach?

751, 754: what is circulation?

754: earlier than what?

762: perhaps, "could be adjusted" instead of "has to be adjusted"?

760-771: this also sounds like an outlook and can be combined with 712-721

Figure 16: You never mention Figure 16b in the text.

778-780: please reformulate the sentence: what is the second record of melt onset?

780: correlations between which dates? Is the information on Fig. 17a similar to the Fig. 16b?

780-783: please explain better what are you doing here. How did you produce a forecast? Is forecast is just the melt onset day derived from altimetry (whatever the second record means)? This is a bit confusing.

789-794: again outlook, can be combined with previous ones and moved to Conclusions

Figure 17: Please explain what ROI are you referring to? In Fig. b) please correct the label of y-axis – the current label is not clear – what is ice road closing delay?

Conclusions: please use this section not only for a dry summary of the results but also for a more general wrap-up (reinforcing the problem importance, filling the knowledge gap, outlook and recommendations etc).

---

## Author Comment (AC1) · 15 Mar 2021

The paper titled "River ice phenology and thickness from satellite altimetry. Potential for ice bridge road operation" by lead author Elena Zakharova and coauthors explored using radar altimetry data to infer river ice phenology and ice thickness. By conducting the study over the lower Ob river in Russia, the authors reported accurate retrieval of river ice phenology and ice thickness by comparing ice phenology/thickness estimation from altimetry data at virtual stations to those obtained manually and those from the in situ gauge records. The authors have done an excellent job of describing the details and nuances of the ice processes, and how it can complicate the radar backscatter signals. The authors thoroughly described the uncertainties of the studies and provided valuable recommendations for future work and an assessment of the social impact of the conducted research.

My major concerns with the paper is the lack of clarity in the methods section. The authors reported many interesting results however, as I detailed below, not all of their methods were well described. Please see below for my comments. I would recommend the authors make the methods clearer and make the figures more informative and easier to read. Overall, I think the paper is well written and the implication and uncertainty of the study thoroughly discussed.

>> The manuscript was significantly edited according to recommendations of Referees. The section of Methods was extended. The sections Results and Discussion were reshaped. All figures were revised. Several figures were removed or combined after revision of corresponding paragraphs as recommended by Referee 2.

**Major comments**
Figure 4: are the dates in the format of dd-mm? I suggest to make the dates more explicit and move the surface types to a more prominent places (e.g. using a.b.c and refer to the surface type in the caption)
>> Figure 4 is redone.

On line 373: the authors argue that decrease in backscatter is proportional to gain in ice thickness. If this is the argument, would it make sense to plot the changes in ice thickness (Hice) against cumsum(dsig0/dt)?
>> Yes, the relationship Hice vs CumSum(dsig0/dt) is shown in the Figure 5b.

On line 375: the authors mention cumsum(dSig0/dt), which should be negative for the freeze-up period. However, in Figure 5b all the values are positive along the x-axis. Am I missing something here?

>> Yes, the cumsum is negative. The X-axis title should be abs(CumSum(dSig0/dt)). The Fig.5b was removed from the text as it was considered to be redundant by the Referee 2.

Lines 387–393: calibration and validation using the eight VSs were mentioned in this paragraph, however, no detailed methods were provided in text nor in figure 6. I would highly recommend providing how the calibration and validation were carried out.

>> We added a phrase describing our calibration approach and moved part of text from the Results Section to the Methods to facilitate the reading.

Figure 5: labels for the subfigures should be placed at more prominent locations. The legends should be placed at a consistent location of the figures.

>>Figure 5 is redone.

Figure 6: it is nice to have a flowchart to guide the readers through the processing steps. However, I found the one presented here hard to follow: data and procedure are better separated and represented using different boxes.

>>Figure 6 is redone.

Line 400: shouldn't phenology estimations be compared to gauge records closest to the VSs?

>> Yes, we did exactly this validation. Necessary details are provided in the new version of manuscript.

Line 413: please clarify how "close to zero" was defined.

>>  The phrase was corrected for "In 56% of the cases this difference is equal to zero".

Figure 7: the author should discuss why for melt end, the results from the manual algorithm have a much bigger bias than that from the automated algorithm.

>> The algorithm was developed for detection of the melt start. The manual implementation of the algorithm is more accurate than automated implementation (what we can naturally expect during an algorithm development, as not all Sig0 behaviour cases can be coded correctly). As the manual implementation tuned for the melt start detection, logically, it should be not good for the melt end detection. Probably, the dedicated paragraph was not clear. This paragraph was re-written.  "Comparing the dates of altimetry-derived melt onset with the ice state records provided by gauging stations, we conclude that manual implementation of our algorithm detects well the start of ice thermal degradation. In 88% of the cases, the difference between manually-retrieved melt dates and in situ observations of first water appearance is less than ±10 days (Figure 7, b). The automated melt date retrievals demonstrate worth accuracy for detection of the melt start, comparing to the manual ones. Only in 54 % of the cases the difference with in situ melt start observations is less than ±10 days. The automated approach is more efficient for the detection of the melt end as ±10 days accuracy was reached in 67% of cases (Figure 7 b,c)."

Figure 8, 9, and 10: the authors need to justify why for the gauge data the mean was used and for the VS data the median.

>> We were not specific enough – the mean did not mean "arithmetic mean". Changed to "median".

Line 444: "significant variability"ăˇAˇ Tdoes this refer to the difference between the manually determined and the gauge mean, or does it refer to the variability amongst the gauge data. Need clarification.

>> Thank you for catching this imprecision. We meant the difference between satellite and in situ observations. The text is corrected for " Significant difference between gauging and virtual stations (order of 20 days)  is observed only for melt start dates in 2014."

Line 440–451: it is easy to attribute years lacking north-south detected difference to local effect. However, such explanation is not satisfying without any evidence backing up the claim, especially given that so many factors (e.g. uncertainties in the percentage of pixels of different surface features) can affect the detected dates.

>> We agree that the uncertainties in estimation of the ice phenology dates from altimetric measurements can also be a reason of lacking of the North-south gradient for certain years. As it was recommended by the Reviewer 2, we removed these results from the manuscript as this question would take a separate subsection to address all remarks and will result in manuscript extension.

Figure 10 and 11: the authors need to clarify or show the location of Tr187 in the x-axis label.
>>Figure and caption are modified

Figure 11: highly recommend the authors using color to represent data from different years or use a better way to differentiate the data.
>> The figure was deleted as in the new version of the manuscript the phrases referred to the figure  were removed.

Lines 494–500: the active melting period (melt end) is highly dynamic and presents a challenge, as the authors noted, to any automated algorithm. I think the patterns presented in Figure 11b–c and the interpretation given in the paper is very interesting. However, it will make a much stronger argument if similar patterns contrasting the similarity at the melt start and variability at the melt end can be found in the in situ gauge data.

>> On the secondary branches, there is only one gauging station. We could not assess the big/small channel ice phenology difference from the in situ observations. By discussing this difference observed in our altimetry retrievals we wanted to demonstrate the value of the remote sensing methods. As the topic stirred a lot of remarks and questions and their addressing would extend the manuscript, we decided to delete it from the new version of the manuscript. The theme of spatial variability of ice phenology dates will be a subject of a new more detailed study based on multi-sensor approach with more solid proof base.

Figure 12: it is rather hard to see the rivers when everything is frozen. I suggest the authors add some labels on the images for a few key locations discussed in the paper to orient the readers.
>> The figure is no more present in the new version of the manuscript as there is no more reference in the text.

Lines 507–508: the authors need to clarify whether the correlation and RMSE were calculated based on the gauge that was left out of the parameter estimation step.
>> We added necessary details in the text in  the Method Section. When designing the Cal/Val exercise, we initially separated our data on the Cal/Val periods (1:1 split) and found that the

results of validation depend on selection of the Cal/Val years. We consider that the use of the leave-one-year-out method allows to avoid the effect of subjectivity, when separating the short time series (10 winters in our case) into Cal/Val sets. Calibrated by this method parameters, calculated as average from parameters received from 9 leave-one-year-out fitting runs, better account for interannual variability. Using this method, the uncertainties can be estimated for all period and not for one left-out year.

Lines 573–576: the authors mentioned both approaches of building relations were evaluated at 11 VSs nearest to gauges. However, only two sets of values were presented for VS 109 and 12. I think it will be really helpful for the authors to explain Table 2 in detail since it is pivotal to the understanding of the algorithm performance.
>> The table 2 was edited. The title was changed, the pairs of VS-gauging stations were grouped for better visual representation, the additional pair of stations used for intermediate tests was removed resulting in 10 pairs (one gauging station surrounded by 2 virtual stations).

Line 592: please explain the term "ridging flags"
>> we referred to quantitative records describing the quality of the ice near the gauging station. The "flags" was replaced by "events".

Figure 14: instead of using decimal year in x-axis labels, it will help readability by converting it to month.
>> Figure 14 is redone using your suggestions.

Line 613: could optical remote sensing provide information on the ice onset on rivers?
>> Yes. This was demonstrated by several works cited in the Introduction. The problem of the optical images is the cloudiness.

The authors should add scale bar and north arrow for all the maps presented in the paper.
>>Done

Figures in the paper are of various styles and should be made in a consistent style with consistent places for subfigure labels and the legend
>> Figures were redrawn.

**Minor comments**

Line 33: "erosion of channels and banks": need citation here.
 >> The reference on Ettema ( 2002) was added
Line 38: "catastrophic flooding": need citation here.
>> The reference on Beltaos et al., (2013) was added
Line 75: "state and regime" is there a difference between the two?
>> We mean solid or liquid state (is the water frozen or not), which is different from the regime (temporal variability). The phrase was changed "... used for observation of the water state (solid/liquid) and regime.."
The paper needs some language editing.
>> We carefully edited the language in the new version  of the manuscript.
Line 126: proposes should be "proposed"
>>Done
Line 198: delete "in" before later freezing, and "in the" thinning of the ice cover.

>> Done

Many cases of unnecessary "an" and "the"

(e.g. Line 187: "The Matplotlib Basemap Toolkit"

>>No longer relevant. Other source was used.

; Line 573: delete the "the" before "both approaches";

>> Done

Line 18: for "an" estimation (should be "the");

>>Done

---

## Author Comment (AC2) · 15 Mar 2021

**We appreciate the Referees' valuable remarks and recommendations and carefully addressed them in the new version of the manuscript. Our answers can be found in the Supplement materials.**

**On behalf of all authors,**

**Elena Zakharova**

Our answers below are marked ">>"

General comment
This study analyses the potential of radar altimetry to detect the seasonal freeze and melt events as well as the ice thickness on the Ob River, Russia. The study employs the backscatter (sigma nought) signal from Jason-2 and Jason-3 missions, as well as in situ observations from several gauging stations.

While the idea behind the study is clear and definitely shows a good potential of the radar altimetry backscatter to monitor river ice, the presentation of the methods and results lacks structure and scientific rigour. The quality of figures and captions is in my opinion poor. I also recommend to concise the paper by focusing on fewer aspects and removing redundant parts. I also insist on a professional English proofreading to simplify and smooth the style and improve the clarity. Please check the terminology: **reflect vs scatter**, bias **vs difference vs delay**, **algorithm vs method vs approach**, etc. Please also check the comparatives: you very often use "higher", "more", "less", etc. without mentioning what do you compare it with.
Please find my comments below. There are both specific and more general ones, but I found it easier to comment the parts at their place.

>> The manuscript was significantly edited according to recommendations. The section of Methods was extended. The sections Results and Discussion were restructured. All figures were revised. Several figures were removed or combined after revision of corresponding paragraphs as recommended. Many paragraphs were deleted or shortened. We hope that modifications done allowed for significant improvement of the text clarity.

Introduction:
In my opinion too long and needs concision. Think about subsections and narrow down your story from general introduction to your specific objectives. For example: The importance of river ice monitoring ⏩ remote sensing observations so far ⏩ altimetry principles and observations of fresh ice (1-phenology, 2-ice thickness) ⏩ Jason mission ⏩ knowledge gap ⏩ your objectives.
>> The introduction was modified and shorten.

49-50: I think you show with your own study that the word "excellent" is a little bit of a stretch.
>> the word was deleted

51: please specify: you mean here the temporal resolution and not the frequency of the EM wave, correct?

>> we meant the temporal resolution. The phrase was changed "... for characterization of the river ice at  temporal resolution suitable..."

61: "Active sensors operating in the microwave region are weather independent and provide higher spatial resolution" – higher than what?
Phrase changed " Active sensors operating in the microwave region are weather independent and provide the spatial resolution higher than MODIS and AVHRR instruments."

What about high- and medium-resolution optical sensors? Please include a sentence or two for the consistency.
>> In first sentence of the paragraph we already mentioned the optical sensors of medium resolution (MODIS and AVHRR) with corresponding citations. We did not find studies, which used high resolution (Landsat or SPOT) optical sensors for **monitoring** freshwater ice phenology and thickness. However, we do not exclude that the HR optical images were used as auxiliary information for study some specific ice process (and not for multi-annual monitoring).

63: to my knowledge, "largely" is an exaggeration for such studies.
> The word is deleted

75: please explain what means "water state and regime"?
>> We mean the solid or liquid state (is the water frozen or not), which is different from the regime (temporal variability). The phrase was changed "... used for observation of the water state (solid/liquid) and regime.."

74-79: there are repetitions, please compact
>>  The phrase was deleted.

81, 82: what are the "materials"?
>> The paragraph was deleted as not relevant to the Introduction.

83: please specify "reflected" or "scattered"?
>> The paragraph was deleted as not relevant to the Introduction.

84-101: I propose not to go deep into the SAR studies, as they are not very relevant to your study. Instead, formulate briefly the main principles of the altimetry, how the data are collected, what are the data obtained (backscatter, waveform, what else?)
You can also combine some of the information with the paragraph 54-79.
>> The paragraph was deleted.

97: Higher than what?

>> The paragraph was deleted as not relevant to the Introduction.

98-99: is the penetration depth important here considering that the typical ice thickness < 2 m? Would C-band have insufficient penetration? I would focus on the properties of Jason mission later when you describe it for your study.
>> The paragraph was deleted as not relevant to the Introduction.

103: you haven't mentioned before, that the altimetry missions are accompanied by radiometers
>> The paragraph was deleted as not relevant to the Introduction.

103-112: is it only about resolution or the different nature of measurements plays a role too?
>> The paragraph was deleted as not relevant to the Introduction.

113-117: is it already a description of the method you are using?
>> The paragraph was moved  as not relevant to the Introduction.

118-132: this reads rather specific and too long. Please consider revising, shortening or moving to the Methods section
>>  The text was moved and reduced to three lines.

132-142: this part is a presentation of your own results and should not appear in the Introduction
>> The paragraph was deleted as not relevant to the Introduction

143-156: this is a place for your objectives, please state them instead of listing the sections
>>The objectives are formulated and the listing was deleted.
151: please introduce and explain what is a virtual station before
>> The text was re-written as recommended above. The definition is done in the section 2.2.2 **Altimetry**

2. Regional setup
There is not even one reference to all the information you provide for your study area. Please provide them.
>> The references are provided. The information about socio-economical situation in the region is taken from personal observations and discussions with local population.

Please work on the study area figure. Make it colorful. There is no way to understand where is the study area on the globe, especially in black and white version. Is it topography or is it an optical imagery? The scale of the globe is too coarse and of the zoom-in is too fine. The zoom-in area lacks geographical coordinates and the tracks are strangely projected. The circles (gauging stations) are invisible. Please think about the reader who has no idea about your research.

>>Figure 1 is completely redone.

173: please show these cities on the map.

>>Done

175-205: If this is part of your results, please move it to the corresponding section. You also introduce your in situ data only later.
>> The text was removed and put after the in situ data description.

195-199: please show and explain the trends, I do not see any.
>> As we mentioned in the first sentence of the paragraph, we did not find a significant trend for all period of observations on gauging stations. Only during last several years some

tendencies could be noted. We deleted the phrase about the tendencies for last years as doubtful.

Figure 2: please improve the quality of the figure. Is that mean on figure a)? why only three stations of five are used?

>>Figure 2 and its caption are redone.

3.1 Altimetry
Please provide more information on the data selection: tracks, cycles, virtual stations, repeat overpass etc.
>> We added information on sampling frequency, distance between measurements etc.

231-233: please provide more information on how did you extracted the data via portal (programs used, codes?) and how did you use Landsat images for the selection. Is it a manual / visual selection?
>> Details are added: " For this study, the satellite measurements were extracted from the geophysical research data records product (GDR) distributed by AVISO+ data portal (avisoftp.cnes.fr). The high-resolution optical Landsat 8 images (https://earthexplorer.usgs.gov) were used for geographical selection of the measurements over the river channel using own Python code allowing the along-track Jason measurements and Landsat image overlapping. The cross-section of an altimetric track with a water body is called the virtual station (VS). The virtual station receives the name containing the track number."

What exactly are the track numbers in relation to cycle and pass numbers?
>> We added phrase " One satellite cycle consists of 127 revolutions and respectively numbered 254 tracks,  odd numbers for ascending  and even numbers for descending orbits.

3.2 Optical imagery
235-243: I do not think you need to describe the optical imagery in a separate data section. The first purpose you already mentioned in the previous section. The second purpose is not represented later in the paper substantial enough. You can describe the images in place.
>> As no more optical  images are used for illustration of spatial heterogeneity of the freezing/melting processes, the description was deleted as recommended.

I think the radiometric measurements deserve to be described in a separate subsection.
>> The measurements from the radiometer were used only as auxiliary information. We tried to avoid completely their usage thinking about extension of our algorithm to other altimetric missions. Between the altimetric missions exists one, which is not equipped by the radiometer (Cryosat 2). However, in the complex cases (we suppose when freezing is long or oxbow lakes contaminate the radar echo), the radiometric measurements are useful. That why we did not enter into detailed description of the radiometer and added only description relevant to our study : band and sampling frequency, footprint size and how altimetric measurements were joint with the radiometric. Please note, that in the altimetric practice , there are two standard ways to joint 20Hz and 1Hz measurements : 1) spatial interpolation of 1Hz measurements, and 2) use of closest 1Hz measurement. In all our previous publications and here as well, we used the first way. Relevant phrase was added.

Please explain what do you refer to as an altimetric measurement, i.e. you talk about a precise selection over the river channel but at the same time mention multiple times that the signal is

contaminated by the surrounding land. I encountered later the 400 m band oscillation – please refer to it early enough.

>> The corresponding phrases are added. " The altimetric radar aboard Jason-2 provided measurements at Ku (13.6 GHz) and C (5.3 GHz) bands with 20Hz sampling frequency allowing for 375 m distance between adjacent measurements. The ground track repeatability of the mission is kept within ±1 km cross-track at the equator. At the latitudes of our study region (63-66°N), the cross-track oscillation band is about 400 m. "

3.3 In situ data
254-255: please clarify what is measured daily and what is measured 3-6 times per month.

>> the phrase is rewritten " The standard protocol of river ice monitoring includes 1) daily visual observations of ice presence/absence and ice type; and 2) 3-6 times per months measurements of ice thickness and on-ice snow depth."

256: do they drill a new hole every time for a new measurement?
>> Yes, the protocol insists on this.

How do you credit the data from gauging stations? Are they publicly available?
>> The data are provided by Hydrometeorological Service and available after request.

4. Methods
4.1 This does not look like methods to me. You describe the environmental processes and the response of altimetric backscatter on them. Create a separate (sub-)section and provide references to all the processes you describe.

>> the new Section was created " **River Ice Processes and Behaviour of Radar Altimetry Signal Over Rivers**"

Exclude your own results and move them to the Results section.

>> This is not our result. This is a description of behaviour of the altimetric backscatter in relation to ice formation/break up mechanisms. We consider  that  this information  is important for understanding of proposed algorithms. Moreover, this is first article dedicated to river ice processes in relation to altimetric measurements and we want to provide the details, which will be useful not only for glaciological community, but also for scientists working on improvement of altimetric water level algorithms for the Arctic rivers.

I might disagree that the presence of ice increases the specularity. The ice-water interface should be rough, shouldn't it?

>> This section serves exactly to introduce the process of formation of river ice (which differs in some way from that in lakes and seas). However, here we speak about first ice.The first ice is usually thin and wet with smooth bottom (nilas type). This ice type provides very specular returns for nadir radars (many publications from sea ice altimetric community address this fact ).  An important ice bottom roughness during the freezing can be found in areas of ice bridging (floes accumulation). However, floating ice in this areas can calm the water and reduces the water surface roughness (we speak about this in the text). In mixed "rough ice/water" radar footprint, the main signal coming from the calmed specular water will mask the weak return from rough bridging ice. We address this question in the Discussion section. The thick river ice may be rough, we agree. But the effect of the water current, which can smooth the ice bottom in scales of radar wave length, can't be excluded.

Mention the difference between the nadir-looking altimeter and side-looking SAR instruments (i.e. low and high backscatter from rough and smooth surfaces).
>> The phrase is added " In opposite to slant-looking SAR instruments, for nadir-looking altimetric radar the smooth surface produces higher return echo than the rough one."

Figure 3: This figure would be more informative if you show a mean backscatter with the range of values across all the stations for one typical year. Then you would guide the reader through the seasonal evolution of backscatter and attribute the backscatter change to environmental processes. Adding the in situ dates of freeze up and melt begin/end onto figure would help. The plot in its current version does not add any value in my opinion.
>> We highlighted the ice period on the figure. However, we preferred to keep the figure in the current form of  the time series. In this form the figure demonstrates that 1) the seasonal variability is common for northern and southern reaches, 2) there is an interannual and spatial variability in the Sig0 in the beginning of the freezing (why we used CumSum(dSig0/dt) and not the absolute Sig0), 3) there are intermediate winter peaks (why we applied a smoothing filter), 4) during certain years, the detection of the first winter peak is not a trivial task (two adjacent high peaks separated by important Sig0 dropdown), etc. We discuss all these issues in different sections of the manuscript and provide now the reference to the figure 3.

304-305: I am not sure that wet snow leads to an increase of the backscatter. To my understanding, wet snow attenuates the signal stronger than the dry snow. Please provide a proof.
>> Yes, the volumetric scattering increases as the water content of snow increases for slant-looking instruments (even at low angles). However, for nadir-looking instruments , the situation differs. The behaviour of backscatter for snow covered surface in nadir angle is presented  in Ulaby et al.  "Snow cover Influence on Backscattering from Terrain , IEEE, 1984" (we based on his fig. 11). Moreover, the Sig0 increase of order of several dB can be seen on fig 3 of our manuscript during several years before and after the spring Sig0 drop (due to ice metamorphism described also in the text). The proofs for the increase of backscattering during episodical melting can be found  in Papa et al., 2001 and in Slatter et al.2019 (see list of manuscript references). We suppose that the increasing water content on the snow surface (due to melting or rain) decreases the signal penetration depth (Ulaby, 1984, Slatter, 2019) and, consequently, the volumetric scattering. Moreover, **at nadir angle the surface reflection increases** as the water liquid content  increases. The total effect is the increase  in the backscatter.

312-313: the sentence is not clear.
>> The sentence was deleted.

314-315: I do not agree that the waves are increasing with the water level. Should depend on the wind?
>> the sentence is changed for : " As the water becomes free of ice , the backscatter decreases due to increased surface roughness induced by wind."

319: Provide more specific title for the section
>> The title is changed on " **Ice onset and break-up from altimetric measurements**"

Was this algorithm ever applied already or it is brand new?
>> This is a new algorithm

320: referring to an example figure as described above would help to follow.
>> The reference was added

320: not sure about the word "annual" in this context
>> Changed on "last high peak of each year"

320: how do you define a peak?
>> We wrote a simple code looking for peaks in time series for the automated routine and do it visually analysing year by year for the manual routine.

321-322: "In the case of a multi-peaky recession limb, this peak should be of order of spring and summer peaks". Why? What is spring and summer peaks?
>>Spring peak is the peak after the ice melt (see fig 3). The summer peaks are also frequent and occur when the water surface roughness is low (nadir looking instrument) due to the calm weather. As we insisted many times in the manuscript, depending on configuration of the virtual station, the land part in the footprint is different. We were not able to setup a definitive Sig0 threshold to select between several autumnal peaks. So, we elaborated several assumption helping us to define the freezing moment. One of them is the peak magnitude, which has to be of order of summer peaks at a given virtual station. The magnitude of summer peaks can differ from VS to VS.

322: what is recession limb?
>>we introduced this word in the text when describing the Sig0 time series in the figure 3. " This peak is followed by a progressive winter decrease, which forms a recession limb on backscatter time series." The "recession limb" term is used widely in river hydrology when describing the water level time series, meaning the progressive decrease of parameter.

322: "If the selection of peak is not straight forward..." – how does the algorithm select a peak?
>> we explain this in the next phrase: '... an additional criterion based on the brightness temperature difference (dTb) between 34.0 and 18.7 GHz frequencies is introduced."

324: Explain why and how do use the dTb and add dTb on the figure described above.
>> We explain this in next phrase: " ...for example, two high peaks within one month or prominence of peak is low..." .  How do we use the dTb is explained in the reply for the Rem.326

326: why exactly t-1, t+2? Why 2K? Can you refer to previous studies or explain this choice?
>> This is a part of our algorithm design. This have not been published before. The criterion was setup experimentally basing on verification of results : the part of successful retrievals comparing to in situ observations  (described in the Results Section). The window [t-1, t+2] means that 1 satelite cycle before (==10 days)  the freezing has started on the land (already mentioned fact) and that this freezing progresses further for next 20 days (t+2cycles). This allows us for rejection the early cold episodes or for handling  the cases of long freezing (alternation of cold and warm episodes) .

Please provide a conceptual scheme for the automated algorithm.
>>  The conceptual scheme is similar to that of manual and described already in the text .

 if you do not use this approach in the end, I do not see a reason to include it in the paper. Or "guided over river ice by the main waveform peak" means that you use this information? Please clarify.

>> We prefer to keep the phrase, as this behaviour of the waveforms has never been shown before **for river ice**. However, we explain, why we did not use this approach. The fig 4a aimed to demonstrate that our statistical approach has a physical background.  And we return to the figure in the section dedicated to the potential improvement of the Hice algorithm. However, following the Referee recommendation, we deleted the figures 5b and c as they are not discussed in the text.

You never refer to the Fig. 5b. The information on a) and b) seems to me redundant. Is a) just one example? Why did you choose only two virtual stations on b)? Why not to include all gauging stations? Improve the quality of the figures, avoid work-in-progress axis titles and legend entries.

>> The figure 5b was deleted

388: how did you create a virtual station?

>> We explain this earlier in the sec. 2.2.2 " The cross-section of an altimetric track with a water body is called the virtual station (VS)." Similar to the in situ gauging station, the virtual station concept was introduced in scientific literature in $1990^{th}$ for inland water studies using altimetry, see for example [Birkett et al., 1998].

Did you use the same stations for the dates of ice freeze-up and ice break extraction?

>> Yes, the same stations.

390: what is "ice thickness relations"?

>> We meant the "ice thickness-backscatter relations". The phrase was edited.

391: by "extrapolated' you mean "applied"?

>> "extrapolated' was changed for  "applied"

391: what is "other "main set""? the rest 40 stations?

>> We meant stations not used for calibration. Phrase changed for "The established relations were applied to other 38 virtual stations..."

393: 2) not clear what do you correlate with what

>>The text was re-written " For each virtual station ($VS_i$) we used the coefficients a and b of those virtual station from the training set ($VS_{jt}$), which expressed the best correlation between $Sig0_i$ and $Sig0_{jt}$."

395: the scheme needs to explain better what are you doing. What do white and grey color mean? What are the grey outlined circles?

Figure 6 is redone.

Results:
Maybe it would make sense to make subsections and give them titles. It is not easy to understand when the topic is suddenly changing. For example, algorithm evaluation, interannual variability, spatial variability, etc…

>> The subsections were made

400-403: this paragraph seems redundant to what you already described in the Methods.

5.1 Ice phenology
Please start with an accurate description of your results.
>> Done. The introduction phrase to the section was deleted.

405-408: this seems to me a repetition of your methods section, at the same time rather a hypothesis of what you think is captured by the altimetric measurements.
407: would open river water always appear rougher than the young ice? Even if the flow is calm and there is no wind?
408: decrease compared to open water?
>> The part of the text referred to rem.405-408 was removed.

411: I think "bias" is not correct term for what you describe since bias has a direction. Difference would do better?
>> Changed for "difference"

411: "accuracy for Jason" – not for the altimeter but for your algorithm, for the date extraction?
>> Changed for  "... altimetry-derived ice phenology dates."

411: how many retrievals are there? 8 stations by 12 years? 48 stations?
>> 10 VS from the training set  by 10 winters (2008-2018).  Only VS near 5 gauging stations are selected for verification.

413: what is close to zero and if it is not exact how did you end up with 56%?
>> Changed for "is equal to zero".

417: less accurate than what?
>> Added "... than using manual routine"

491: what results and what do you refer to here?
>> The phrase was deleted

424: better than what? Do you mean the algorithm or manual approach?
>> The phrase was changed for " ... manual implementation of our algorithm detects well  the start of ice thermal degradation"

426-428: what means "least" here and what do 54% and 67% refer to?
>> The phrase was changed " The automatically derived melt date estimations demonstrate worth accuracy for detection of the melt start, comparing to the manually derived estimations. Only in 54 % of the cases the difference with in situ melt start observations is ±10 days."

430: what do you mean by outliers? Please explain. Is it possible to adapt the algorithm? What about radiometric measurements? Do they not help? Can the manual peak extraction be affected if the person knows in advance the true date of freeze up and melt from the in situ measurements?
>> "outliers" changed on "unrealistic retrievals". Yes, it is possible to improve the algorithm and we speak about it in the discussion.  The radiometric measurements helped significantly, but

unfortunately not in 100% of the cases. Probably, a better radiometric criteria can be developed. The algorithm improvement is the subject of our future studies.

Yes, of course, if one look firstly on the in situ measurements, it will be difficult to exclude the subjectivism in the retrievals. However, we did the manual date retrieving without using the gauging data.

433: melting and freezing before 10 April and after 10 June? How did you define these dates?
>> This is expert knowledge. We have being working in the region since 2004 and know quite well the hydrological situation. We looked at our previous publications dedicate to the altimetry and the Ob River discharge and added 3 weeks for earliest (considering potential climate change or extreme years) and 2 weeks for latest water level rise start. We did not speak about this as 1)we considered this is not an important information and 2) the manuscript was already long.

Figure 7: please use the total number instead of norm pdf. Please mark the 90% cases on the graph.
>> Figure 7 presents the histograms normalised on total number of observations and not cumulative probability function, e.g. we cannot show 90% of cases. One can obtain all percentage that we discuss in the text by selecting the range (for example +-10 days) and summarising the percentage taken from the y axis for -10d, 0d, +10d.

How do you explain
• Bimodality and bias towards earlier freeze up of the automatic algorithm for freeze up?
>> In the automatic implementation some early Sig0 high peaks are detected as the ice onset and produce the second mode on the plot 7a. Probably, these peaks are followed by the high peak 2 cycles later. When analysing these cases visually, the progressive decreasing of the Sig0 after the 2nd peak argues for its selection. So, the choice of the second peak is evident for manual retrieving. However, we could not tune the automatic routing for handling this situation. We did several tests, but lost efficiency in the cases of multi-peaky recession Sig0 limb (see for example fig. 3, VS 12, winters 2018, 2019). We hope to improve the automatic routine in the future.

Do I understand correctly that negative values correspond to the earlier date?
>> Yes.

• Bias towards the later melt start for both automatic and manual approach?
>> Sorry, but we do not observe the bias toward the late melt start for manual approach on the fig 7b. It is true for the automated routine. As we explained in the text, with 10 days Jason repeat cycle we conceder +-10 days as acceptable accuracy for the dates retrievals. And as we mentioned in the text, the automatic routine is better for the melt end detection (which occurs ~2-4 weeks later). So, it is logically to see the second high mode toward later detection on the fig 7b.

• Poor results for the melt end for both approaches, especially for the manual approach?
>> The melt algorithm was designed to detect the melt start. So, this is logically true, that manual routine is not suitable for the melt end (fig.7c). The visual selection of the Sig0 rise start on the Sig0 time series plot is straightforward in most cases. However, to code all possible situations that can occur in the nature (small intermediate peaks at the end of winter, small peaks in the rising Sig0 limb etc.) was not possible. As a result, the automated routine only

partly handled the melt start (fig 7b). The other part of detected automatically melt dates corresponds to the melt end dates observed on the gauging stations (fig 7c). This is probably due to these intermediate peaks during the Sig0 rise. Obviously, the radiometric dTb criterion is less effective during the spring than during the autumn. Its adjustment is the matter of our future studies.

436: as mentioned, this is generally not a bias.
>> Changed for "difference"

437: why 11 stations? You mentioned 48 in the text. Why not to include all of them?
>> The verification of ice phenology and ice thickness algorithms is done for the virtual stations from training set. As we explained earlier, we selected those located north and south from the 5 gauging stations + one additional situated in several km from one from the training set. Now we keep only 10 VSs in the training set to facilitate the understanding.

438: I thought that training set applies to ice thickness retrieval? Why do you need a training set for the ice dates?
>> We do not need a training set for phenology algorithm. We did a verification of the algorithm only for VS from this set as these stations are located close to gauging stations. Doing the verification of the dates on the training set allowed us to avoid potential uncertainties related 1) to difference in the freezing/melting dates between main and secondary branches (now this section is removed as recommended); 2) to effect of longitudinal gradient, which can be observed in the altimetric dates for certain years (now this part is also removed to shorten the manuscript).

439: how do you evaluate that? How do you define what is a good sensitivity? I think you need to find a rigid approach to evaluate your algorithm performance. Maybe you do not even need to go on with the automatic algorithm in describing multiyear variability and spatial variability? You can choose manual and show the results only for it as it is clearly better.
>> The paragraph was re-written.

440: not the altimeter but your algorithm?
>> We agree. The text was modified.

441: what do you mean by "noisy"? Noisier than what?
>> The paragraph was re-written.

442: "Nevertheless, a clear coherence exists between the corresponding time series". Please clarify this sentence. What do you mean by coherence, what are the corresponding time series, and why is it clear. Not clear to me.
>> The paragraph was re-written.

439-445. Please find more consistent and smooth way to describe your results, than by picking some years and saying that some events are noticeable.
Why on Fig. 9 you present the different events (melt start and melt end) AND different approaches (manual and automatic)? Please use the same approach for consistency.
446: why important?
>> The paragraph was re-written and the figure 9 was removed as the text was shortened.

447: Why Salekhard station is not included from the very beginning? "Adding" to where? Describe this gradient, i.e. earlier freeze up at the northern stations? If you talk about trends, provide the parameters, i.e. days per degree (or km).
448: average what? Why "calculated"? Are you talking about the algorithm?
449-451: either show what you mean or remove. Very difficult to follow. What years, what half?
452: the same comment as before, please describe these gradients more detailed.
452-453: what do you mean by "gradient in the order of 20 days"? What is denominator of the gradient? Km?
454: the same here.
>> The paragraph was removed to shorten and simplify the text. All above remarks are no longer relevant.

Figure 8: why do you use median for the altimetric data and mean for the gauging stations?

Please use legend to display the information in the caption, it is very difficult to follow.

>> We were not specific enough – mean did not mean "arithmetic mean". Changed to "median". Legend added to the figure

458: does it mean 4 gauging stations average? Why 20 virtual stations? How did you decide which ones to include?
>> The details are now provided. The median is compared to median. Only four gauging stations located on the main river branch are selected, as the secondary branches can demonstrate some difference in ice phenology dates. Only 20 VS located on the main river branch were selected for consistency with in situ observations.

Figure 9: as already mentioned, make it consistent with the approaches. Comments for Fug. 8 applies here as well.

>>Fig 8 and 9 were simplified and combined

Figure 10: what is the x-axis? Please use more reader-friendly labels. Use figure titles in addition to a,b,c

>>Fig 10 is now deleted as no more text related to this figure exists.

Why do you show manual approach for a and b and the automatic one for c?
Please be consistent. Please indicate stations on the red line. Use legend to explain the lines and shaded areas.
474: what do you mean by "different years"?
>>The remarks are no more relevant. The text was modified.

479-500: this whole section is a little hard to follow. How did you select the shown years? How certain you can be with the algorithm results? What about in situ data? I am not sure that the way of presenting those results as in Fig. 11 is optimal. Would it be possible to use mean values for the entire period of observations instead of only three years?

486-487: "This is shown clearly on a Sentinel-2 optical image…" I do not see anything clearly on the Sentinel image. Please use color version, and mark on the image what is what (land, ice, water, main channel, narrow channels, etc). The same comment applies to the Fig. 12b.
489: branches, not brunches
489-491: Please show the image. If there is a polynya until March, does it mean it is not freezing at all during some years?
492: "between the branches" – do you mean main and secondary branches?
494-496: "At the beginning of ice degradation local morphological controls only play a small role (Figure 11b). Their role amplifies during mechanical break-up, which is better captured by our automated algorithm (Figure 11c).". How do Figures 11b and 11c illustrate both statements?
497: uncertainty in what?
497: higher that what?
500: please explain Fig. 12b in more details and support the statement.
>> The whole section dedicated to main/secondary brunches phenology comparison was removed from the new version. This allowed us to shorten significantly the manuscript.

5.2 Ice thickness
Would it make sense to show the Fig. 5b here? To support the equation 1?
>> We moved the equation to the Method section. So there is no needs to move the Fig 5b

501: what means different gauging stations? All of them? Some of them?
>> " Coefficients a and b of the equations were estimated for each gauging - virtual station pair from training set. Using leave-one-year-out method for each pair we obtained a set of a and b coefficients. The average values from each set were used for ice thickness estimation for a given VS. The accuracy of the ice thickness retrievals was evaluated using correlation coefficient and Root Mean Square Error calculated between retrieved and observed Hice for all 2008-2018 period."

501-503: how do you come up with 9 runs? Please explain the approach more detailed.
>> For 10 winters of 2008-2018, the leave-one-year-out test gives 10 runs. The text was corrected

How to read Table 2? How did you sort rows in the Table 2? Why there are two Pitlar, two Gorki, and two Muzhi entries?
>> The Table 2 was reshaped. The rows are now sorted by the virtual station. The additional (11th) station was removed for simplification. As it can be seen on the fig 1 , one in situ station is surrounded by two virtual stations, that is why each gauging station name is provided 2 times.

Why did you decide to work with an individual relationship for each station and not with one universal relationship?
>> Our previous study dedicated to the lake ice (the paper in preparation) demonstrated that Sig0 - Hice relations vary from lake to lake. We expected the same for the river ice. We introduced some modification to the lake algorithm and obtained quite interesting results - except for 1 case, the coefficient a and b have low variation. We discussed this fact in the manuscript. Further investigations are needed to investigate this issue.

How would Fig. 5b look if you include all the gauging stations?

>>The fig 5b will look messy  with all gauging stations.

What do the correlation coefficients and RMSE describe? The relationship between backscatter and ice thickness? Then RMSE should be not in m but m per dB? Why correlation coefficient and not coefficient of determination, especially considering that the relationship is not linear? Or R and RMSE describe the relationship between in situ and altimetry ice thicknesses?
>> The text was modified :" The accuracy of the ice thickness retrievals was evaluated using correlation coefficient and Root Mean Square Error calculated between retrieved and observed Hice for all 2008-2018 period." In this case the RMSE is in meter units.

509-510: How do I see on the Fig. 13 that those stations are northern? I'm confused here, I thought that the stations 138 and 161 are used in the training set? Please include all the stations in the Fig. 13 and use gradient color scale for the timing of measurements. The same for the spatial distribution of the stations – maybe different point style?
>> Here should be the reference to the Table 2. The text is edited and significant modifications were introduced. The VS numbers for northern and southern stations was added into the text. We preferred to not colour the timing, as we suppose that, logically, low ice thickness occurs in the beginning of ice period and high ice thickness occurs in its end.

bigger in size and better resolved, use different line colors and styles (if you keep individual years). In the left part of the graphs, especially on c) nothing is visible. Explain x-axis, what is Tr187? Give the direction of the north. An additional map with the location of the shown stations would be useful here.
>> The figure 11 was deleted from the new version of the manuscript

Fig.12. Please see the comments before. What band do you use or is it a color composite? Also please include coordinates, scale. Please put the labels on the in situ and virtual stations. What is the difference between open circles and squares for the virtual stations?
>> The figure 12 was deleted from the new version of the manuscript

540-542: where could I see it myself? In the table 2? Then please organize the Table in a clearer way. Table 2: please organize it in a more understandable way and provide more explanations on how to read it.
>> The Table 2 was reshaped and re-organised and details were added to the corresponding section of the text.

555-608: Please include these sections into the Results. This is clearly a continuation of your results, and not discussion.
>> The section was moved to the Results.

557: please indicate which parameters did you use, there are many different ones.
>> Details are added: "... coefficients a and b of the Equation (1)..."

562: please include a short explanation how did you choose the window size
>> The details were added: "The size of applied window allowed for preserving the magnitudes and spatial heterogeneity of ice thickness in spatial domain, as well as for reducing the residual noise in temporal domain, which is left after smoothing of backscatter time series with Loess filter".  We did not enter into technical details how we adapt our window as it can be found in different corresponding manuals and explanation could result in one more paragraph of

important size. Instead, we provided the criteria, which were important for selection of the window.

Fig. 13: please see comments to this figure above.
>> comments were addressed and reply provided above.

569: please change the word "analog", it is unclear what you mean.
>> The text was modified to avoid this word.

569-576: I think it would make sense to move this paragraph to the beginning of the section, and to describe the ice thickness product after that. Do I understand correctly that you use the second approach to create the ice thickness product?
>> The paragraph was moved

571: why only four stations if you have five? Please mark on Fig. 1 the clusters of virtual stations attributed to the in situ stations based on their proximity.
>> Only 4 gauging stations are located on the main river branch. Here, we discuss estimations of ice thickness only for the main branch of the entire river reach. Corresponding phrase is introduced.

571-573: please reformulate sentence, it is unclear. What is main VS? What is time shift?
>> The text was modified.

573: "The performance of the both approaches was evaluated at 11 virtual stations nearest to the location of the gauging stations". Now I am again confused. These 11 are in the training set? But why?
>> The text was modified and simplified. We provide the results only for one (best) approach and do not speak about other one (which was tested, but not used)

574-575: how can I see the results of the first approach?
>> The text was modified. We deleted information about first tested approach as it was not used and hope this simplified the text.

584: "The interannual variability in maximum ice thickness retrieved from altimetric measurements at many virtual stations indicates a clear decrease from 2008 to 2012." Is this something shown in Fig. 15? Then refer to the figure 15 right there. What means many? Why did you include only 2 of in situ and 2 of virtual stations in the Figure? Could you show a plot similar to the Fig. 8-9,, which would include all of the stations?
>> The reference to the Figure 15a was placed as recommended. The figure 15 presents only 2 examples as the plotting all stations results in massy picture.

592: please explain what is ridging flag
>> the word "flag" was changed for "event"

Figure 14: I suggest to combine the yearly plots into one multiyear plot, and add some vertical lines to mark the timing - to show the interannual variability. Please indicate the north-south direction, add unit on the color scale.
Please describe results shown in Figure 14 in the text. Right now, you only mention that you created the product. What is the area we see in this product? What is the extent of it?

>> Figure 14 is redone using above-mentioned suggestions. However , the combination into one multiyear plot lead to figure becoming unreadable, so we keep the initial layout.

>> We addressed all remarks from the line 600 in the new modified phrase. " The interannual variability of altimetric ice thickness on 1 December differs from those, observed  on gauging stations. However, this difference is not high and lies within algorithm uncertainties 0.07-0.18 m. Besides the geophysical reasons and algorithm simplicity noted above, the degradation in quality of the in situ time series and the low representativeness of the one-hole sampling protocol can be evoked. "

>> The reasons are general for many Arctic areas: not sufficient financial support, shrinkage of the ground network, absence of well trained stuff etc. The word "lower" changed on "low".

>> This section was deleted

>> The agreement in interannual variability of our retrievals of Hice_max  with the observations on the gauging stations (fig 14 and 15a) is the basis for this statement.

>> Yes, first time we noted this fact in the text in relation to the Figure 13 (see low ice thickness points corresponding to the beginning of the freezing). This is also can be visible from the Figure 15 b, where the plots of H_alti lie above the plots of H_insitu.

>> We did not use this approach because of the algorithm has not yet been developed.
In this section we speak about future potential improvement and indicate several possible ways.

>> The sentence was reformulated. " In our algorithm the ice thickness estimation starts from the date of first ice (bank ice or frazil floes) appearance. Usually, the river reach in area of virtual station at this moment is not fully frozen. The detection of the date of the first consolidated ice (e.g. fully frozen reach) could help to reduce Hice estimates in the beginning of freezing."

>> This is new.

Figure 15: please see above the comments to this Figure. Why there are two plots for the b)?

>>Figure is redone, the caption is changed

The Discussion section has only one subsection which actually discusses the results in the context of the physics of the radar signal return and potential errors related to that. Another subsection considers a study case of an ice road and applies the developed methods to this study case, which again reads more like results. I was missing a discussion of your results in the context of the relevant studies (which are not only methodologically relevant).
>> Sorry, but we did not understand what do you mean exactly under " discussion of your results in the context of the relevant studies".  We significantly modified the Discussion section.

I have an impression that the whole section 6.2 would benefit from shortening and compacting. Some of the discussed issues are mere speculations and raise more questions than give answers, and some other points are not really relevant to your study (e.g. layering as you pointed out in 682).
>> The discussion section was restructured and now consist of two parts:
**6.1 Geophysical factors affecting radar altimetry measurements over river ice**

**6.2 Potential improvement of algorithms**

The subsection 6.1 was significantly shortened as recommended.

646: "…grows gradually from January until April." – is there no wind redistribution?
649: Please include the figure 4 here if needed
652-653: please explain better what you mean with the ratios and 40%, and 25%.
>> the section was re-written and the comments taking into consideration rem. 646 - 653

660: you mean the power relationship becomes weaker? What correlation do you mean?
>> Yes, thank you for suggestion how to ameliorate the phrase.

665: "Further congelation of inter-floes volume as well as ice growth lead to leveling of the ice lower boundary.". Can you support this hypothesis? Do your observations of the decreasing backscatter (669-671) contradict this statement?
668: please explain what do you mean by the first two cycles
669: please explain what do you mean by the note "due to the waveform peak power"?
>> The text was edited to avoid the contradiction and shortened  as recommended. The remarks 646 - 669 are no longer relevant.

703: I do not think that we have seen any clear tendencies in the referred section.
705: can changes be robust?
706-707: please provide a reference for this statement
>> The part of the text relevant to Rem.703-707 was removed

708-709: how do you define an outlier? These are the valid observations, right?
>> The word is changed for "... measurements untypical for a given month... " . We detected some strange records in the data  provided by one of the gauging stations.  We could not find any geophysical process that could explain such a seasonal variability of Hice.

712-721: this part sounds like an outlook to me and should belong to the Conclusions section.

>> This part was modified and moved to Conclusion

746: please expand on what you mean by a delay. Delay to what? Is it the best term in this case? Please check it further in the text as well.
>> word "delay" is replaced by "ahead".

746-748: this sentence is hard to follow. Exception from what? Why this is an exception? Where is the Salekhard river reach?
>> The sentence was deleted for simplification.

751, 754: what is circulation?
>> The word was changed for "traffic"

754: earlier than what?
>> The phrase was modified

762: perhaps, "could be adjusted" instead of "has to be adjusted"?
>> Thank you for suggestion. The phrase was modified

760-771: this also sounds like an outlook and can be combined with 712-721
>> This part was modified and moved to Conclusion

Figure 16: You never mention Figure 16b in the text.
>> The reference on the fig 16b was added.

778-780: please reformulate the sentence: what is the second record of melt onset?
>> reformulated, see reply to rem. 780-783:

780: correlations between which dates? Is the information on Fig. 17a similar to the Fig. 16b?
>> No, Fig 16b shows the interannual changes of the predicted date, while the fig 17a shows the relationship between "second earliest melt onset" (see reply on the rem.780-783) and observed dates.

780-783: please explain better what are you doing here. How did you produce a forecast? Is forecast is just the melt onset day derived from altimetry (whatever the second record means)? This is a bit confusing.
>> The sentence was re-written " Using altimetric retrievals of the melt start for entire set of 48 virtual stations for each year we search the second earliest melt date".
The approach is based on the fact that the melting starts in the south of the region and progress to the north. By detecting the melt start in southern reaches, we suppose to be able to predict the melt (and road closure) in the northern reaches.

789-794: again outlook, can be combined with previous ones and moved to Conclusions
>> This part was moved to Conclusion

Figure 17: Please explain what ROI are you referring to? In Fig. b) please correct the label of y-axis – the current label is not clear – what is ice road closing delay?
>> The title is modified.

Conclusions: please use this section not only for a dry summary of the results but also for a more general wrap-up (reinforcing the problem importance, filling the knowledge gap, outlook and recommendations etc).

>> The section was modified and extended.

---

## Referee Report (RR1)

**General comment**

The authors have done substantial revisions and improved the quality of presentation (both text and figures). However, I still have a lot of concerns and suggestions. As for the majority of my questions the authors just removed a lot of text and figures from the manuscript, I feel that we should have another detailed look into what is left.

I provide here the structure of the paper for an overview:

1. Introduction
2. Regional setup and data
   2.1. Study Region
   2.2. Data
      2.2.1. In situ data
      2.2.2. Altimetry data
3. Temporal variability of radar altimetry signal over frozen rivers
   3.1. Backscatter variability
   3.2. Waveform changes
4. Methods
   4.1. Ice onset and break up algorithm
   4.2. Ice thickness algorithm
5. Results
   5.1. Ice phenology algorithm verification
   5.2. Ice thickness retrievals
   5.3. Ice thickness estimation for the entire studied river reach
   5.4. Winter ice bridge roads operation forecast
6. Discussion
   6.1. Factors affecting ice thickness retrievals from altimetry
   6.2. Potential improvement of algorithms
7. Conclusion**s**

In the **Introduction** I still miss a thorough description of the knowledge gap which authors fill with your study. It is clear that river ice needs monitoring, that the remote sensing is the great tool for that, and that authors saw and implemented a good potential of the altimetry backscatter. I would like to see more details on the altimetry principles as well as some review on the existing studies using altimetry for the fresh ice monitoring. From this authors can draw the knowledge gap and the objectives. Also, for example, authors mention some SAR studies on the ice thickness (there are also many on the ice phenology which I think you should mention too) but do not provide any drawbacks of them, i.e. why do we need to use the altimetry at all, if we have SAR?

(See also my comments for the first review).

**Regional setup and data**: I would split the section into two sections, to avoid numbering of the third order (2.2.1 etc).

Figure 2b: I actually see some trend for the maximum ice thickness.

Consider adding similar graphs for four other stations in a supplementary figure.

The whole section **Temporal variability of radar altimetry signal over frozen rivers** is a mix of own results from this study and some discussion of the previous studies. I understand that authors first investigate your data and then build an algorithm based on own findings, previous studies, and

known facts. But I think it should be possible to find a way for rigorous presentation of own results and their discussion with respect to other studies. In general, there are a lot of speculations in this section which are given without any references or proofs.

Here I also would repeat my suggestion to use a typical (or atypical) backscatter cycle over one year as an example, and to illustrate all ice events and corresponding backscatter changes within one year. Authors added color in Figure 3 for the ice cover period, this is already helpful but still it is difficult to follow references to this figure in the text when authors describe the seasonal changes. This also includes Methods. Also, please add the cycle of TB on such graph.

**Results**

I could follow the results until the attempt to validate the 2D product. Do I understand correctly that:

- authors retrieve the ice thickness at all VS using the relationship between backscatter and in situ ice thickness
- then interpolate the ice thickness from the VS to the entire river
- then extract again the ice thickness at some VS (why those?) from the interpolated product
- and then compare it with the in situ ice thickness?

Sounds like a lot of data juggling here, especially considering interpolating and smoothing. I understand the intention to validate the 2D product but I am not sure that it is possible to achieve here. Why not to extract the ice thickness exactly at the location of the gauging stations?

Also, there are some discussion and speculations in this part of the results which should be moved to the Discussion section. In the Discussion section I still miss some discussions on the place of this study in the context of the other river ice studies, be that in situ observations or other remote sensing techniques, other Arctic rivers, or maybe even some connection to the lake ice studies. The Discussion in the form as it is now would fit to a purely methodological paper but this one is a combination of methodology and scientific results. And that is of course then hard to fit into a common paper structure too. I think that after my questions and suggestions, authors would need to make some amendments to the Discussion and Conclusions parts, as well as to the Abstract. Therefore, I leave it for the next round.

Regarding the style, I noticed the following issues:

- choice and mixing of tenses. Past or present, be consistent.
- missing words
- order of the words in sentences
- wrong or missing prepositions
- typos
- spaces
- lower case instead of indices
- dates format is inconsistent
- writing out versus spelling out numbers
- in situ or in-situ, italic or not?

I can see that the professional proofreading was not accomplished and would like to see that done for the next round of the revisions.

Specific comments:

41: icy conditions – do you mean "ice conditions"?

42: "…for people who are required…" – seems to me redundant in the sentence

51: please add that clouds are limiting factors for the optical sensors, not for the river monitoring in general

58: please check the order of the words in the sentence

81: extends approximately… a preposition is missing?

84: reference to the Figure 1 is odd here – move it to a more general description of the study area.

Figure 1: when I proposed to add color to the Figure 1, I mainly meant the overview part from the first version. I think the black and white zoom-in figure looks better. Just add colors to the gauging station symbols, the main cities, and the new overview map to the previous version. Sorry for the confusion.

101: something is missing between "stations" and "water"

113: order of words

114: "installation of ice cover" sounds odd to me

123: ice onset and melt **date**

145-147: you mention studies for the ice thickness retrievals but you use AMR for the ice phenology. Please clarify.

150: I think you can remove "Jason-2 and -3", as you do not introduce Jason-3 at this point.

154: difference (bias) – why do you need the word "bias" in parenthesis here? Again, is it the difference or the bias?

157: can you provide a more recent access date?

159: what is ICE1 algorithm? Any references? Please provide a short explanation how is the backscatter coefficient defined and retrieved?

163: it is not the Python code which overlaps Jason measurements?

166: "…the stations names were extended" – "the names of the stations located on the secondary branch were extended…"

178: "installation of ice cover" – ice does not install, please use another word.

180: what do you mean with the word "intercepts"?

193: please introduce Sig0, see also my comment to the line 159

193: Δt is the time period between two consecutive observations? Please mention.

Figure 3: the vertical lines indicating the new year are not visible. Open water line in the legend is also barely visible, especially when printed.

You decided to show the backscatter time series for these 2 stations based on their location (north and south), correct? Please mention it in the caption.

How did you decide where to start and finish the red line, i.e. the ice cover period? Is it based on the in situ observations? Or these are the results of your algorithm implementation? Please mention.

226: I do not see what do you call an intermediate peak in Figure 4. Please indicate it on the figure.

248: "Freezing on the floodplain and banks" – do you mean freezing of the land surface, i.e. soil or sand? Please reformulate.

252: I think it is the other way around – the backscatter increase marks the ice decay.

268: "we used a relative backscatter decrease…" – add for what exactly

278-280: and then what dates did you use for the ice thickness estimation period? The best of automated vs manual?

284: power function, not equation. In the equation, you have Hice_alti in the left part – should it be H in situ? Please use subscript instead of lowercase.

You decided to exclude the scatterplot but I think it is important to show it. Please show all gauging stations (and corresponding virtual stations) with different symbols or colors as well as the power fits. Alternatively, one set of all stations together, with one fit. We can decide later whether to include it or not.

287: "gauging VSs": "-"is missing?

289: why do you use "mean" and "average" interchangeably?

291: "thickness" is missing

321: please explain in the text of the paper why automated algorithm was better for the melt end date retrieval compared to the manual approach.

Figure 6: if you show the histogram, should the y-axis be called pdf? I am not sure, please check and correct if needed. "Freeze up" lost "e" in the title, and also the legend. You can explain M and A in the legend instead of the caption. In general, the quality of this figure is not good enough, please work on it (font size, visibility, etc).

329: If I understood correctly, for the ice phenology you do not train anything - the 10 VS are just for the validation of the retrieved dates.

336-337 and Figure 7: as you show and describe the results of the manual retrieval, why do you refer to the algorithm here? Please also add why you do not provide the graph for the melt end (poor results?).

Do I understand correctly that you show that the manual approach works, in general, better, and for the ice thickness retrievals you use the manually retrieved dates? Please make it clear in the text.

339-344: I think you do not need to explain the legend in the caption, it should be clear enough. The information on what stations are in (4 gauging, 20 virtual) is important. The max-min red lines on b) are not visible.

345-350: I think, we, in general, see the decrease of the accuracy from the north to the south? Interesting! But it may also be a result of the proximity of the VS to the gauging station: Pitlar station is the closest to its VS. Include the distance to the gauging station for each of the VS in the table.

Please include that point into the discussion.

350-352: "for many years and many locations…" – I do not really see that by looking at the figure, except for the station VS12. Please provide then some kind of a quantification of your statement.

361: I do not understand why do you need to refer to the south or north here. You simply do it for all 5 gauging stations, correct?

Table 2. please explain better the content of the table in the caption. I understand now what do you show there but it took me 2 rounds of revisions and very careful repetitive reading. For example, it is really confusing that you provide coefficients for the power fit and R and RMSE for the validation regression – both next to each other.

Figure 8: please be consistent with the used terminology: backscatter measurements or altimetric measurements.

I think it would be nice to arrange the figure vertically from north to south.

386-391:

please include a short explanation how did you choose the window size
>> The details were added: "The size of applied window allowed for preserving the magnitudes and spatial heterogeneity of ice thickness in spatial domain, as well as for reducing the residual noise in temporal domain, which is left after smoothing of backscatter time series with Loess filter". We did not enter into technical details how we adapt our window as it can be found in different corresponding manuals and explanation could result in one more paragraph of important size. Instead, we provided the criteria, which were important for selection of the window.

Thank you for the details. I think that some more details would not hurt here. What are the corresponding manuals, can you cite them? For example, why 40 km window size preserve the spatial heterogeneity of ice thickness? What is changing on the scale of 40 km? Why is it important at all to smooth temporally? Is there a possibility of oversmoothing and affecting results too much by it?

394: starting form this point I am again lost. Which in situ observations are you referring to when speaking about interstation areas?

399-400: how do you derive this information from the Figure 9? What VS are there?

442: please give some reference to the data

445: what means 4 days ahead – simply the difference between predicted and observed dates? Or that one can predict the date 4 days in advance? What would that mean?

447: how do you come up with 4 days of accuracy?

455: mention that the dates are for Salekhard ice road and also add more details in the caption.

457-466: I also have troubles following this. You use the relationship between altimetric melt onset and real ice road closure dates to correct the altimetric date. But then you cannot compare it with the real closure date again, because the two datasets are not independent anymore, can you?

---

## Referee Report (RR2)

**General comment**

The authors have done revisions and improved the quality of presentation (both text and figures). However, several questions and suggestions were answered only partly. Below I provide a new round of comments.

I asked the authors for the professional proofreading already two times but it seems that authors ignore this suggestion. I see that another Referee made the same suggestion and authors responded: We paid the Elsevier publisher Service for English grammar correction for the 2nd manuscript version....It always worked well for our previous publications. .. Not this time... In new version we corrected many typos, missing words, rearranged the sentences. I hope we detected all catchy errors; Few occasional errors (articles and prepositions) could hopefully be corrected by specialists from journal technical team (what was the case with our last article published by EGU publisher).

I find this response unsatisfactory. I find that the manuscript requires both scientific and grammatic proofreading. I suggest the authors find professional proofreading service for the scientific texts – not only for English grammar but also for consistency and scientific rigor.

Some of the questions are answered only partly or not even thoroughly read and comprehended as it seems:

Fig. 11: For a comparison of ice thickness from 2D product and gauging station, why not to extract the ice thickness exactly at the location of the gauging stations?

339-344: I think you do not need to explain the legend in the caption, it should be clear enough. The information on what stations are in (4 gauging, 20 virtual) is important. The max-min red lines on b) are not visible.
Reply: Number of stations used is provided in the figure caption. The line width was increased. The legend was removed.

Why did you remove the legend? The suggestion was to use legend instead of the caption. Please also refer to the journal guideline and rework all of your legends and captions correspondingly:

A legend should clarify all symbols used and should appear in the figure itself, rather than verbal explanations in the captions (e.g. "dashed line" or "open green circles").

Please distinguish clearly throughout the manuscript when you refer to the visual picking of the phenology dates ("manual") or to the automated algorithm. I am not sure that the manual selection can be called an algorithm (for example, line 436).

Sections 5.4 and 5.5 clearly belong to the Results. The suggestion from the other referee:

*2) The method sections should be expanded with a section describing how the results are validated and all the additional analysis performed, which currently is described in the result section.*

It means you should describe HOW the results are validated but not the outcome of the validation.

132: do you mean "on average"?

135: how is the **complete** freeze-over defined?

140: when?

142: this statement contradicts with the paragraph below.

143-146: please check the grammar and syntaxis. Add numbers – how much later and how much thinner.

Figure 2: a) and b) are missing. I like the figure more now. I disagree that adding other stations does not provide any additional information. Please add Kazym Mys as well in b) as you did in your response to the review. I think you do not need labels "Date" and "Years" as it is clear. Please use more regular time intervals on a) – first date of the month, for example?

3.2 Altimetry data – you describe here brightness temperature data as well, so reconsider the title please.

For consistency, I suggest to describe altimetric measurements from both Jason-2 and -3 satellites, and then add the AMR instrument description afterwards. Thus, move 175-178 upwards.

163: "used" instead of "considered"?

165: can you provide a number or range?

171-173: should the reference go to the end of the sentence?

182: would it make more sense to move the sentence about ICE1 to the line 185 before the sentence "The ICE1 retracking algorithm…"

210: please add something like "based on our own interpretation of the altimetric data from this study"

237-…: I think the opposition here is wrong – you should oppose calm - rough water surface, not calm - river, as river surface can well be calm as well.

264: avoid saying "Our studies…", just cite them as any other studies.

268: better refer here to the Fig 3c, as, again, in Fig 3a one cannot see anything in detail.

Figure 3a: Why for the TB the labels are 37 and 18? Shouldn't they be 34 and 18.7?

Figure 3c: very nice figure! A question - you mark melt onset at the same point as you mark ice free period. Should it be corrected?

289: retrieved using manual / visual approach?

296-298: you could refer to the Fig 3c here

299: please explain what is spring-summer peak (maybe in Figure 3c) and why its height would be suitable for the freezing detection?

304: please explain why do use the difference between 34 and 18.7 GHz and the value of 2K (any references?)

The formulae 1 and 2 are not mentioned in the text and look useless to me. You describe this all in the text and you provide Fig.4. What is the operator "length"? By max you probably mean local maximum but then it should be reflected in the formula. dTB and ΔTB are the same? I know that you included the formulae after suggestion of the other referee but it does not look mathematically rigorous to me.

By the way, why did not you use the relative backscatter decrease/increase for the phenology dates retrieval as well? Is there any reason behind?

317: please give more details – how, why, etc

Fig.4: ΔTB < 2 in [t-1: t+2] – does your formula imply consecutive dates within this interval?

For the break-up ΔTB I am also not sure that the formula is correct.

344-345: including all years of observations?

Figure 5: include in situ station names in the plots title. VS135 is given twice, please check.

377-381: in my opinion this paragraph belongs to the previous section.

406-409: move this part to the Discussion please as this is your interpretation

Figure 7: please rework the Figure that it fits the style of other figures in the manuscript. Increase the font and line width, give titles to the subplots, remove the unnecessarily fine grid, etc.

448, 458: instead of 249 you mean 240?

535-570: In my view, this part lacks consistency. Why don't you provide figure for the ferry operation stop (535-538) as you do for the ice road dates? Why do you need to correct the prediction for the closing date but not for the opening? Do you conclude that the corrected prediction of the road closing is sufficiently accurate for the forecast as opposed to the road opening date? If you correct the opening date prediction, would you be able to state that the forecast is reliable? In any case, I am personally quite skeptical about such correction. For me it would be enough if you showed not corrected dates for the closing prediction as well.

Describe how do you calculate the leading time of the forecast. Provide the information on the difference between predicted and observed dates (RMSE, max-min) and on the leading time in a consistent and systematic manner for all three cases.

541: do you mean Fig. 12a instead of 11a?

541-547: you can also mention that the predicted dates are consistently earlier than the actual dates of the road opening.

543-547: you should move this part into the Discussion

546-547: maybe not so much of interannual variability but an overall trend for an earlier opening of the road demonstrated by both time series?

541, 563: please be consistent – is it also RMSE for the opening date?

**Discussion**

7.1 I suggest to name this subchapter "Factors affecting altimetric backscatter signal" or something like that, as it reflects better what you discuss here.

7.2 Here in the beginning you actually discuss retrievals of the phenology dates and ice thickness. Please consider restructuring/renaming.

You could start with ice phenology dates retrievals (manual and automatic), the factors influencing the accuracy of these retrievals, and potential improvements. Would inclusion of SAR data be

beneficial for the phenology dates retrieval? Are there any SAR-based river ice phenology studies you could compare your results with?

Then you could move to the ice thickness retrieval and do the same. You mention two studies in the section 4 (Unterschulz et al and Mermoz et al) which use SAR data for the river ice thickness retrieval – could you compare you results with those?

I also suggest that you give a separate and clearly distinguished paragraph where you discuss your forecast prototypes and their viability.

616, 619: wrong Figure numbers provided

617-…: again, what about the road **closing** date prediction?

627-630: Do you mean that the detection of the first consolidated ice would require another sensor / data? If it is possible with the same dataset, why did not you try it? Please explain in the text.

Could you provide for this subchapter some information on the availability of the in situ observational stations on the other Arctic rivers? That would be a great outlook on the potential future studies in a large geographical context.

**Conclusions** – I think the title should be in plural

665-666: I would say that a generally low number of in situ observations and their general infeasibility to cover vast areas are the main drivers.

668: freeze-up, breakup **dates**?

669: again, do you refer here to the automated one? Briefly reiterate please the basic principle of the algorithm.

674: please mention that there are 5 stations and 12(?) years of the simultaneous measurements

675: what is it in the percentage of the max/average ice thickness?

681: mean accuracy?

681: please reconsider what you report here based on my earlier suggestions in the Results

683-700: very good!

---

## Author Response (AR2)

We appreciate the Referees' valuable remarks and recommendations and carefully addressed them in the new version of the manuscript. Please find final response attached as supplement.

On behalf of all authors,

Elena Zakharova

**Referee 1**
**General comment**
The authors have done substantial revisions and improved the quality of presentation (both text and figures). However, I still have a lot of concerns and suggestions. As for the majority of my questions the authors just removed a lot of text and figures from the manuscript, I feel that we should have another detailed look into what is left.
I provide here the structure of the paper for an overview:
1. Introduction
2. Regional setup and data
2.1. Study Region
2.2. Data
       2.2.1. In situ data
       2.2.2. Altimetry data
3. Temporal variability of radar altimetry signal over frozen rivers
3.1. Backscatter variability
3.2. Waveform changes

4. Methods
4.1. Ice onset and break up algorithm
4.2. Ice thickness algorithm

5. Results
5.1. Ice phenology algorithm verification
5.2. Ice thickness retrievals
5.3. Ice thickness estimation for the entire studied river reach
5.4. Winter ice bridge roads operation forecast

6. Discussion
6.1. Factors affecting ice thickness retrievals from altimetry
6.2. Potential improvement of algorithms

7. Conclusions

In the **Introduction** I still miss a thorough description of the knowledge gap which authors fill with your study. It is clear that river ice needs monitoring, that the remote sensing is the great tool for that, and that authors saw and implemented a good potential of the altimetry backscatter. I would like to see more details on the altimetry principles as well as some review on the existing studies using altimetry for the fresh ice monitoring. From this authors can draw the knowledge gap and the objectives. Also, for example, authors mention some SAR studies on the ice thickness (there are also many on the ice phenology which I think you should mention too) but do not provide any drawbacks of them, i.e. why do we need to use the altimetry at all, if we have SAR?
(See also my comments for the first review).

Reply. The details on the altimetry principles are added and review of the existing studies dedicated to the altimetry application for freshwater ice is extended. Very few studies used radar altimetry for lake ice. We sited 2 studies applied altimetry for monitoring lake ice phenology (no other studies exists). Unfortunately, **ONLY ONE** study dedicated to the lake ice thickness exists (Beckers et al., 2017). This study used another approach based on height estimation from the altimetric waveform. This method is more difficult to implement for the **narrow** rivers. We cited this study in the Introduction and explained the approach used in the section 4.2. We also mentioned that their method can have a potential issue when applied to narrow rivers as the intermediate peak on the waveform exists, but additional studies are needed to well understand from which surface it comes (floodplain surface, ice/air interface ?) . Several more studies used SAR instruments for river ice phenology dates and ice thickness were added. (Sobiech et al., 2013. Sun and Trevor 2018, Zhang et al., 2019).

**Regional setup and data**: I would split the section into two sections, to avoid numbering of the third order (2.2.1 etc).
Reply: The section was split.
Figure 2b: I actually see some trend for the maximum ice thickness.
Reply: Thank you. The text was modified.
Consider adding similar graphs for four other stations in a supplementary figure.
We already tried to plot all stations, but decided that the look of figure with 4 stations is not nice (see figure below). Moreover, as one can see, the figure with 4 stations does not provide any additional information comparing to the general description of the Ob R. ice regime that is already given in the text. In the text we mentioned main differences between the stations and provided all important dates and values in order to avoid the presenting of the messy plot.
Also, at the southern stations the observations began in 1980ies and have an important gap in 1990ies (see Table1).
Nevertheless, we updated the figure in the manuscript adding 2 more station with long observations without the gaps.

[Figure]

The whole section **Temporal variability of radar altimetry signal over frozen rivers** is a mix of own results from this study and some discussion of the previous studies. I understand that authors first investigate your data and then build an algorithm based on own findings, previous studies, and known facts. But I think it should be possible to find a way for rigorous presentation of own results and their discussion with respect to other studies. In general, there are a lot of speculations in this section which are given without any references or proofs.
Reply: The section was re-written. Many references were added and many general descriptions of low importance related to the ice formation and regime, that can be found in the cited literature were

deleted. Unfortunately, to our knowledge there are no studies dedicated to analysis of altimetric backscatter behavior over frozen rivers. Several studies, done by the authors investigated the variability of the altimetric backscatter over lake ice (Kouraev et al., 2007, 2015). We cited these works. Many statements has been drawn from authors' own research conducted during the period between the cited publications and present work. We hope that the new version of the section is now better structured. We also think that although several statements looks speculative, they could be seen as hypothesis or assumptions, which are proved during validation of proposed algorithms.

Here I also would repeat my suggestion to use a typical (or atypical) backscatter cycle over one year as an example, and to illustrate all ice events and corresponding backscatter changes within one year.

Reply: The plot was added

Authors added color in Figure 3 for the ice cover period, this is already helpful but still it is difficult to follow references to this figure in the text when authors describe the seasonal changes. This also includes Methods. Also, please add the cycle of TB on such graph.

Reply: The TB lines were added

**Results**

I could follow the results until the attempt to validate the 2D product. Do I understand correctly that:

1) authors retrieve the ice thickness at all VS using the relationship between backscatter and in situ ice thickness

2) then interpolate the ice thickness from the VS to the entire river

3) then extract again the ice thickness at some VS (why those?) from the interpolated product

4) and then compare it with the in situ ice thickness?

Sounds like a lot of data juggling here, especially considering interpolating and smoothing. I understand the intention to validate the 2D product but I am not sure that it is possible to achieve here. Why not to extract the ice thickness exactly at the location of the gauging stations?

Reply: Yes, that is right. The Hice extracted from the 2D product was used for case study, namely, for prediction of Salekhard ice road opening/closure dates. And in the paragraph preceding the case (4) we explain why we need to assess the quality of interpolated product exactly for selected parameters : Hice at December 1 (for ice roads) and Hice_max (for climate change monitoring). Of cause, we could stop the processing of satellite-derived data at the level of Hice time series at 48 VSs. However, the idea behind elaboration of the 2D product was a potential application of satellite retrievals for areas between VSs and for other ice roads existing in the study region. Many of these ice roads are not maintained by local authorities and used by local population at one's own risk. The objective was not the validation, but the evaluation of goodness for the specified task.

We modified the text to better present this idea. " The elaborated maps can be used for evaluation of ice thickness and ice phenology dates in areas between virtual stations. For instance, two useful parameters could be extracted from the 2D product: the maximum ice thickness and ice thickness observed on 1 December. From a practical standpoint, knowledge of the maximum river ice thickness is relevant for hydro-climate change monitoring, while the ice thickness determined on 1 December is crucial for local and regional socio-economic stakeholders, as this is the average date for the opening of the ice bridge road to the north of the study area at Salekhard. To assess the goodness of the 2D product for practical use, we compare the interannual dynamics of the mentioned parameters derived from 2D product and observed on gauging stations."

The other Referee suggested to move the sub-sections dedicated to algorithms' verification to the Methods. We did it. We also moved, for consistency, the part of the current sub-section dedicated to the elaboration of the 2D product to the Methods, as this part took some volume after adding details on window selection. So, we tried to combine and address all Referees' requests for restructuring of the manuscript.

Also, there are some discussion and speculations in this part of the results which should be moved to the Discussion section. In the Discussion section I still miss some discussions on the place of this study in the context of the other river ice studies, be that in situ observations or other remote sensing

techniques, other Arctic rivers, or maybe even some connection to the lake ice studies. The Discussion in the form as it is now would fit to a purely methodological paper but this one is a combination of methodology and scientific results. And that is of course then hard to fit into a common paper structure too. I think that after my questions and suggestions, authors would need to make some amendments to the Discussion and Conclusions parts, as well as to the Abstract. Therefore, I leave it for the next round.

Reply: The paragraph with discussion of the source of errors was moved to the Discussion. The discussion on altimetry drawbacks (required by other Referee) was added. A paragraph about potential combination of altimetry with other RS techniques for river ice studies was extended. We hope that we now clearly expressed our opinion that we see the place of altimetry as a one of the element within multi-instrument approaches of river ice monitoring. We can't say more, as the number of studies dedicated to the river ice is really low comparing to other cryospheric thematic. It is too early for the "Road maps" and it is out of scope of this study.

Unfortunately, we could not understand which amendments we should to make to the Conclusion and especially to the Abstract. In these sections we presented only the results of study, which did not change since last review.

Regarding the style, I noticed the following issues:
• choice and mixing of tenses. Past or present, be consistent.

• missing words

• order of the words in sentences

• wrong or missing prepositions

• typos

• spaces

• lower case instead of indices

• dates format is inconsistent

• writing out versus spelling out numbers

• in situ or in-situ, italic or not? Corrected

I can see that the professional proofreading was not accomplished and would like to see that done for the next round of the revisions.

Specific comments:

41: icy conditions – do you mean "ice conditions"?

Reply:  corrected for "ice conditions"

42: "…for people who are required…" – seems to me redundant in the sentence

Reply: changed for   ".. for people who perform..."

51: please add that clouds are limiting factors for the optical sensors, not for the river monitoring in general

Reply:  corrected for "... are limiting factors for monitoring river ice at high latitudes using optical sensors."

58: please check the order of the words in the sentence

Reply: corrected for " Passive microwave and thermal satellite instruments have demonstrated capability for the retrieval of ice thickness for large lakes (Kang et al., 2014; Duguay et al., 2002, 2015; Gunn et al., 2015; Kheyrollah Pour et al., 2017)."

81: extends approximately… a preposition is missing?

Reply: corrected for " The lower reach of the Ob River extends for approximately 800 km..."

84: reference to the Figure 1 is odd here – move it to a more general description of the study area.

Reply: the reference to the Fig.1 was moved to one of the previous sentences.

Figure 1: when I proposed to add color to the Figure 1, I mainly meant the overview part from the first version. I think the black and white zoom-in figure looks better. Just add colors to the gauging station symbols, the main cities, and the new overview map to the previous version. Sorry for the confusion.

Reply: We modified the figure and added colors as it was asked

101: something is missing between "stations" and "water"

Reply: modified for "... all gauging stations **monitoring** water level."

113: order of words

Reply: modified for " According to in-situ observations at the gauging stations, ice formation begins between 23and 27 October. For the last 20 years, the earliest and latest records were 1 October and 18 November, respectively."

114: "installation of ice cover" sounds odd to me

Reply: modified for " …the full freezing can take up to 10 days."

123: ice onset and melt **date**

Reply: the word "date" was introduced.

145-147: you mention studies for the ice thickness retrievals but you use AMR for the ice phenology. Please clarify.

Reply: the sentence was modified: " Brightness temperature measurements acquired with other passive microwave radiometers, such as SSM/I  and AMSR-E, have demonstrated good performance for the retrieval of **ice phenology dates** and ice thickness on large lakes **in Russia** and Canada (**Kouraev et al.,2007**,  Kang et al., 2014)."

150: I think you can remove "Jason-2 and -3", as you do not introduce Jason-3 at this point.

Reply:  phrase was modified

154: difference (bias) – why do you need the word "bias" in parenthesis here? Again, is it the difference or the bias?

Reply: the word " bias" was removed

157: can you provide a more recent access date?

Reply: the link was verified 2021/06/30.

159: what is ICE1 algorithm? Any references? Please provide a short explanation how is the backscatter coefficient defined and retrieved?

 Reply: the citations were provided and the description of how the radar altimeter backscatter is estimated is added.

163: it is not the Python code which overlaps Jason measurements?

Reply: modified for "... Python code allowing the overlapping along-track Jason measurements and Landsat images."

166: "…the stations names were extended" – "the names of the stations located on the secondary branch were extended…"

Reply: modified as suggested.

178: "installation of ice cover" – ice does not install, please use another word.

Reply: modified for "The freezing in river channels starts from the banks..."

180: what do you mean with the word "intercepts"?

Reply: modified for "traps"

193: please introduce Sig0, see also my comment to the line 159

Reply: The Sig0 is introduced in the line 159 (altimetry data description) and explained in the lines 165-170 presenting now the ICE 1 retracking algorithm.

193: Δt is the time period between two consecutive observations? Please mention.

Reply: explanation for Δt was added.

Figure 3: the vertical lines indicating the new year are not visible. Open water line in the legend is also barely visible, especially when printed.

Reply: the figure was modified, the vertical lines were highlighted. All figures are now in 400dpi. Some degradation of quality die to figures re-sizing by the Word can be expected.

You decided to show the backscatter time series for these 2 stations based on their location (north and south), correct? Please mention it in the caption.

Reply: correct, two virtual stations are located in north and in south of study region. Necessary details are added in the caption.

How did you decide where to start and finish the red line, i.e. the ice cover period? Is it based on the in situ observations? Or these are the results of your algorithm implementation? Please mention.

Reply: details were added :" **Data for period of ice cover retrieved from altimetric measurements are shown as thick dark red line."**

226: I do not see what do you call an intermediate peak in Figure 4. Please indicate it on the figure.

Reply: arrows are added.

248: "Freezing on the floodplain and banks" – do you mean freezing of the land surface, i.e. soil or sand? Please reformulate.

Reply: the sentence was modified " Freezing of small oxbow lakes on the floodplain and soils and bogs on the banks usually occurs earlier than in the big channels of the Ob river."

252: I think it is the other way around – the backscatter increase marks the ice decay.

Reply: we slightly modified the sentence: as this is the ice decay which is responsible for the backscatter increase."The beginning of the ice cover decay (thermal melting) leads to the spring backscatter increase. "

268: "we used a relative backscatter decrease…" – add for what exactly

Reply: modified for "... we used a relative backscatter decrease for ice thickness estimation instead of the absolute backscatter values..."

278-280: and then what dates did you use for the ice thickness estimation period? The best of automated vs manual?

Reply: the phrase was modified for "Starting from the first date of freezing (defined using manual algorithm),..."

284: power function, not equation. In the equation, you have Hice_alti in the left part – should it be H in situ? Please use subscript instead of lowercase.

Reply: sorry, here is in situ ice thickness. Thank you pointing this out.. The equation was modified. As the same equation was used for estimation of satellite-derived ice thickness, the general form of Hice was used.

You decided to exclude the scatterplot but I think it is important to show it. Please show all gauging stations (and corresponding virtual stations) with different symbols or colors as well as the power fits. Alternatively, one set of all stations together, with one fit. We can decide later whether to include it or not.

Reply: additional figure showing all VS-GS pairs from the training set with their power fits was added. Please, do not exclude/reshape this figure from the manuscript during the next revision. We deleted the short version of the figure (showed only 2 VS-GS pairs) from the first version of the manuscript to reduce the total length of the text (as it was recommended). But from the new referee comments we see now, that the length is no more a drawback for the article and we can provide all necessary information regardless total manuscript volume. We considered as well that the plotting of points from all VS-GS pairs together is not a good solution. Moreover, the coefficients of the specific fitting curves are presented in the table 2, so it is not necessary to plot/compare these curves on the same subplot.

We consider as well that the fitting of the points by a unique (for all VS) curve (while we hopefully have a chance to obtain a specific coefficients) is not proven. The unique curve will, obviously, lead to deterioration of the results of validation. Probably the "unique curve approach" can be used for extrapolation of the method for other river reaches or for other rivers of the region (north of the Western Siberia) in absence (or limited number) of in situ observation. However, this question takes more investigations.

287: "gauging VSs": "-"is missing?

Reply: Thank you for the correction.

289: why do you use "mean" and "average" interchangeably?

Reply: the sentence was modified for "mean" in both cases.

291: "thickness" is missing

Reply: Thank you for the correction.

321: please explain in the text of the paper why automated algorithm was better for the melt end date retrieval compared to the manual approach.

Reply: the phrase was modified for "**The method was designed for detection of the melt start.** Manual estimation of dates associated with break-up allows for better control of the complex variability of the backscatter during the spring than automated estimation. It is likely that the automatic approach passes over complex cases and detects in these cases the melt end or even provides unrealistic early/late melt dates estimates."

Figure 6: if you show the histogram, should the y-axis be called pdf? I am not sure, please check and correct if needed. "Freeze up" lost "e" in the title, and also the legend. You can explain M and A in the legend instead of the caption. In general, the quality of this figure is not good enough, please work on it (font size, visibility, etc).

Reply: The figure was re-plotted with several corrections. The quality of the figure is 400 dpi. We called the y-axis "Norm pdf" as it presents the normalised on the total number of observations values of pdf. We mentioned this in the figure caption as well. Other reviewer asked us to plot all plots in one line. We did not manage to plot the long words in the legend within the plot limits in readable font size, so in the legend; we kept only the letters.

329: If I understood correctly, for the ice phenology you do not train anything - the 10 VS are just for the validation of the retrieved dates.

Reply: Yes this is true.

336-337 and Figure 7: 1) as you show and describe the results of the manual retrieval, why do you refer to the algorithm here? 2) Please also add why you do not provide the graph for the melt end (poor results?). 3) Do I understand correctly that you show that the manual approach works, in general, better, and for the ice thickness retrievals you use the manually retrieved dates? 4) Please make it clear in the text.

Reply. 1) The reference to algorithm in the figure caption was removed. 2) The new phrase related to the question 321 " **The method was designed for detection of the melt start**" now answers to the current question about the absence of results for melt end detection. We did not aim the development of "open water" or melt end date algorithm, as considered that ice weakening associated with thermal degradation (melt start) is more important information for people safety. However, it is interesting task that we will explore in our future work. 3) Yes, we demonstrated that altimetry-based approach works and our manual retrievals are good. Unfortunately, our automated retrievals are not as good as manual ones and further development (coding) is necessary for their amelioration (when the funding will be available). We consider that now we better explained in the text why. 4) We added the following phrase " As the manual algorithm of ice dates detection demonstrated better accuracy than automated algorithm, it was selected for further analysis of results and for use with the ice thickness retrieval algorithm."

339-344: I think you do not need to explain the legend in the caption, it should be clear enough. The information on what stations are in (4 gauging, 20 virtual) is important. The max-min red lines on b) are not visible.

Reply: Number of stations used is provided in the figure caption. The line width was increased. The legend was removed.

345-350: I think, we, in general, see the decrease of the accuracy from the north to the south? Interesting! But it may also be a result of the proximity of the VS to the gauging station: Pitlar station is the closest to its VS. Include the distance to the gauging station for each of the VS in the table. Please include that point into the discussion.

Reply: The distance to gauging station is now provided in the Table 2. We do really observe some tendency of degradation of validation scores with the distance. However, the correlation of R and RMSE with the distance is low -0.45 and 0.39 respectively and the p-values for these correlations are very high and equal correspondingly to 0.20 and 0.26. Sorry, we do not know how to correctly present these results and even not sure that they have any importance. The main reasons for lower accuracy of Hice retrievals on southern stations, as we think, are given in the manuscript in the current section (we suppose this is the effect of the polynia for VS12) and in the Discussion in the section 6.1 (ice hummocking/ridging in area of VS109).

350-352: "for many years and many locations…" – I do not really see that by looking at the figure, except for the station VS12. Please provide then some kind of a quantification of your statement.

Reply: We meant relative errors. The phrase was modified.

361: I do not understand why do you need to refer to the south or north here. You simply do it for all 5 gauging stations, correct?

Reply: the reference on "south/north" was deleted

Table 2. please explain better the content of the table in the caption. I understand now what do you show there but it took me 2 rounds of revisions and very careful repetitive reading. For example, it is really confusing that you provide coefficients for the power fit and R and RMSE for the validation regression – both next to each other.

Reply: for clarity, the Table 2 was split into 2 tables. Table 2 now presents only the results of validation and the Table 3 presents the scores from cross-validation experiment. The details were also added to the header of the Table 3. In many scientific publications of AGU and Elsevier publishers the scores and different parameters of equations are often provided altogether in one table for compaction or facilitating the results overview. Following this practice, we decided to put the fitting coefficients and the validation scores together in one table.

Figure 8: please be consistent with the used terminology: backscatter measurements or altimetric measurements.

Reply : backscatter measurements was changed for altimetric measurements

I think it would be nice to arrange the figure vertically from north to south.

Reply: the figure was re-arranged vertically with VS pairs from north to south

386-391: please include a short explanation how did you choose the window size

Reply: The details were added: "The size of applied window allowed for preserving the magnitudes and spatial heterogeneity of ice thickness in spatial domain, as well as for reducing the residual noise in temporal domain, which is left after smoothing of backscatter time series with Loess filter". We did not enter into technical details how we adapt our window as it can be found in different corresponding manuals and explanation could result in one more paragraph of important size. Instead, we provided the criteria, which were important for selection of the window.

Thank you for the details. I think that some more details would not hurt here. What are the corresponding manuals, can you cite them? For example, why 40 km window size preserve the spatial heterogeneity of ice thickness? What is changing on the scale of 40 km? Why is it important at all to smooth temporally? Is there a possibility of oversmoothing and affecting results too much by it?

Reply: The manuals and an article providing theoretical background we based on, when selected the smoothing window are below .

- de Smith M J (2015) STATSREF: Statistical Analysis Handbook -A comprehensive handbook of statistical concepts, techniques and software tools . The Winchelsea Press, Winchelsea, UK
- Lotov et al., INTERACTIVE DECISION MAPS, Approximation and Visualization of Pareto Frontier. Applied Optimization Series, Volume 89. SPRINGER SCIENCE+BUSINESS MEDIA, LLC. Ed.: Pardalos P. ISBN 978-1-4613-4690-6, 2004.
- Kamenev G., A Multicriteria Method for Identification and Forecasting. Mathematical Models and Computer Simulations, 2018, Vol. 10, No. 2, pp. 154‑163.

The smoothing was necessary because of irregular temporal satellite sampling (the satellite over-passes different VSs at different dates), gaps in satellite data product and Hice_alti time series, eventual outliers in Hice retrievals etc.  This is a general procedure applied for many spatial altimetric high level (L3 and L4) data products (see for example https://sextant.ifremer.fr/record/bd5a176b-350e-4d5f-8683-da457637bdcb/).   When selecting the window, we compared the Hice extracted from smoothed product  at location of VSs (20 VSs for the Big Ob) with the "unsmoothed" Hice retrieved from satellite. Three criteria were employed for window selection: correlation coefficient, RMSE and difference between maximal Hice retrieved for each year (see Figures A and B bellow).

In space domain (upper panel of Figure A) the correlation coefficient (RN) and RMSE statistics deteriorate when applying the windows in the range 15-40 km. The degradation slows down at windows higher than 40 km. We did not analyse the reason, probably, it is related to the fact that the maximal distance between the VS in the study region is 42 km.  The difference between maximuml Hice increases proportionally  to the size of spatial window throughout whole tested range of windows (5-60 km) and served for control of reducing of Hice seasonal magnitude due to the smoothing.

In temporal domain (low panel of the Figure A), the window size has highest effect on RMSE statistic. In windows lower than 15 days (for 0 km spatial window) and 40 days (for 60 km spatial window)  the changes in RMSE are low (Fig. A b, low panel). For 40 km spatial window, the selection of 30 days window (== smoothing on monthly scale) looks adequate. The decrease in correlation coefficient and in Hmax difference  for 30 days window is low (0.99-0.94 and  0.02-0.05 m for Rcor and RMSE respectively).

Figure B represents 3D view of variation of the statistics regarding the window size. In theory (Lotov 2004, Kamenev, 2018), the optimal solutions (can be multiple)  lies within an cross-over of surfaces.  For this first version of the Hice spatial product  we did not solve the problem analytically as it is described in these publications, as at the present time in the current manuscript,  the spatial product served only for demonstration of potential use of satellite Hice and phenology dates retrievals, for example, for ice road operation dates forecast.  We hope to do this in the future, when corresponding funding for product amelioration and development will be available.

Of cause, there was a risk of oversmoothing.  However, we hope that we avoided this situation as the selected window allowed for keeping the average Hice_max difference and RMSE within uncertainties estimated during validation of the product ( 4 and 3 cm respectively). The correlation between smoothed and "unsmoothed" Hice was 0.99.

[Figure]

a                                                          b

Figure A. Effect of spatial and temporal window size on correlation coefficient, RMSE, and difference in Hice_max between Hice time series extracted from smoothed product and Hice time series retrieved at 20 VSs along the main channel of the Ob (used for gridding and smoothing).

[Figure]

Figure B. Correlation coefficient, RMSE and difference between maximal Hice Normalised on their magnitude(max-min) values. Normalisation allowed for better graphical representation.

394: starting form this point I am again lost. Which in situ observations are you referring to when speaking about interstation areas?

Reply: We meant that there are no any additional in situ observation  (published by other researchers or done by us ) in areas between gauging stations. The phrase was modified " In the absence of validation data for  reaches located far from gauging stations,..."

399-400: how do you derive this information from the Figure 9? What VS are there?

Reply: The color of the figure (varying from blue to yellow as shows  the colorbar) corresponds to the ice thickness. The maximum ice thickness is observed in the end of ice season. The yellow color on the plots (highest values) degrades from 2009 to 2012 to the greenish color and becomes again more yellow starting from 2013 . We think that this is well seen on the figure. We hope that  modification introduced makes the phrase more clear. "The interannual variability in maximum ice thickness retrieved from altimetric measurements  indicates a clear decrease from 2008 to 2012 for 90% of the area of the studied river reach (Figure 9)".

442: please give some reference to the data

Reply: The reference was added."  The information on dates of operation of the ice road was kindly provided by the State Traffic Service of Yamalo-Nenetzky Autonomous District (Russia)."

445: 1) what means 4 days ahead – simply the difference between predicted and observed dates? Or that one can predict the date 4 days in advance? 2) What would that mean?

Reply: . We meant 4 days in advance. The prediction is good four days in advance. Similar to the weather forecast, the longer the prediction interval, the worse the forecast.

2) It is difficult to say what it does mean exactly. The "predictor area" is located in 65-75 km upstream. Probably, the ice formation starts at both reaches at the same time and the Traffic Service considers that the ferry exploitation for next  4 days is still safe. It could be also that in the southern reach the ice appears earlier than for Salekhard city reach due to difference in  morphology (island, sand bank, shallower water etc). It could be other reasons. Unfortunately, the area of the Salekhard ice road is not covered by Jason satellites and we can't say what happens between these two reaches without an additional study with use of other satellites (other altimeters, SARs or optical missions).  An investigation

of capacity of other altimeters takes an additional efforts as the missions covering polar regions (> 66°N) have 27-35 days period. Such a period is not adapted for the operational monitoring. However, during our recent studies dedicated to the Ob R. discharge retrievals from the altimetric measurements (Zakharova et al., 2020), a multi-satellite method of water level time series construction was developed. Probably, similar approach could be applied for construction of multi-satellite backscatter time series.

447: how do you come up with 4 days of accuracy?
Reply: We added explication. " The dates when the Hice reaches 30 cm in location of four northermost VSs were extracted from the spatio-temporal smoothed product. These dates were compared with the dated provided by the State Traffic Service. The mean difference between these dates was four days. " Corrections were introduced.

455: mention that the dates are for Salekhard ice road and also add more details in the caption.
Reply: Details were added.

457-466: I also have troubles following this. You use the relationship between altimetric melt onset and real ice road closure dates to correct the altimetric date. But then you cannot compare it with the real closure date again, because the two datasets are not independent anymore, can you?

Reply: The main information about predictive capacity of the approach is contained in the Fig 12a. From the Fig12 a we see that there are both systematic and random differences between observed dates and AMO2. Moreover, the AMO2 parameter is a predictor, not the forecast. We supposed that we can correct the "predictor dates" on systematic (modelled or known) difference (what we did) and, then, we can evaluate only the residual difference left due to random or unknown errors. The latter gave us 3 days of RMSE. We modified the phrase for: " The residual difference of the forecasted and observed dates evaluated as RMSE is 3 days". Unfortunately, the length of available data was not enough to do full calibration/validation procedure for the forecast method (e.g. develop correction equation from one period and validate it over other period). It can be done in the future with longer time series. Moreover, the aim of this section was the demonstration of capacity and potential application of our results. We do not pretend to develop a robust operational system for the road opening/closure prediction. Development and comparison of different predictive algorithms is another very interesting task for future work.

Please, note as well, that in the forecast two parameters are important : the forecast precision and the forecast leading time (for AMO2 predictor the leading time is given in Fig12b) . We probably could elaborate a predictor that is independent on ground observations (for example, the certain value of ice thickness in specific area/period), however with the AMO2 predictor we could demonstrate one more approach how the product can be used for medium-term/ short-term forecast.

**Referee2**

Summary:

In this study, the authors have developed an algorithm to derive river ice phenology and thickness from satellite altimetry more specifically using the backscatter coefficient. The paper intends to provide complementary methods to estimate ice phenology and thickness as these parameters are important for ice road safety and climate studies. The results were compared to in situ data. Though the algorithm to predict the date of ice onset and melt is premature, the overall result has potential and adds value. Before the paper potentially can be accepted I recommend some changes. Please, see the comments below

General comments:

1) The methodology is only vaguely described, this needs to be improved. In section 4.1 a flow diagram could help to show the algorithm and then adding a more mathematical description would also help. Please, also provide more detail regarding the manual approached which seems to work better.

Reply: The flow diagram for ice phenology and more mathematical description were added.

Due to variety of factors affecting the temporal variability of the backscatter in winter and multi-peaky character of time series (notably in area of known development of winter polynya) we decided to run a manual retrievals to understand, how well we coded the algorithm. The criteria in both routines (manual and automates) were the same. We modified the text to clarify this question.

"...Their complex impact on the backscatter variability during freezing and melting makes it difficult to address all variations in an automated manner. Because of this, we decided to retrieve the ice phenology dates manually using visual analysis of backscatter (and TB if necessary) time series for each VS and to compare the performance of the automatic freeze/melt detection routine with its manual implementation. Both, manual and automated routine used the same criteria. "

In 4.2 It is not clear how it was decided that expression (1) was the best choice. If possible please add a figure that demonstrates how the relationship between the accumulated backscatter and the in situ ice thickness was established and what it looks like.

Reply: The scatterplot of $\Sigma(\Delta Sig0/\Delta t)$ vs Hice insitu is added. Details about fitting procedure were added. " Among tested fitting functions (linear, polynomial and power), the power equation (1) produced the best fit between the cumulative backscatter difference and in-situ ice thickness measurements. The selection of the function was based on maximisation of correlation coefficient between $\Sigma(\Delta Sig0/\Delta t)$ and in situ Hice and minimisations of root mean square error (RMSE) calculated between retrieved and observed Hice. "

2) The method sections should be expanded with a section describing how the results are validated and all the additional analysis performed, which currently is described in the result section.

Reply: We moved the 2 sub-sections dedicated to verification of retrievals from the Results to the Methods as recommended.

3) Currently, the result section is a mixture of results, methods, and an interpretation of the results. Though, a matter of style, this in my opinion makes the paper difficult to read and understand. In this case, it is more difficult to separate the actual result obtained from data, methods, and statistics from the author's interpretation. I, therefore, recommend rearranging this section, so the result section only objectively presents the results.

Reply: We hope that we guessed correctly the difference between results obtained from data and statistics from our interpretation. The 2 verification sub-sections were moved to the Method section. One paragraph with discussion of sources of uncertainties was moved to the Discussion. We were asked by other Referee to provide more methodology about elaboration of 2D product. Now, the volume of corresponding text allowed us to create a full subsection in the Methods dedicated to this part : **"5.3 Elaboration of 2D spatio-temporal ice thickness product".** We hope that the new structure will facilitate the reading.

4) Section 3.2, I do not understand why this section is in the data section, since you do not use the waveform info in the algorithm, or am I missing something. I would not expect that you will find two peaks (related to the ice/air and ice/water surfaces) in a Jason waveform due to the bin distance of 46 cm and an expected ice thickness of approximately 1 m, at least not often. The two peaks are not clearly seen in figure 4. Maybe this section should be moved to a discussion section regarding improving the algorithm.

Reply: The sections 3.1 and 3.2 were re-written as requested by other Referee and present now the background for the methods. The section name (now sec. 4) was changed. The waveform figure was kept here to demonstrate that the use of backscatter has a physical sense. Namely, the statement given in Beckers et al., that the main peak is from ice bottom, is important and the figure illustrates the decrease in amplitude of this peak in winter. This provides the proof that the backscatter decrease is related to the main peak decrease and, consequently, to the ice growth. Concerning the bin distance, we verified the waveform shape evolution over freshwater ice of different thickness on many lakes, and everywhere, we found this intermediate peak in the presence of snow-on-ice layer. Even for Hice < 40 cm the intermediate peak exists in Jason2 measurements ...in waveforms of other altimetric missions... and even in C-band... Actually, several ongoing studies investigate this phenomena and I hope upcoming publications will help to clarify this question.

5) In the discussion section I lack some comments on the strengths and weaknesses of the method and its use. e.g. One limitation is that you need in situ data to establish the thickness relationship, but if the relationship can be applied for another river it will add value.

Reply: Corresponding paragraph was added

6) The paper contains many language errors and should be proofread by a native English speaker or a proofreading service.

Reply: We paid the Elsevier publisher Service for English grammar correction for the 2nd manuscript version....It always worked well for our previous publications. .. Not this time... In new version we corrected many typos, missing words, rearranged the sentences. I hope we detected all catchy errors; Few occasional errors (articles and prepositions) could hopefully be corrected by specialists from journal technical team (what was the case with our last article published by EGU publisher).

Specific comments

Title: consider changing it to "River ice phenology and thickness from satellite altimetry. Potential for climate studies and ice bridge road operation

Reply: Thank you for the suggestion. The title was changed

L 61-62: How does this work, maybe add a sentence.

Reply: the phrase was added." ..., mainly via establishing a statistical relation between backscatter and in situ ice thickness"

L 63: "High resolution" please define.

Reply: values were provided

L 114 "installation of .." -> "formation of "

Reply: modified

L 122-123: Do not write "significant" unless you have performed a test

Reply: Yes we performed the test for all in situ stations. The results are not presented in full form as the manuscript is too big. We added several details in the sentence.

L 171. "In our previous studies (Kouraev et al., 2005; Zakharova et al., 2019, 2020), we noted that over" -> Previous studies (Kouraev et al., 2005; Zakharova et al., 2019, 2020), showed ..."

Reply: Thank you for the suggestion. The text was modified.

L 322 What do you mean by an "acceptable accuracy"?

Reply: we explained why we selected ±10 days as acceptable accuracy in the second sentence of the subsection. " Considering the 10-day repeat overpass of the satellite and the distance between the gauging stations and VS, we considered a 10-day time-step difference (e.g. ±10 days) as an acceptable accuracy for altimetry-derived ice phenology dates." The Jason repeat cycle is 10 days. The algorithm can be perfect with 0 days difference or deviate for some value multiples of 10. We think that 1 step difference from perfect match (==1 cycle==10 days) is acceptable accuracy when using Jason. It is evident, that we can't select criteria between 0 and 10 days, and 20 days bias is unpractical (==bad results) for river ice regime (from geophysical and from practical points of view). Anyway, we provided the percentage of 0 days cases and 10 days cases as for freeze up as for breakup. Thus, one can decide which criterion/accuracy is suitable for a specific task.

L 335 "A significant difference" rephrase if you have not performed a test

Reply: changed for "considerable"

L 350: "Error" is an imprecise formulation please clarify; RMSE, sd, ...

Reply: changed for "RMSE"

+ several languages errors not specified

Figures:

Figure 1: Please add lines to indicates the branches of the Ob River. Color code the VS used as training and test.

Reply: The Ob R. channels were highlighted, the training VS stations were colored in yellow.

Figure 3: Are the red lines based on in situ data? please explain in the figure text.

Reply: plotted Ice period was taken from the satellite retrievals done using manual routine. Details are added to the figure caption.

Figure 6: put the three figures on one line

Reply: Done

---

## Author Response (AR3)

We thoroughly replied for all comments and introduced all modifications.

On behalf of all authors,
Elena Zakharova
contact: zavocado@gmail.com

General comment

The authors have done revisions and improved the quality of presentation (both text and figures). However, several questions and suggestions were answered only partly. Below I provide a new round of comments.

I asked the authors for the professional proofreading already two times but it seems that authors ignore this suggestion. I see that another Referee made the same suggestion and authors responded: We paid the Elsevier publisher Service for English grammar correction for the 2nd manuscript version....It always worked well for our previous publications. .. Not this time... In new version we corrected many typos, missing words, rearranged the sentences. I hope we detected all catchy errors; Few occasional errors (articles and prepositions) could hopefully be corrected by specialists from journal technical team (what was the case with our last article published by EGU publisher).

I find this response unsatisfactory. I find that the manuscript requires both scientific and grammatic proofreading. I suggest the authors find professional proofreading service for the scientific texts – not only for English grammar but also for consistency and scientific rigor.

Reply. The text was sent for professional proofreading. We hope that in this version of the manuscript will satisfy both scientific and grammatic requirements. The corresponding certificate can be provided if necessary.

Some of the questions are answered only partly or not even thoroughly read and comprehended as it seems:

Reply. We thoroughly replied for all comments and introduced all modifications. The problem could arise from misinterpretation of certain remarks (because of their vague formulation) or from difficulty to trace the corrections introduced, as the text was significantly re-structured to match the recommendations of both Referees.

Fig. 11: For a comparison of ice thickness from 2D product and gauging station, why not to extract the ice thickness exactly at the location of the gauging stations?

Reply: The figure was re-plotted with the values extracted for location of the ground stations.

339-344: I think you do not need to explain the legend in the caption, it should be clear enough. The information on what stations are in (4 gauging, 20 virtual) is important. The max-min red lines on b) are not visible. Old Reply: Number of stations used is provided in the figure caption. The line width was increased. The legend was removed.

Why did you remove the legend? The suggestion was to use legend instead of the caption. Please also refer to the journal guideline and rework all of your legends and captions correspondingly:

A legend should clarify all symbols used and should appear in the figure itself, rather than verbal explanations in the captions (e.g. "dashed line" or "open green circles").

Please distinguish clearly throughout the manuscript when you refer to the visual picking of the phenology dates ("manual") or to the automated algorithm. I am not sure that the manual selection can be called an algorithm (for example, line 436).

Reply: The legend was added and the caption was edited correspondingly. The "manual" or "automated" algorithm is now changed on "manual routine" or "automated routine". On the line 436 the word "algorithm" was changed on the word "approach". However, following the definition of the term of "algorithm" ("a finite sequence of well-defined instructions" that we described in the equation 1, which was deleted in the current version after the Referee's request), we consider that the use of the word "algorithm" in relation to the manual retrievals (issuing from manual implementation of the developed sequence of instructions) is possible. Nevertheless, we changed, where it was not critical, the word "algorithm" to the word "approach". The word "algorithm" was kept only for general cases.

On the lines 429-429 we indicated that "As the manual routine demonstrates better accuracy than the automated one, it was therefore selected for further analysis of results and for use with the ice thickness retrieval algorithm." Starting from here, we meant only the retrievals from the manual routine.

Sections 5.4 and 5.5 clearly belong to the Results. The suggestion from the other referee:

2) The method sections should be expanded with a section describing how the results are validated and all the additional analysis performed, which currently is described in the result section.

It means you should describe HOW the results are validated but not the outcome of the validation.

Reply. We interpreted this remark differently. Now, we added the phrase on how the results were validated and moved back the subsections to the Results.

132: do you mean "on average"?

Reply. No. The phrase was changed " Snow depth records represent values calculated as average from three snow-depth measurements located around the hole."

135: how is the complete freeze-over defined?

Reply. The complete freeze-over is defined from the records ( flag corresponding to freeze-over state) of the gauging stations. Details are added : "according to records provided by gauging stations".

140: when?

Reply: the phrase " of the Ob River" was added.

142: this statement contradicts with the paragraph below.

Reply. the phrase " but has not yet resulted in a significant change in the ice regime of the entire lower Ob River" was deleted.

143-146: please check the grammar and syntaxis. Add numbers – how much later and how much thinner.

Reply. Corrections were introduced. Trends were evaluated.

Figure 2: a) and b) are missing. I like the figure more now. I disagree that adding other stations does not provide any additional information. Please add Kazym Mys as well in b) as you did in your response to the review. I think you do not need labels "Date" and "Years" as it is clear. Please use more regular time intervals on a) – first date of the month, for example?

Reply. a) and b) were added. Axis labels are usually mandatory. We checked how other articles published in The Cryosphere present the time series and found that the Years, Time, Dates labels are given in all plots. We would prefer to keep the axis labels. The 4th station was added on the subplot b.

The time interval was changed ( for monthly, 10-days and 20-days intervals). Unfortunately, the use of first date of the month degrades the figure readiness.

3.2 Altimetry data – you describe here brightness temperature data as well, so reconsider the title please. For consistency, I suggest to describe altimetric measurements from both Jason-2 and -3 satellites, and then add the AMR instrument description afterwards. Thus, move 175-178 upwards.

Reply. The Title was modified and the lines were moved upward.

163: "used" instead of "considered"?

Reply. Modified.

165: can you provide a number or range?

Reply. We did not find any publication which provides the size of the Jason-2 radar C-band footprint. The band-C is rarely used. Our information is based on our own unpublished observations of behaviour of C-band waveforms over the lake ice in areas close to lake banks. However, as many other authors we have not been interested in estimation of the Jason-2 C-band radar footprint and do not have the range values that we can publish here. Nevertheless, our statement can be supported by a study of Jiang Ch. et al., A Study of the Technology Used to Distinguish Sea Ice and Seawater on the Haiyang-2A/B (HY-2A/B) Altimeter Data. Remote Sens. 2019, 11, 1490; doi:10.3390/rs11121490. The HY-2A/2B missions are equipped with the Posedon radar instrument those footprint (according to this publication) over the flat surface was 1.9 km and 10 km respectively for Ku-band and C-band. To avoid any mistakes we deleted the phrase under the question from our manuscript.

171-173: should the reference go to the end of the sentence?

Reply. Two publications (Kouraev et al., 2007; and Du et al., 2017) are dedicated only to ice phenology. If we place them all together it will give an impression that the use of AMR for ice thickness retrievals is widely-applied method.

182: would it make more sense to move the sentence about ICE1 to the line 185 before the sentence "The ICE1 retracking algorithm..."

Reply. The phrase was moved.

210: please add something like "based on our own interpretation of the altimetric data from this study"

Reply. The phrase was added.

237-...: I think the opposition here is wrong – you should oppose calm - rough water surface, not calm - river, as river surface can well be calm as well.

Reply. The phrase was modified.

264: avoid saying "Our studies...", just cite them as any other studies.

Reply. The phrase was modified.

268: better refer here to the Fig 3c, as, again, in Fig 3a one cannot see anything in detail.

Reply. The phrase was modified. We slightly improved the resolution of the figure 3a and added the crosses on the top of the intermediate peaks.

Figure 3a: Why for the TB the labels are 37 and 18? Shouldn't they be 34 and 18.7?

Reply. The Figure was modified.

Figure 3c: very nice figure! A question - you mark melt onset at the same point as you mark ice free period. Should it be corrected?

Reply. The Figure was modified.

289: retrieved using manual / visual approach?

Reply. The phrase was added.

296-298: you could refer to the Fig 3c here

Reply. The phrase was added.

299: please explain what is spring-summer peak (maybe in Figure 3c) and why its height would be suitable for the freezing detection?

Reply. "Summer peaks" was added on the Figure 3c. The following phrase was added " This helps to distinguish the peaks related to appearance of first ice (higher peak) from the peaks related to appearance of water on ice (smaller peaks)".

304: please explain why do use the difference between 34 and 18.7 GHz and the value of 2K (any references?)

Reply. References were added and explanation for 2K threshold was provided. " As during the winter the $\Delta TB$ values vary around zero and do not exceed 2K (Figure 3c), we select the backscatter peak at time t, if in a time frame of (t-1, t+2) of satellite cycles at least three of the four $\Delta TB$ values are <2 K. "

The formulae 1 and 2 are not mentioned in the text and look useless to me. You describe this all in the text and you provide Fig.4. What is the operator "length"? By max you probably mean local maximum but then it should be reflected in the formula. dTB and $\Delta TB$ are the same? I know that you included the formulae after suggestion of the other referee but it does not look mathematically rigorous to me.

Reply. We deleted formulas 1 and 2.

By the way, why did not you use the relative backscatter decrease/increase for the phenology dates retrieval as well? Is there any reason behind?

Reply. For sea ice and lakes ice, the relative backscatter decrease/increase approach is quite robust. In more complex case of the river ice, its performance degrades. For ice thickness, the use of relative changes allows for reducing the effect of initial conditions on Sig0 values related to configuration of VS: the higher the portion of land in the footprint the lower the backscatter is in the beginning of freezing. Moreover, the correlations of in situ Hice with relative Sig0 are stronger than the correlations with absolute Sig0.

317: please give more details – how, why, etc

Reply. Details are added.

Fig.4: $\Delta TB$ < 2 in [t-1: t+2] – does your formula imply consecutive dates within this interval?

Reply. No, any combination.

For the break-up $\Delta TB$ I am also not sure that the formula is correct.

Reply. we deleted formulas 1 and 2.

344-345: including all years of observations?

Reply. No, for each year. The clarifications were introduced.

Figure 5: include in situ station names in the plots title. VS135 is given twice, please check.

Reply. Station names were included and the number was corrected.

377-381: in my opinion this paragraph belongs to the previous section.

Reply. The paragraph was moved up.

406-409: move this part to the Discussion please as this is your interpretation.

Reply. The section was moved to the Discussion

Figure 7: please rework the Figure that it fits the style of other figures in the manuscript. Increase the font and line width, give titles to the subplots, remove the unnecessarily fine grid, etc.

Reply. The figure was re-worked according to recommendations.

448, 458: instead of 249 you mean 240?

Reply. Yes, thank you.

535-570: In my view, this part lacks consistency.

1)Why don't you provide figure for the ferry operation stop (535-538) as you do for the ice road dates?

2) Why do you need to correct the prediction for the closing date but not for the opening?

3)Do you conclude that the corrected prediction of the road closing is sufficiently accurate for the forecast as opposed to the road opening date?

Reply. 1) The plot for the ferry closing dates was added to the fig.12.

2) We modified the text and deleted the phrase about "correction".

3) Yes, we can conclude that the road closing prediction is sufficiently accurate. We provided a table containing main statistics allowing the evaluation of the accuracy of the forecasts.

4)  If you correct the opening date prediction, would you be able to state that the forecast is reliable?

In any case, I am personally quite skeptical about such correction. For me it would be enough if you showed not corrected dates for the closing prediction as well.

Reply. We modified the text and deleted the phrase containing the word "correction". We also deleted the Figure 12b, showing this "correction".

We do not state that we elaborated a reliable forecasting system for the ice road operation. This part of the manuscript was a "case study" aimed just a demonstration of capacity of satellite observations for the particular socioeconomic application. A development of reliable forecasting approach basing only on ~10 years of available data on the Salekhard ice road operation (i.e. observations) is out of question. This can be done in the future basing on suggested here (or different) predictors derived from the current or extended altimetric product. Several corresponding sentences were added in to the new subsection of the Discussions.

Describe how do you calculate the leading time of the forecast. Provide the information on the difference between predicted and observed dates (RMSE, max-min) and on the leading time in a consistent and systematic manner for all three cases.

Reply. We provided a table containing main statistics useful for evaluation of the  forecast. We deleted phrase "leading time" to avoid any disagreements.

541: do you mean Fig. 12a instead of 11a?

Reply. Yes, the number was modified.

541-547: you can also mention that the predicted dates are consistently earlier than the actual dates of the road opening.

Reply. We added the suggested phrase. Note, that the main phrase was moved to Discussions.

543-547: you should move this part into the Discussion

Reply. The part was moved.

546-547: maybe not so much of interannual variability but an overall trend for an earlier opening of the road demonstrated by both time series?

Reply. The phrase was modified according the suggestion.

541, 563: please be consistent – is it also RMSE for the opening date?

Reply. The table with statistics was added. The text was modified correspondingly.

Discussion

7.1 I suggest to name this subchapter "Factors affecting altimetric backscatter signal" or something like that, as it reflects better what you discuss here.

Reply. The title was changed

7.2 Here in the beginning you actually discuss retrievals of the phenology dates and ice thickness. Please consider restructuring/renaming.

You could start with ice phenology dates retrievals (manual and automatic), the factors influencing the accuracy of these retrievals, and potential improvements. Would inclusion of SAR data be beneficial for the phenology dates retrieval? Are there any SAR-based river ice phenology studies you could compare your results with? Then you could move to the ice thickness retrieval and do the same. You mention two studies in the section 4 (Unterschulz et al and Mermoz et al) which use SAR data for the river ice thickness retrieval – could you compare you results with those?

Reply. We extended and restructured this subsection starting from comparison of manual and automated routines. Then, we compared our ice thickness retrievals with similar statistics (RMSE) found in other studies (unfortunately only few provided RMSE).

The SAR data will certainly helpful for the river ice phenology refining and we will take a contact with the SAR specialists when the dedicated funding will be available.

I also suggest that you give a separate and clearly distinguished paragraph where you discuss your forecast prototypes and their viability.

Reply. The subsection was added

616, 619: wrong Figure numbers provided

Reply. The text was restructured and this sentence does not exist anymore.

617-…: again, what about the road closing date prediction?

Reply. The subsection discussing the ice road dates prediction was added

627-630: Do you mean that the detection of the first consolidated ice would require another sensor / data? If it is possible with the same dataset, why did not you try it? Please explain in the text.

Reply.  We think that this will take an additional efforts and time and likely will call for multi-satellite data, at least for validation of algorithm. Our current phenology algorithm is the result of our experience in elaboration of river water level retrievals in the Arctic that debuted more than 10 years ago. The backscatter has been used in our water level retrieval algorithm. Probably, we could use for detection of the consolidated ice the same dataset. We got several ideas recently when applying the Sentinel-3 altimeters data for the lake ice.  We are going to explore these ideas in relation to the river ice when the funding will be available.

Nevertheless, we mentioned utility of the multi-sensor approach in the text.

Could you provide for this subchapter some information on the availability of the in situ observational stations on the other Arctic rivers? That would be a great outlook on the potential future studies in a large geographical context.

Reply. We are not able to provide this information in the framework of the current manuscript. This would be possible to make such an assessment if the unified database with the station list and their coordinates were available for public access. This is not the case for Russian stations. Two publically available databases allowing to make similar assessment rapidly (Arctic-RIMS, GRDC) are dedicated to discharge stations. Their part in total number of stations is probably 1/5- 1/7.   Unfortunately, the requested assessment takes extensive river-by-river search in large geographical domain: the big Siberian rivers with the seasonal ice cover are of >3500 km length.

Conclusions – I think the title should be in plural

Reply. Modified

665-666: I would say that a generally low number of in situ observations and their general infeasibility to cover vast areas are the main drivers.

Reply. We added this phrase to the sentence.

668: freeze-up, breakup dates?

Reply. Modified

669: again, do you refer here to the automated one? Briefly reiterate please the basic principle of the algorithm.

Reply. Manual one. Details were added.

674: please mention that there are 5 stations and 12(?) years of the simultaneous measurements

Reply. Details were added.

675: what is it in the percentage of the max/average ice thickness?

Reply. 7-18%. The phrase was added.

681: mean accuracy?

Reply . The phrase was re-written, providing the RMSE values as the measure of uncertainty.

681: please reconsider what you report here based on my earlier suggestions in the Results

Reply . The phrase was re-written, providing the RMSE values as the measure of uncertainty.

683-700: very good!